# Sex differences in oncogenic mutational processes

Constance H. Li [1,2,3], Stephenie D. Prokopec [1], Ren X. Sun[1,4], Fouad Yousif[1], Nathaniel Schmitz[1], PCAWG Tumour Subtypes and Clinical Translation*, Paul C. Boutros [2,3,4,5,6,7,8] & PCAWG Consortium*

Sex differences have been observed in multiple facets of cancer epidemiology, treatment and biology, and in most cancers outside the sex organs. Efforts to link these clinical differences to specific molecular features have focused on somatic mutations within the coding regions of the genome. Here we report a pan-cancer analysis of sex differences in whole genomes of 1983 tumours of 28 subtypes as part of the ICGC/TCGA Pan-Cancer Analysis of Whole Genomes (PCAWG) Consortium. We both confirm the results of exome studies, and also uncover previously undescribed sex differences. These include sex-biases in coding and non-coding cancer drivers, mutation prevalence and strikingly, in mutational signatures related to underlying mutational processes. These results underline the pervasiveness of molecular sex differences and strengthen the call for increased consideration of sex in molecular cancer research.

[1] Computational Biology Program, Ontario Institute for Cancer Research, Toronto, ON, Canada. [2] Department of Medical Biophysics, University of Toronto, Toronto, ON, Canada. [3] Department of Human Genetics, University of California, Los Angeles, CA, USA. [4] Department of Pharmacology & Toxicology, University of Toronto, Toronto, ON, Canada. [5] Vector Institute for Artificial Intelligence, Toronto, Canada. [6] Department of Urology, University of California, Los Angeles, CA, USA. [7] Jonsson Comprehensive Cancer Center, University of California, Los Angeles, CA, USA. [8] Institute for Precision Health, University of California, Los Angeles, CA, USA. *List of authors and their affiliations appears at the end of the paper. ✉email: Paul.Boutros@mednet.ucla.edu

Sex disparities in cancer epidemiology include an increased overall cancer risk in males corresponding with higher incidence in most tumour types, even after adjusting for known risk factors[1,2]. Cancer mortality is also higher in males, due in part to better survival for female patients in many cancer types, including those of the colon and head and neck[3]. Interestingly, female colorectal cancer patients respond better to surgery[4] and adjuvant chemotherapy, though this is partially due to biases in tumour location and microsatellite instability[5]. Similarly, premenopausal female nasopharyngeal cancer patients have improved survival regardless of tumour stage, radiation or chemotherapy regimen[6]. There is a growing body of evidence for sex differences in cancer genomics[7–12], but their molecular origins and clinical implications remain largely elusive.

Previous studies have mostly focused on protein coding regions, leaving the vast majority of the genome unexplored. We hypothesise that there are uncharacterised sex differences in the non-coding regions of the genome. Using data from the Pan-cancer Analysis of Whole Genomes (PCAWG) project[13], we perform a survey of sex-biased mutations in 1983 samples (1213 male, 770 female) from 28 tumour subtypes, excluding those of the sex organs (Supplementary Data 1). The PCAWG Consortium aggregated whole-genome sequencing data generated by the ICGC and TCGA projects. These data were re-analysed with standardised, high-accuracy pipelines to align to the human genome (reference build hs37d5). Our study leverages mutation calls generated by PCAWG working groups[13–16] to identify molecular associations with sex. We exclude the X and Y chromosomes to focus on autosomal sex differences in cancers affecting both men and women, but there are known to be significant X-chromosome mutational differences between tumours arising in men and women[8]. Our analysis reveals sex differences in specific genes and in genome-wide phenomena including mutation signature activity. These sex-biases occur not only at the pan-cancer level across all 1983 tumours, but also in individual tumour subtypes.

## Results

**Sex-biases in driver genes, mutation load and tumour evolution.** We began by investigating sex differences in driver gene mutation frequencies, focusing on 165 coding and nine non-coding mutation events[14] (Supplementary Data 2). We used proportion tests to identify candidate sex-biased events with a false discovery rate (FDR) threshold of 10%. These putative sex-biased events were modelled using logistic regression (LGR) to adjust for tumour subtype-specific variables (model descriptions and variable breakdown in Supplementary Data 1). Finally, we vetted these sex-biased events in two ways: we assessed the impact of covariate imbalances in the data using repeated down-sampling analysis; we also implemented extended regression models to adjust for additional variables like stage or grade, which were only available for a greatly reduced subset of the data (see "Methods" section). We confirmed that all sex-biases remained significant under this additional scrutiny. This statistical framework formed the basis for our analysis of all genomic features.

Tumour subtype-specific sex-biased driver mutations included CTNNB1 mutation frequency in liver hepatocellular cancer (Liver-HCC), with more male-derived samples harbouring CTNNB1 mutations: (male: 31%, female: 13%, 95% CI: 8.1–28%, prop-test $q$ = 0.048, LGR $q = 1.4 \times 10^{-3}$, Fig. 1a, Supplementary Fig. 1). This mirrors our previous finding of sex-biased CTNNB1 mutation frequency in liver cancer from TCGA exome sequencing data, with similar effect sizes (male: 33% vs. female: 12%[11]). We also identified a large sex-disparity in a non-coding driver event in

thyroid cancer (Thy-AdenoCA): TERT promoter mutations were observed in 64% of male-derived samples compared with only 11% of female-derived samples (95% CI: 17–89%, prop-test $q$ = $6.9 \times 10^{-3}$, LGR $q = 0.074$, Fig. 1a, Supplementary Fig. 1), again supporting a previous finding[17]. We did not find pathogenic germline variants in TERT or CTNNB1 that might bias the detection of sex-associated somatic mutations in these genes. Other putative sex-biased events were detected, but were either not statistically significant after multivariate adjustment at present sample sizes (Supplementary Data 2), or were attributed to over-represented tumour subtypes (Supplementary Fig. 2).

Our previous work[11] found sex-biased mutation density across a number of tumour subtypes, including cancers of the liver, kidney and skin. We therefore investigated mutation density here to identify tumour subtypes where the cancer genomes of one sex accumulate more somatic single nucleotide variants (SNVs) than those of the other sex. Returning to our statistical framework, we first used Mann–Whitney $U$-tests to identify putative sex-biases, and then applied multivariate linear regression (LNR) on Box–Cox transformed mutation load to adjust for possible confounders. The Box–Cox transformation applies a power function to modify the shape of a variable's distribution to better approximate a normal distribution. It preserves monotonicity and is often applied to make data more suitable for regression analysis (see "Methods" section). We also compared the total number of somatic SNVs and further divided mutations by coding and non-coding SNVs to determine whether sex-biases may be influenced by specific genomic contexts. Across all pan-cancer samples, we identified higher mutation prevalence in male-derived samples in all three contexts (coding LNR $q = 7.3 \times 10^{-4}$, non-coding LNR $q = 6.4 \times 10^{-4}$, overall LNR $q = 1.9 \times 10^{-6}$; Supplementary Data 3). These sex-biases remained significant even after adjusting for tumour subtype, ancestry and age in multivariate analysis, and after evaluating the effects of imbalanced tumour subtype and sex sample sizes (Fig. 1b, left; Supplementary Figs. 2, 3).

We investigated somatic SNV burden in each of the 23 individual tumour subtypes with at least 15 samples ($n_{male} + n_{female} \geq 15$), applying the same statistical approach using tumour subtype-specific models (Supplementary Data 1). We found sex-biased mutation load in three tumour subtypes (Fig. 1b, right), with trending higher male coding mutation load in thyroid cancer (difference in location = 0.26 mut/Mbp, 95% CI = 0.12–0.43 mut/Mbp, $U$-test $q = 0.028$, LNR $q = 0.10$), and higher male SNV load in hepatocellular cancer and kidney renal cell cancer (Kidney-RCC) for all three genomic contexts (Supplementary Data 3). We compared the group rank differences of coding and non-coding mutation load between the sexes and found that in renal cell cancer, the differences were similar at 0.40 mut/Mbp for non-coding mutations and 0.37 mut/Mbp for coding mutations. In hepatocellular cancer however, the median sex-difference in non-coding mutation load was higher than the difference in coding mutation load (non-coding difference = 0.84 mut/Mbp vs. coding difference = 0.53 mut/Mbp). There was a similar effect for pan-cancer mutations (non-coding difference = 0.60 mut/Mbp vs. coding difference = 0.41 mut/Mbp) suggesting mutation context may have a role in sex-biased SNVs in some tumour subtypes.

On detecting sex differences in both the mutation frequency of specific drivers as well as SNV density in the same tumour subtypes, we asked whether one may bias the other. For instance, higher CTNNB1 mutation frequency in male-derived tumours may simply be due to more mutations occurring in those same samples. We therefore looked for associations between SNV burden with CTNNB1 mutation in hepatocellular cancer, and with TERT promoter mutation in thyroid cancer.

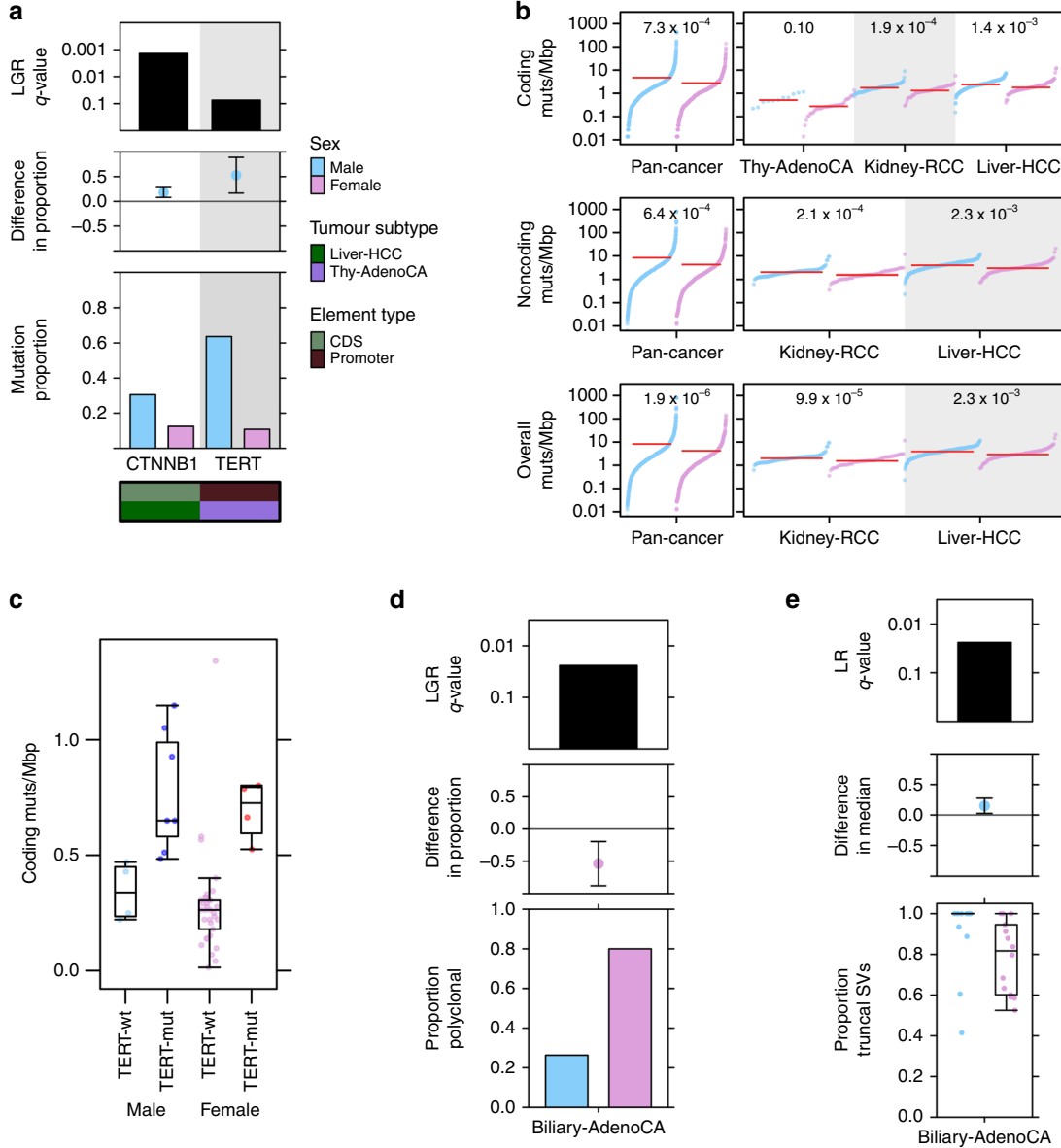

**Fig. 1 Sex-biases in mutation frequency of driver genes, SNV density and tumour evolution. a** From top to bottom, each plot shows the logistic regression *q*-value for the sex effect; difference in proportion of mutated samples between the sexes with blue denoting male-dominated bias; and mutation proportion for each gene. Covariate bars indicate mutation context and tumour subtype of interest. **b** The burden of somatic SNVs for coding, non-coding and overall mutation load. Linear regression *q*-values are shown. **c** Coding mutation load for thyroid adenocarcinoma samples compared by sex and presence or absence of *TERT* promoter mutations. **d** The proportion of polyclonal samples and **e** the proportion of truncal structural variants in biliary cancer. Tukey boxplots are shown with the box indicating quartiles and the whiskers drawn at the lowest and highest points within 1.5 interquartile range of the lower and upper quartiles, respectively. Error bars show the 95% confidence interval for the difference in proportions or medians between the sexes.

We did not find a significant association between SNV burden and *CTNNB1* mutation in hepatocellular cancer. In thyroid cancer however, *TERT* promoter mutation was associated with increased overall mutation burden (median$_{TERT-wt}$ = 0.32 mut/Mbp vs. median$_{TERT-mut}$ = 0.82 mut/Mbp, *U*-test $p = 7.9 \times 10^{-8}$). We further confirmed the association using a linear regression model (linear regression $p_{TERT} = 2.4 \times 10^{-5}$, $p_{sex} = 0.37$, Fig. 1c). To assess whether the sex-bias in *TERT* promoter mutation frequency might be due to sex-biased accumulation of SNVs, we examined tumour-matched mutation timing data generated by the PCAWG consortium[15]. We found that of eleven polyclonal samples with *TERT* promoter mutations, nine of these were earlier occurring truncal events.

We continued investigating whether sex-biased driver mutations might occur at different stages of tumour evolution between men and women and examined tumour subclonal architecture. Focusing only on thyroid tumours with *TERT* promoter mutations and liver tumours with *CTNNB1* mutations, we compared the proportions of polyclonal vs. monoclonal tumours between the sexes (Supplementary Fig. 4). We did not find sex-biased polyclonality in *TERT* promoter-mutated tumours, but did detect a putative bias in the proportion of polyclonal *CTNNB1*-mutated tumours (80% of male-derived tumours are polyclonal vs. 46% of female-derived tumours, 95% CI = −0.019–0.70, prop-test $p = 0.039$). We therefore accounted for polyclonality when comparing the timings of the mutations in these driver events. On

subsequently examining the frequency of clonal vs. subclonal driver mutation events between the sexes, we found that while there were differences in the proportions of truncal mutations (e.g. 100% of *TERT* promoter mutations were truncal events in male-derived vs. 50% truncal events in female-derived thyroid cancer patients), no comparisons were statistically significant.

We expanded our clonality analysis to perform a general survey of clonal structure and mutation timing across all tumour subtypes and mutations (Supplementary Data 4). We found that female-derived biliary adenocarcinoma (Biliary-AdenoCA) tumours were frequently polyclonal, whereas most male-derived tumours were monoclonal (26% male-derived samples are polyclonal vs. 80% female-derived, 95% CI = 19–88%, prop-test $q = 0.063$, LGR $q = 0.026$; Fig. 1d). In addition, we found intriguing evidence suggesting there may be sex differences in the mutation timing of structural variants (SVs) in this tumour subtype. Structural variants in male-derived samples were more frequently truncal events than in female-derived samples (median male percent truncal SVs = 100% vs. median female = 82%, 95% CI = 0.9–32%, $U$-test $q = 0.081$, LNR $q = 8.6 \times 10^{-3}$; Fig. 1e). Though other comparisons did not reach our statistical significance threshold, we found some interesting trends that may merit future study, including in oesophageal cancer (Eso-AdenoCA) where SVs in female-derived samples were more frequently truncal events while SVs in male-derived samples occurred more frequently in subclones (median male percent truncal SVs = 55%, median female = 100%; Supplementary Fig. 5), and in medulloblastoma, where insertion-deletions (indels) were more frequently truncal events in female-derived samples than male (median male percent of truncal indels = 65%, median female proportion of truncal indels = 70%; Supplementary Fig. 6). Our analysis of sex differences in tumour evolution identified some sex-biased events and hint at putative sex-biases that should be further explored in future analyses.

**Sex-biases in genome instability and CNAs.** Next, we examined percent genome altered (PGA), which provides a summary of copy number aberration (CNA) load. A proxy for genome instability, PGA is a complementary measure of mutation density to somatic SNV burden. Although we did not find associations between sex and autosome-wide PGA, we observed sex-biases in the copy number burden for specific chromosomes (Fig. 2a). In pan-cancer analysis, male-derived samples exhibited a slight but significant higher percent chromosome altered for chromosome 7 even after accounting for tumour subtype, ancestry and age (median male PGA-7 = 5.4%, median female PGA-7 = 0.37%, 95% CI = $9.4 \times 10^{-4}$–$2.4 \times 10^{-3}$%, $U$-test $q = 5.0 \times 10^{-3}$, LNR $q = 0.024$; Supplementary Data 5). In individual tumour subtypes, we found sex-biased PGA in renal cell cancer (chromosomes 7 & 12) and hepatocellular cancer (chromosomes 1 & 16). On further scrutinising these sex-PGA associations using extended models, we found that grade was a likely confounder in renal cell cancer, though the sex effect after correcting for this variable was still trending (extended LNR $q = 0.17$). By looking at copy number gains and losses separately, we additionally identified chromosomes with sex-biases in the burden of copy number gains and losses (Supplementary Fig. 7 and Supplementary Data 5), including sex-biased percent copy gained on chromosomes 5, 8 and 17 in pan-cancer tumours. These biases in chromosome instability were robust to imbalanced sex sample sizes (Supplementary Fig. 8).

We next compared CNA frequency on the gene level to identify specific genes lost or gained at sex-biased rates. Across all pan-cancer samples, we found 2,502 genes with sex-biased gains across 13 chromosomes (LGR $q$-value < 10%, Fig. 2b,

Supplementary Data 6, 7, Supplementary Figs. 2, 9), These genes were all more frequently gained in male-derived samples than female, with differences in copy number gain frequency up to 10% on chromosomes 7 and 8. Genes with male-dominated copy number gains include the oncogene *MYC* (male gain frequency = 37% vs. female gain frequency = 28%, 95% CI = 5.2–14%, prop-test $q = 2.5 \times 10^{-3}$, LGR $q = 0.068$). The driver *CTNNB1* was also more frequently gained in male samples (male gain frequency = 8.9% vs. female gain frequency = 5.2%, 95% CI = 1.4–6.1%, prop-test $q = 0.016$, LGR $q = 0.053$). We did not find pan-cancer sex-biased copy number losses.

We repeated this analysis for every tumour subtype independently and found sex-biased CNAs in renal cell and hepatocellular cancer (Supplementary Data 6 and 7). In renal cell cancer (Kidney-RCC), 1,986 sex-biased gains all occurred more frequently in male-derived samples, with differences in frequency up to 35% (Fig. 2c). They spanned across chromosomes 7 and 12, agreeing with our finding of male-dominated genome instability in these chromosomes (Fig. 2a; Supplementary Fig. 7). Using an extended renal cell cancer model accounting for grade, we obtained a high confidence set of 969 genes altered by sex-biased gains (extended model $q < 0.1$), with the remaining 1017 genes having a trending sex effect (extended model $q < 0.17$). In contrast to the male-dominated gains in pan-cancer and renal cell findings, we found higher female frequency of copy number losses in hepatocellular cancer (Fig. 2d). We identified 2226 genes with higher copy number loss rates in female-derived samples. As observed in renal cell cancer some of these losses span whole chromosomes, in this case chromosomes 3 and 16. Extended modelling in Liver-HCC incorporating stage and grade resulted in a list of 1797 high confidence sex-biased genes (extended model $q < 0.1$).

The concurrence between sex-biased PGA and gene-specific events in renal cell and hepatocellular cancer suggested that PGA could be used to guide identification of additional sex-biased CNAs on the gene level. We more closely examined regions of interest in tumour subtypes of that did not have sex-biased CNAs in our general CNA analysis, but did have putatively sex-biased genome instability ($U$-test $q < 0.2$): biliary cancer, B-cell non-Hodgkin lymphoma (Lymph-BNHL), and chronic lymphocytic leukaemia (Lymph-CLL). We identified an additional 203 genes on the p-arm of chromosome 8 that were more frequently lost in female-derived biliary tumours (Supplementary Fig. 10). These copy number losses were 50% more common in female-derived tumours and affect genes such as *DLC1*, a known tumour suppressor gene in hepatocellular cancer that is thought to have a similar role in gallbladder cancer[18]. Although we did not identify additional sex-biased CNAs in non-Hodgkin lymphoma, chronic lymphocytic leukaemia or melanoma, our sex-biased PGA results suggest these as regions of interest for future work.

**Sex-biases in mutational signatures.** We hypothesised that sex differences in mutation density and tumour evolution characteristics might be driven by sex differences in mutational processes. In addition to single base substitution (SBS) signatures, which have been well annotated and linked to tumour aetiology[19,20], we also examined doublet base substitution (DBS) and small insertion-deletion (ID) signatures. Sex differences in a mutational signature could shine insight on molecular differences between the sexes. For each of 47 validated PCAWG SBS, 11 DBS and 17 ID signatures[16], we performed a two-stage analysis. We first compared the proportions of signature-positive samples between the sexes; that is, we looked at the proportions of samples with any mutations attributed to the signature to determine

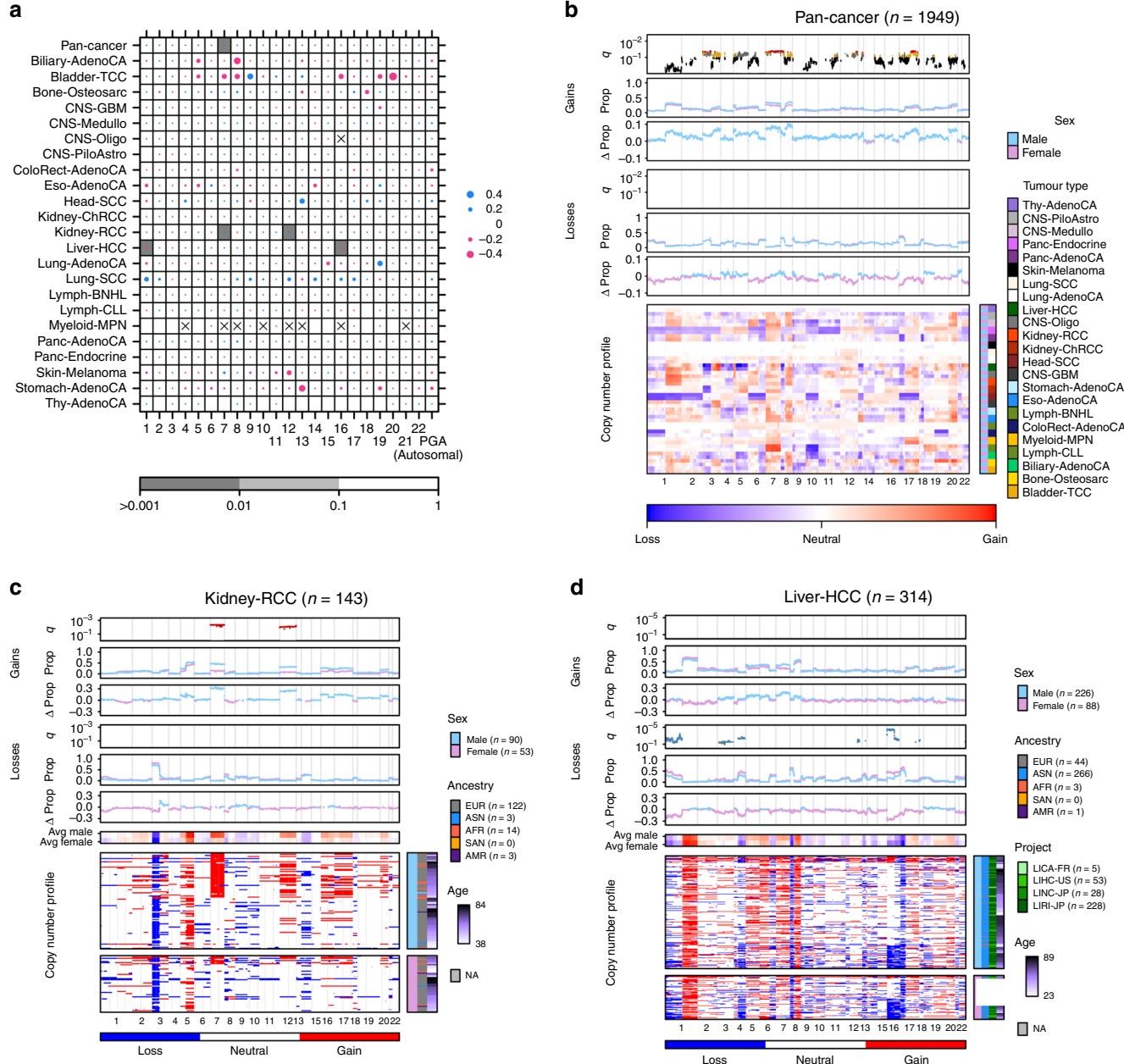

**Fig. 2 Sex-biases in percent chromosome altered are reflected in gene-specific events. a** A summary of associations between sex and genome instability across tumour subtypes. Dot size shows difference in median percent genome altered or percent chromosome altered between the sexes. Dot colour shows direction of bias, with blue indicating higher instability in male-derived tumours and pink indicating higher instability in female-derived tumours. Background shading shows *q*-values from multivariate linear regression. Sex differences in CNAs for **b** pan-cancer, **c** kidney renal cell cancer, and **d** hepatocellular cancer. Each plot shows, from top to bottom: the *q*-value showing significance of sex from multivariate linear modelling with yellow/green points corresponding to 0.1 < *q* < 0.05, deep blue/red points corresponding to *q* < 0.05, and grey points indicating hits that were attributed to covariate sample size imbalances and rejected; the proportion of samples with aberration; the difference in proportion between male and female groups for copy number gain events; the same repeated for copy number loss events; and the copy number aberration (CNA) profile heatmap. The columns represent genes ordered by chromosome. Light blue and pink points represent data for male- and female-derived tumours respectively.

whether there was a relationship between each signature and sex. Then, we focused on signature-positive samples and compared the percentage of mutations attributed to each signature between the sexes to assess relative signature activity. For both analyses we used univariate techniques to identify putative events, adjusted for additional variables using linear models with SNV density as a variable, and compared the distributions of attributed mutations with Kolmogorov–Smirnov tests. We also evaluated hits using the added scrutiny of down-sampling and extended regression models (see "Methods" section; Supplementary Figs. 11, 12).

At the pan-cancer level, we identified three signatures that occurred more frequently in one sex over the other (Fig. 3a; Supplementary Data 8). SBS1 was more commonly detected in female-derived samples (88% of male-derived vs. 97% of female-derived, $\chi^2$-test $q = 9.2 \times 10^{-10}$, LGR $q = 3.0 \times 10^{-6}$) and was also associated with higher signature activity in these samples (male median percent mutations attributed to SBS1 = 8.6%, female median = 10%, *U*-test $q = 5.5 \times 10^{-3}$, LNR $q = 0.059$). Conversely, signatures SBS17a and SBS17b were detected in a larger proportion of male-derived samples (16% of male-derived vs.

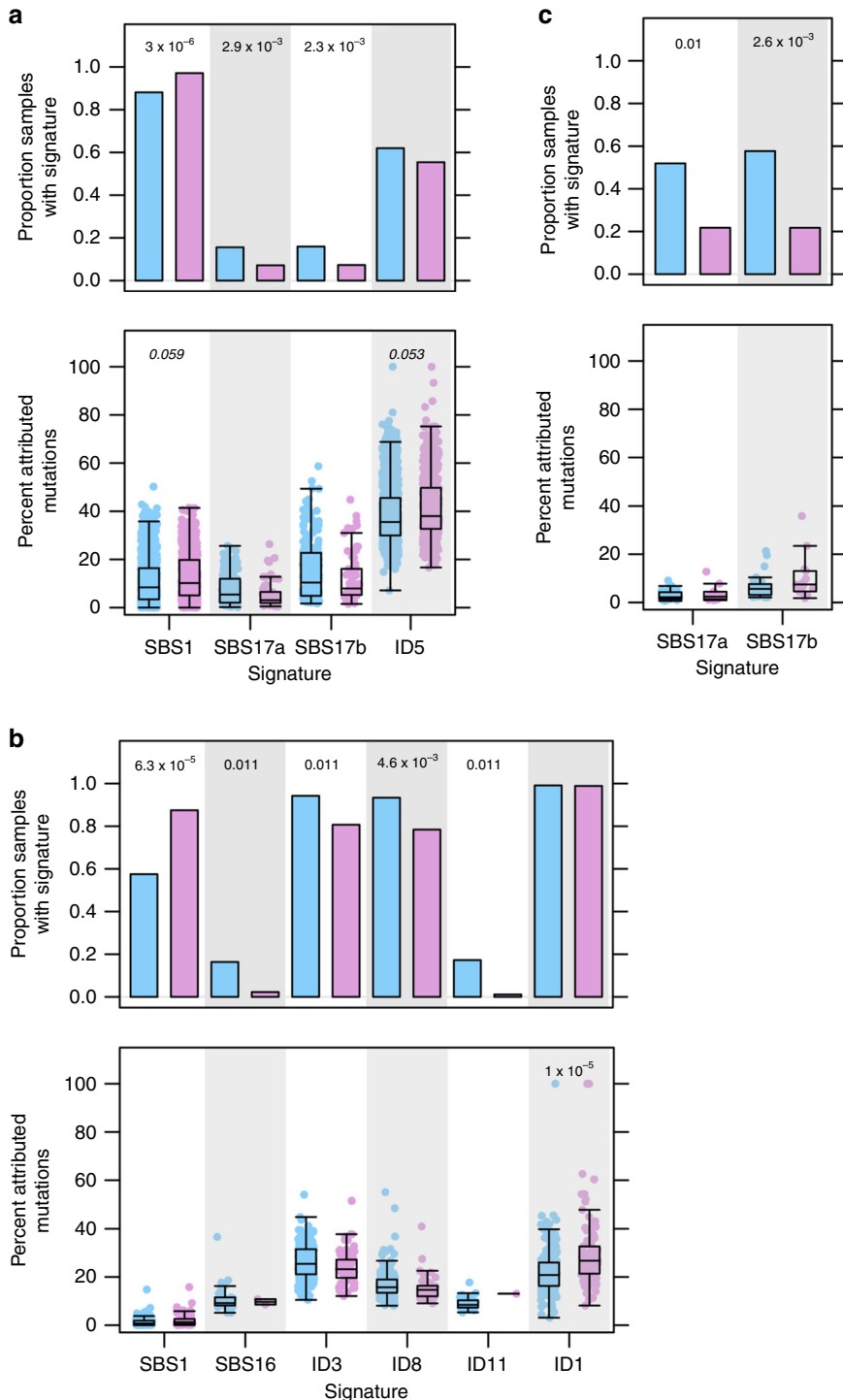

**Fig. 3 Sex differences in mutational signatures related to mutational processes.** Comparisons between proportions of signature-positive samples (top) and signature activity (bottom) for **a** pan-cancer comparisons, **b** liver hepatocellular cancer, and **c** B-cell non-Hodgkin lymphoma. FDR-adjusted q-values for multivariate logistic regression (top) and multivariate linear regression (bottom) shown only for significant comparisons. Blue shows male- and pink shows female-derived tumours. Tukey boxplots are shown with the box indicating quartiles and the whiskers drawn at the lowest and highest points within 1.5 interquartile range of the lower and upper quartiles, respectively.

7.2% of female-derived). SBS1 is thought to be caused by deamination of 5-methylcytosine to thymine, resulting in base substitutions. Signatures SBS17a and SBS17b are of unknown aetiology. We also identified a sex-bias in indel signature ID5, which had higher activity in female-derived tumours (male median percent attributed mutations = 35%, female median = 38%, $U$-test $q = 1.1 \times 10^{-3}$, LNR $q = 0.053$). ID5 mutations are

clock-like and may accumulate in normal cells. Both SBS1 and ID5 are correlated with age, but our multivariate model accounts for this variable and sex-bias remains significant.

Since mutational processes are disease-specific, we repeated the mutational signatures analysis in each tumour subtype. We identified six sex-biased signatures in hepatocellular cancer (Fig. 3b; Supplementary Data 8). We again detected a

female-dominated bias in the proportion of SBS1-positive samples (58% of male-derived vs. 88% of female-derived, $\chi^2$-test $q = 3.5 \times 10^{-5}$, LGR $q = 6.3 \times 10^{-5}$). We also detected a male-dominated bias in SBS16 (16% of male-derived vs. 2.2% of female-derived, $\chi^2$-test $q = 9.8 \times 10^{-3}$, LGR $q = 0.011$). A previous study[21] described this sex-biased signature and an association between more CTNNB1 mutations and higher activity of SBS16 in an independent dataset; these findings agree with what we report here for PCAWG data. There were also four sex-biased ID signatures in hepatocellular cancer: ID3 (94% of male-derived vs. 81% of female-derived, $\chi^2$-test $q = 5.0 \times 10^{-3}$, LGR $q = 0.011$), ID8 (93% of male-derived vs. 78% of female-derived, $\chi^2$-test $q = 3.5 \times 10^{-3}$, LGR $q = 4.6 \times 10^{-3}$) and ID11 (17% of male-derived vs. 1.1% of female-derived, $\chi^2$-test $q = 3.5 \times 10^{-3}$, LGR $q = 0.011$) occurred more frequently in male-derived samples. ID3 is associated with tobacco smoke, and ID8 with double-stranded break repair. ID11 has unknown aetiology. Although ID1 was detected at similar rates between the sexes, a greater proportion of ID1-attributed mutations were found in female-derived than male-derived samples (male median percent mutations attributed to ID1 = 21%, female median = 27%, U-test $q = 2.4 \times 10^{-6}$, LR $q = 1.0 \times 10^{-5}$). Using our extended hepatocellular model to further scrutinise these signatures, we found that all remained sex-biased after accounting for these variables except in ID3, where the effect was trending (extended model $q$-value = 0.12). Mutations associated with ID1 are thought to result from slippage during DNA replication and are associated with defective DNA mismatch repair, suggesting that while male- and female-derived tumours harbour defective DNA repair at similar rates, it is responsible for a larger proportion of mutations in female-derived tumours. Taken together, sex-biases in the aetiology underlying the molecular landscape of hepatocellular cancer begin to emerge. In this tumour subtype, spontaneous or enzymatic deamination of 5-methylcytosine to thymine and defective mismatch repair occur more frequently in female patients and are also responsible for more mutations. Conversely, tobacco smoking is more common in male patients though the number of mutations attributed to tobacco smoke is not different between the sexes; this leads to more tobacco-associated male hepatocellular tumours.

In B-cell non-Hodgkin lymphoma, we identified significant differences in the proportions of SBS17a- and SBS17b-positive tumours (Fig. 3c; Supplementary Data 8). More male-derived samples had mutations associated with these signatures. There were also several intriguing sex differences in mutational signatures that did not meet our significance threshold. For instance in thyroid cancer, DBS2 accounts for a higher percentage of mutations in male-derived samples (male median percent mutations attributed to DBS2 = 50%, female median = 33%, Supplementary Data 8). The association of DBS2 with tobacco smoking suggests that future insight in this signature may provide molecular explanations for the sex-specific associations between smoking and thyroid cancer risk[22]. As the aetiologies of these mutational signatures become better known, we can better understand how underlying mutational processes lead to molecular sex-biases. We may also be able to discern environmental and lifestyle factors even in the absence of reported data, allowing us to better account for confounding factors.

Finally, to ensure that our findings were not skewed by differences in sequencing quality, we checked for sex-biases in quality control (QC) metrics. These included comparing the coverage, read length, and overall quality summaries of both tumour and normal genomes. We mirrored our main analyses and used Mann–Whitney U-tests or $\chi^2$ tests and linear modelling to check each QC metric. We did not find sex-biases in any QC metric in pan-cancer or tumour subtype analysis after multiple adjustment except in raw somatic mutation calling (SMC)

coverage. SMC coverage was higher in male-derived samples in six tumour subtypes including thyroid cancer and oesophageal cancer, and was higher in female-derived samples in lung adenocarcinoma and B-cell non-Hodgkin lymphoma (Supplementary Data 9 and Supplementary Fig. 13). Although we do not find sex differences in comparing the SMC coverage pass/fail rates using a recommended minimum of 2.6 gigabases covered, it is prudent to consider sex-biased SMC in relation to our findings.

## Discussion

Our analysis of whole-genome sequencing data from the PCAWG project uncovered sex differences in the largely unexplored non-coding autosomal genome. In addition to validating previously reported findings in a novel dataset, we present sex-biases in measures of non-coding mutation density, tumour evolution and mutation signatures. These sex-biases suggest differences in the origins and trajectories of tumours between men and women, and that they are influenced by different endogenous and environmental factors. Although many of our findings describe pan-cancer differences, we have also uncovered an intriguing glimpse into tumour subtype-specific differences in cancers such as those of the liver and kidney.

These results should be taken within context of a number of caveats. As we use techniques like the Box–Cox transformation to make the data better suited for our statistical methods, there are likely characteristics that our models are unable to account for. An alternate approach using robust modelling may be better suited for future analyses. Secondly, the tumour subtype-specific results are bound by subtype sample size, and lack of annotation data restricts the ability to account for confounding variables. It is therefore important to consider these results within context of the multivariable models used, which do not directly capture characteristics such as tobacco smoking history. Many of our core multivariate regression models omit stage and grade due to a large number of missing values. We follow up this core regression with extended modelling as an additional level of scrutiny. Although these extended models do include stage or grade, they are run on a much smaller (up to 50%) subset of the data and there is a corresponding loss of statistical power. Finally, there are imbalances across covariate sample sizes, such as over-representation of some tumour types in pan-cancer analysis. We evaluated these imbalances using down-sampling analysis and rejected results that were biased by these imbalances. Nevertheless, pan-cancer analysis is dependent on the tumour subtypes included in the cohort and some findings may reflect subtype-specific trends rather than general characteristics across all cancers.

Future increases in sample size and robust associated annotation will allow for the detection of smaller effects and the control of more confounders. Such large datasets are critical in validating the preliminary findings we have described in this study. Increasing the diversity of donors will also allow the study of intriguing cross-variable questions such as investigating whether sex differences are universal across races, or if there are race-specific sex differences. Our results are based on single region sequencing, which can bias the clonal reconstruction for these tumours. Future work sampling multiple regions will allow us to detect sex differences in more precise reconstructions at a greater resolution. We will also be able to leverage germline data to assess whether there are sex-biases in inherited variants that affect the variants we observe in somatic mutation profiles.

Nevertheless, our analyses of driver genes and copy number alterations suggest functional impacts of genomic sex-biases on the transcriptome and tumorigenesis. By using signatures to distinguish between mutations attributed to lifestyle factors such

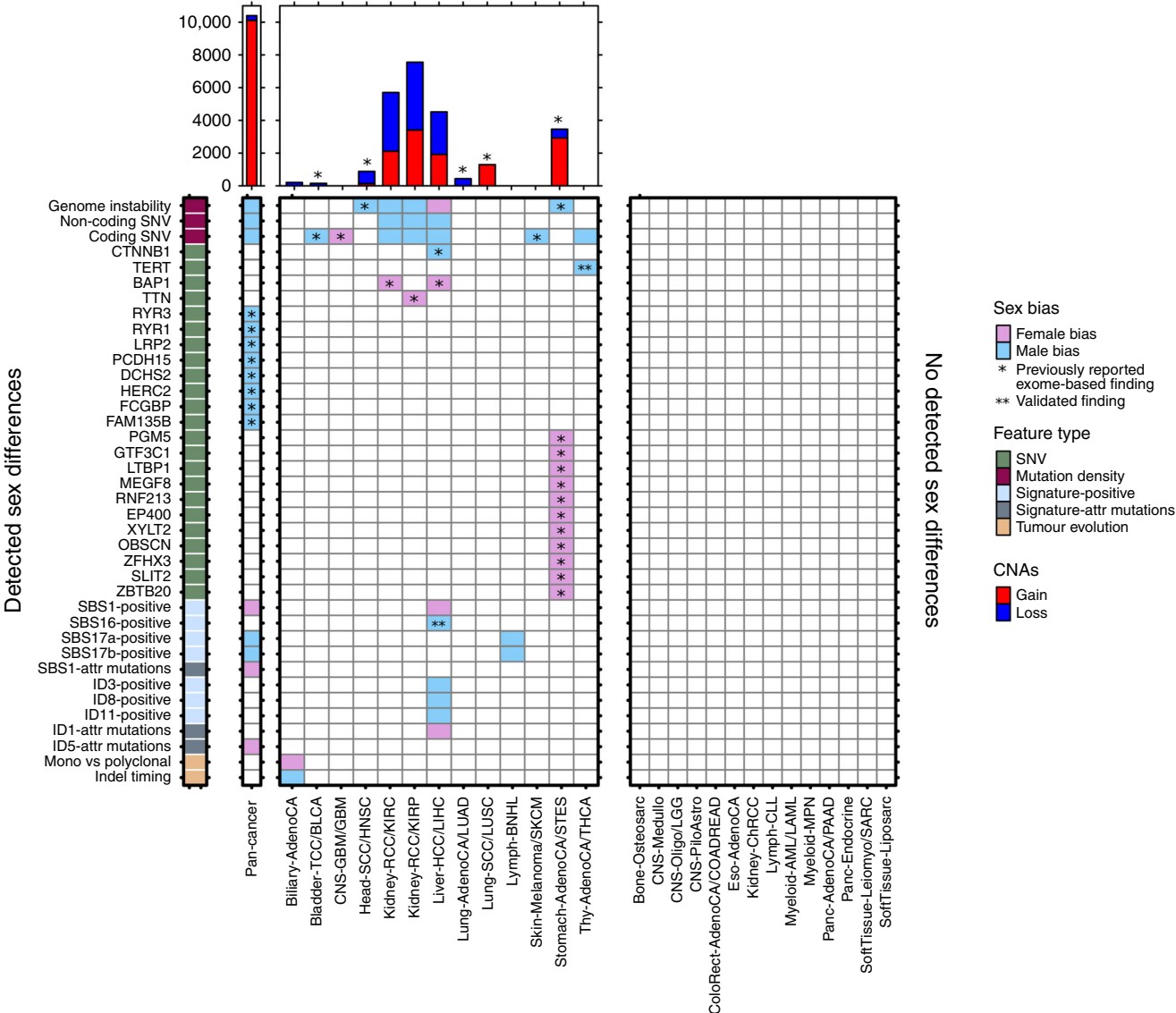

**Fig. 4 The landscape of sex differences in cancer genomics.** Summary of genomic features found to be sex-biased in pan-cancer analysis or in specific tumour subtypes. Results from both PCAWG and TCGA analyses are shown. Direction of sex-bias is shown in coloration denoting which sex has higher or more frequent aberration of the genomic feature. Top plot shows union of genes found to be involved in sex-biased CNAs. Starred indicate findings exclusively from exome sequencing data ($n = 7131$), un-starred indicate findings from PCAWG data ($n = 1983$), and double-starred indicate findings also described in other studies.

as smoking history, we can better describe sex differences related to biological factors such as hormone activity. And despite low tumour subtype-specific sample numbers, our mutation timing and mutational signatures findings at both the pan-cancer and tumour-subtype level hint at underlying mutational processes that may give rise to molecular sex-biases. Combined with our previous work in whole-exome sequencing, we present a landscape of sex-biases in cancer genomics and mutational processes (Fig. 4).

It is becoming clear that sex differences occur across many mutation classes and the portrait of differences for each tumour subtype is a unique reflection of active mutational processes and tumour evolution. We have performed here a pan-cancer analysis of sex differences in whole-genome sequencing data and catalogued previously undescribed sex-biases. However, increased study of molecular sex differences in future large-scale sequencing efforts is needed to strengthen the findings we present here, to determine why men and women have molecularly different

tumours, and to determine how this information can be leveraged to improve patient care.

## Methods

**General statistical framework**. We only included non-sex-specific tumour subtypes in our analysis and focused on the autosome, excluding the sex chromosomes. Covariate data include genomically matched sex, age at diagnosis, and imputed ancestry.

For each genomic feature of interest, we performed three stages of analysis. At stage one, we use non-parametric univariate tests (Pearson's $\chi^2$ proportion or Mann–Whitney $U$-test) first, followed by false discovery rate adjustment to identify putative sex-biases of interest ($q < 0.1$).

At stage two, we further investigate these putative sex-biases by using multivariate linear or logistic modelling to account for potential confounders using bespoke models for each tumour subtype. Confounders were included as independent variables in each model. Supplementary Data 1 describes the model variables for each tumour context, as well as detail on when analyses included multivariate modelling. Variables were included based on availability of data (<15% missing), sufficient variability (at least two levels) and collinearity.

Discrete data were modelled using logistic regression. Continuous data were first transformed using the Box–Cox family and modelled using linear regression.

The Box–Cox family of transformations is a formalised method to select a power transformation to better approximate a normal-like distribution and stabilise variance. We used the Yeo–Johnson extension to the Box–Cox transformation that allows for zeros and negative values[23]:

$$
y_i^{\lambda} = \begin{cases}
\frac{(y_i+1)^{\lambda}-1}{\lambda}, & \text{if } \lambda \neq 0,\ y \geq 0 \\
\log(y_i + 1), & \text{if } \lambda = 0,\ y \geq 0 \\
-\frac{(-y_i+1)^{2-\lambda}-1}{2-\lambda}, & \text{if } \lambda \neq 2,\ y < 0 \\
-\log(-y_i + 1), & \text{if } \lambda = 2,\ y < 0
\end{cases}.
$$

FDR adjustment was performed for $p$-values for the sex variable significance estimate and an FDR threshold of 10% was used to determine statistical significance. For some tumour subtypes, the multivariate step is never performed because there are no univariate hits to evaluate.

The third stage of analysis involves re-evaluating our stage two sex-biases with a battery of additional modelling:

For pan-cancer findings, we evaluate the effect of unbalanced tumour subtype sample sizes by repeatedly and randomly down-sampling to the median subtype sample size with replacement ($n_{median} = 48$). For each down-sampled dataset, we record the difference between the male and female median/proportion, as well as the $p$-value from the relevant univariate test (Supplementary Fig. 2). We repeat this 10,000 times for each finding to generate distributions of male–female differences and $p$-values. We calculate a 95% confidence interval using the male–female difference distribution and reject findings where this confidence interval overlaps with 0. We also reject findings where the median down-sampled $p$-value is greater than the $p = 0.05$ threshold.

For both pan-cancer and tumour subtype-specific findings, we evaluate the effect of unbalanced sexes when either female or male donors account for >60% of samples. We down-sample to the smaller number of samples with replacement and record the difference between the male and female median/proportion, as well as the $p$-value from the relevant univariate test (Supplementary Figs. 1, 3, 8, 9 and 11). We repeat this 10,000 times for each finding to generate distributions of male–female differences and $p$-values. We calculate a 95% confidence interval using the male–female difference distribution and reject findings where this confidence interval overlaps with 0. We also reject findings where the median down-sampled $p$-value is greater than the $p = 0.05$ threshold. We present the median down-sampled $p$-values throughout Supplementary Data 2–8.

For tumour subtype-specific results, we also use extended models that incorporate additional variables such as tumour stage. Because this leads to up to 50% data loss, we only investigate a subset of results in this way. All extended modelling results are presented in Supplementary Data 2–8.

Specific details are provided for each analysis below.

**Driver event analysis.** We focused on driver events (syn11639581) described by the PCAWG consortium[14]. Driver mutation data were binarized to indicate presence or absence of the driver event in each patient. For the first stage of our analysis, we compared proportions of mutated genes between the sexes using univariate proportion tests. A $q$-value threshold of 0.1 was used to select genes for further multivariate analysis in stage two using binary logistic regression. FDR correction was again applied and genes with significant pan-cancer sex terms were extracted from the models ($q$-value < 0.1). Driver event analysis was performed separately for pan-cancer analysis and for each tumour subtype.

**Clonal structure and mutation timing analysis.** Subclonal structure and mutation timing calls[15] were downloaded from Synapse (syn8532460). Subclonal structure data were binarized from number of subclonal clusters per sample to monoclonal (one cluster) or polyclonal (more than one cluster). The proportion of polyclonal samples was calculated per sex and compared in the first stage of analysis using proportion tests for both pan-cancer and tumour subtype analysis. The univariate $p$-values were FDR-adjusted across all tumour subtypes to identify putatively sex-biased clonal structure. These cases were further scrutinised in stage two using logistic regression. A multivariate $q$-value threshold of 0.1 was used to determine statistically significant sex-biased clonal structure.

Mutation timing data classified SNVs, indels and SVs into clonal (truncal) or subclonal groups. The proportion of truncal variants was calculated for each mutation type ($\frac{\text{Number truncal SNVs}}{\text{Total SNVs}}$, etc) to obtain proportions of truncal SNVs, indels and SVs for each sample. These proportions were compared in stage one of analysis between the sexes using two-sided Mann–Whitney $U$-tests and univariate $p$-values were FDR-adjusted to identify putatively sex-biased mutation timing. In stage two, linear regression was used to adjust for confounding factors and a multivariate $q$-value threshold of 0.1 was used to determine statistically significant sex-biased mutation timing. The mutation timing analysis was performed separately for SNVs, indels and SVs.

**SNV density analysis.** Consensus SNV calls were downloaded from Synapse (syn7357330). Overall SNV density per patient was calculated as the sum of SNVs across all genes on the autosomes and scaled to mutations/Mbp. Coding mutation prevalence only considers the coding regions of the genome, and non-coding prevalence only considers the non-coding regions. Mutation load was first

compared between the sexes using Mann–Whitney $U$-tests for both pan-cancer and tumour-type specific analysis. Comparisons with $U$-test $q$-values meeting an FDR threshold of 10% were further analysed using linear regression to adjust for tumour subtype-specific variables. Mutation load analysis was performed separately for each mutation context, with pan-cancer and tumour subtype $p$-values adjusted together.

**Chromosome and genome instability analysis.** Consensus copy number data were obtained from Synapse (syn8042988). Ploidy-adjusted calls were used to identify segments with copy number gains and losses. The number of bases in copy number gained or lost segments were summed per chromosome and divided by chromosome size to obtain percent chromosome gained and lost, respectively. All segments affected by any copy number aberration were also summed and treated in the same way to calculate percent chromosome altered. Percent copy number gained, lost and altered were also calculated over the autosomes. In stage one, genome and chromosome instability were compared in pan-cancer and tumour-subtype analysis using Mann–Whitney $U$-tests to identify putatively sex-biased chromosome and genome instability. In stage two, putatively sex-biased events were further analysed using linear regression modelling. Genome instability analysis was performed separately for each tumour subtype with FDR adjustment performed over percent copy gained, loss and altered comparisons together.

**Genome-spanning CNA analysis.** Consensus copy number data (syn8042988) were processed to gain/neutral/loss calls per gene. For each gene, we compared the proportion of gains for each sex using proportion tests. For putative sex-biased genes that passed an FDR threshold of 10%, we followed up with multivariate logistic regression to adjust for tumour subtype-specific covariates (Supplementary Data 1). We repeated this analysis for copy number loss. This genome-spanning analysis was performed separately for losses and gains for each tumour subtype.

**Mutational signatures analysis.** The number of mutations attributed to each SBS (syn11738669), DBS (syn11738667) and ID (syn11738668) signature[16] per sample was downloaded from Synapse. For each signature, we compared the proportion of samples with any mutations attributed to the signatures ("signature-positive") using $\chi^2$-square tests to identify univariately significant sex-biases. Signatures with putative sex-biases were further analysed using logistic regression.

We also compared the proportions of mutations attributed to each signature. The numbers of mutations per signature were divided by total number of mutations for each sample to obtain the proportion of mutations attributed to the signature. In the first stage of analysis, we used Mann–Whitney $U$-tests to compare these proportions of attributed mutations, and Kolmogorov–Smirnov tests to compare their distributions between the sexes. Putative sex-biased signatures were further analysed using linear regression after Box–Cox adjustment.

In addition to tumour subtype-specific covariates, we included SNV density in all multivariate mutational signatures models to account for bias in calling more signatures in SNV-dense samples. Signatures that were not detected in a tumour subtype were omitted from analysis for that tumour subtype. We also used Kolmogorov–Smirnov tests to compare the distributions of attributed mutations and kept results where the sex-difference was significant or trending.

**Reporting summary.** Further information on research design is available in the Nature Research Reporting Summary linked to this article.

## Data availability

Somatic and germline variant calls, mutational signatures, subclonal reconstructions, transcript abundance, splice calls and other core data generated by the ICGC/TCGA Pan-cancer Analysis of Whole Genomes Consortium are described in the marker paper[13] and available for download at https://dcc.icgc.org/releases/PCAWG. Additional information on accessing the data, including raw read files, can be found at https://docs.icgc.org/pcawg/data/. In accordance with the data access policies of the ICGC and TCGA projects, most molecular, clinical and specimen data are in an open tier that does not require access approval. To access potentially identification information, such as germline alleles and underlying sequencing data, researchers will need to apply to the TCGA Data Access Committee (DAC) via dbGaP for access to the TCGA portion of the dataset, and to the ICGC Data Access Compliance Office (DACO) for the ICGC portion. To access somatic single nucleotide variants derived from TCGA donors, researchers will also need to obtain dbGaP authorisation. In addition, the analyses in this paper used a number of datasets that were derived from raw PCAWG sequencing data and variant calls (Supplementary Data 10). The individual datasets are available at Synapse (https://www.synapse.org/), and are denoted with synXXXXX accession numbers (listed under Synapse ID); all these datasets are also mirrored at https://dcc.icgc.org, with full links, filenames, accession numbers and descriptions detailed in Supplementary Data 10. Tumour histological classifications were reviewed and assigned by the PCAWG Pathology and Clinical Correlates Working Group (annotation version 9; syn10389158, syn10389164). Ancestry imputation was performed using an ADMIXTURE[24]-like algorithm by the PCAWG Germline Cancer Genome Working Group based on germline SNP profiles

determined by whole-genome sequencing of the reference sample and are available in Supplementary Table 1 of the PCAWG marker paper[13]. The consensus somatic SNV and indel (syn7357330) file covers 2778 whitelisted samples from 2583 donors. Driver events were called by the PCAWG Drivers and Functional Interpretation Group (syn11639581). Consensus CNA calls from the PCAWG Structural Variation Working Group were downloaded in VCF format (syn8042988). Subclonal reconstruction was performed by the PCAWG Evolution and Heterogeneity Working Group (syn8532460). SigProfiler mutation signatures were determined by the PCAWG Mutation Signatures and Processes Working Group for single base substitution (syn11738669), doublet base substitution (syn11738667) and indel (syn11738668) signatures.

## Code availability

The core computational pipelines used by the PCAWG Consortium for alignment, quality control and variant calling are available to the public at https://dockstore.org/search?search=pcawg under the GNU General Public License v3.0, which allows for reuse and distribution. All statistical analyses and data visualisation were performed in the R statistical environment (v3.4.3) using the BPG[25] (v5.9.8) and car (v3.0-2) packages.

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

## Acknowledgements

We thank all the members of the Boutros lab for insightful discussions. This study was conducted with the support of the Ontario Institute for Cancer Research to P.C.B. through funding provided by the Government of Ontario. This work was supported by the Discovery Frontiers: Advancing Big Data Science in Genomics Research program, which is jointly funded by the Natural Sciences and Engineering Research Council (NSERC) of Canada, the Canadian Institutes of Health Research (CIHR), Genome Canada and the Canada Foundation for Innovation (CFI). P.C.B. was supported by a Terry Fox Research Institute New Investigator Award and a CIHR New Investigator Award. This work was supported by an NSERC Discovery grant and by Canadian Institutes of Health Research, grant #SVB-145586, to P.C.B. This work was supported by the NIH/NCI under award number P30CA016042 and an operating grant from the National Cancer Institute Early Detection Research Network (1U01CA214194-01). We acknowledge the contributions of the many clinical networks across ICGC and TCGA who provided samples and data to the PCAWG Consortium, and the contributions of the Technical Working Group and the Germline Working Group of the PCAWG Consortium for collation, realignment and harmonised variant calling of the cancer genomes used in this study. We thank the patients and their families for their participation in the individual ICGC and TCGA projects.

## Author contributions

C.H.L. and P.C.B. initiated the project. C.H.L., S.D.P., R.X.S., F.Y. and N.S. analysed data. P.C.B. supervised research. C.H.L. and P.C.B. wrote the first draft of the manuscript, which all authors edited and approved. The PCAWG Tumour Subtypes and Clinical Translation working group and the PCAWG Consortium network provided variant calls, clinical annotation data and insightful commentary.

## Competing interests

The authors declare no competing interests.

## Additional information

# PCAWG Tumour Subtypes and Clinical Translation

Fatima Al-Shahrour[9], Gurnit Atwal[1,5,10], Peter J. Bailey[11], Andrew V. Biankin[11,12,13,14], Paul C. Boutros[1,2,3,4], Peter J. Campbell[15,16], David K. Chang[11,12], Susanna L. Cooke[11], Vikram Deshpande[17], Bishoy M. Faltas[18], William C. Faquin[17], Levi Garraway[19], Gad Getz[20,21,22,23], Sean M. Grimmond[24], Syed Haider[1], Katherine A. Hoadley[25,26], Wei Jiao[1], Vera B. Kaiser[27], Rosa Karlić[28], Mamoru Kato[29], Kirsten Kübler[17,20,21], Alexander J. Lazar[30,31], Constance H. Li[1,2], David N. Louis[17], Adam Margolin[32], Sancha Martin[15,33], Hardeep K. Nahal-Bose[34], G. Petur Nielsen[17], Serena Nik-Zainal[15,35,36,37], Larsson Omberg[38], Christine P'ng[1], Marc D. Perry[34,39], Paz Polak[20,21,22], Esther Rheinbay[17,20,21], Mark A. Rubin[40,41,42,43,44], Colin A. Semple[27], Dennis C. Sgroi[17], Tatsuhiro Shibata[45,46], Reiner Siebert[47,48], Jaclyn Smith[49], Lincoln D. Stein[1,10], Miranda D. Stobbe[50,51], Ren X. Sun[1], Kevin Thai[34], Derek W. Wright[11,52], Chin-Lee Wu[17], Ke Yuan[53,54,55] & Junjun Zhang[34]

[9]Bioinformatics Unit, Spanish National Cancer Research Centre (CNIO), Madrid 28029, Spain. [10]Department of Molecular Genetics, University of Toronto, Toronto, ON M5S 1A8, Canada. [11]Wolfson Wohl Cancer Research Centre, Institute of Cancer Sciences, University of Glasgow, Glasgow, UK. [12]Cancer Division, Garvan Institute of Medical Research, Kinghorn Cancer Centre, University of New South Wales (UNSW Sydney), Sydney, NSW, Australia. [13]South Western Sydney Clinical School, Faculty of Medicine, University of New South Wales (UNSW Sydney), Liverpool, NSW, Australia. [14]West of Scotland Pancreatic Unit, Glasgow Royal Infirmary, Glasgow, UK. [15]Wellcome Sanger Institute, Wellcome Genome Campus, Hinxton, Cambridge CB10 1SA, UK. [16]Department of Haematology, University of Cambridge, Cambridge CB2 2XY, UK. [17]Massachusetts General Hospital, Boston, MA 02114, USA. [18]Weill Cornell Medical College, New York, NY 10065, USA. [19]Department of Medical Oncology, Dana-Farber Cancer Institute, Boston, MA 02215, USA. [20]Broad Institute of MIT and Harvard, Cambridge, MA 02142, USA. [21]Harvard Medical School, Boston, MA 02115, USA. [22]Center for Cancer Research, Massachusetts General Hospital, Boston, MA 02129, USA. [23]Department of Pathology, Massachusetts General Hospital, Boston, MA 02115, USA. [24]Centre for Cancer Research, Victorian Comprehensive Cancer Centre, University of Melbourne, Melbourne, VIC 3052, Australia. [25]Lineberger Comprehensive Cancer Center, University of North Carolina at Chapel Hill, Chapel Hill, NC 27599, USA. [26]Department of Genetics, University of North Carolina at Chapel Hill, Chapel Hill, NC 27599, USA. [27]MRC Human Genetics Unit, MRC IGMM, University of Edinburgh, Edinburgh EH4 2XU, UK. [28]Bioinformatics Group, Division of Molecular Biology, Department of Biology, Faculty of Science, University of Zagreb, Zagreb, Croatia. [29]Department of Bioinformatics, Research Institute, National Cancer Center Japan, Tokyo 104-0045, Japan. [30]Departments of Pathology, The University of Texas MD Anderson Cancer Center, Houston, TX 77030, USA. [31]Department of Genomic Medicine, The University of Texas MD Anderson Cancer Center, Houston, TX 77030, USA. [32]Computational Biology Program, School of Medicine, Oregon Health & Science University, Portland, OR 97239, USA. [33]Hematology, Hospital Clinic, Institut d'Investigacions Biomediques August Pi i Sunyer (IDIBAPS), University of Barcelona, Barcelona, Spain. [34]Genome Informatics Program, Ontario Institute for Cancer Research, Toronto, ON M5G 0A3, Canada. [35]Academic Department of Medical Genetics, University of Cambridge, Addenbrooke's Hospital, Cambridge CB2 0QQ, UK. [36]MRC Cancer Unit, University of Cambridge, Cambridge CB2 0XZ, UK. [37]The University of Cambridge School of Clinical Medicine, Cambridge, UK. [38]Sage Bionetworks, Seattle, WA 98109, USA. [39]Department of Radiation Oncology, University of California San Francisco, San Francisco, CA 94518, USA. [40]Department for Biomedical Research, University of Bern, Bern 3008, Switzerland. [41]Bern Center for Precision Medicine, University Hospital of Bern, University of Bern, Bern 3008, Switzerland. [42]Englander Institute for Precision Medicine, Weill Cornell Medicine and New York Presbyterian Hospital, New York, NY 10021, USA. [43]Meyer Cancer Center, Weill Cornell Medicine, New York, NY 10065, USA. [44]Pathology and Laboratory, Weill Cornell Medical College, New York, NY 10021, USA. [45]Division of Cancer Genomics, National Cancer Center Research Institute, Tokyo 104-0045, Japan. [46]Department of Oncologic Pathology, Dana-Farber Cancer Institute, Harvard Medical School, Boston, MA, USA. [47]Institute of Human Genetics, Ulm University and Ulm University Medical Center, Ulm 89081, Germany. [48]Human Genetics, University of Kiel, Kiel 24118, Germany. [49]Oregon Health and Science University, Portland, OR, USA. [50]CNAG-CRG, Centre for Genomic Regulation (CRG), Barcelona Institute of Science and Technology (BIST), Barcelona 08028, Spain. [51]Universitat Pompeu Fabra (UPF), Barcelona 08003, Spain. [52]MRC-University of Glasgow Centre for Virus Research, Glasgow G61 1QH, UK. [53]Li Ka Shing Centre, Cancer Research UK Cambridge Institute, University of Cambridge, Cambridge, UK. [54]University of Glasgow, Glasgow G61 1BD, UK. [55]School of Computing Science, University of Glasgow, Glasgow G12 8RZ, UK.

## PCAWG Consortium

Lauri A. Aaltonen[56], Federico Abascal[15], Adam Abeshouse[57], Hiroyuki Aburatani[58], David J. Adams[15], Nishant Agrawal[59], Keun Soo Ahn[60], Sung-Min Ahn[61], Hiroshi Aikata[62], Rehan Akbani[63], Kadir C. Akdemir[64], Hikmat Al-Ahmadie[57], Sultan T. Al-Sedairy[65], Fatima Al-Shahrour[9], Malik Alawi[66,67], Monique Albert[68], Kenneth Aldape[30,69], Ludmil B. Alexandrov[15,70,71], Adrian Ally[72], Kathryn Alsop[73], Eva G. Alvarez[74,75,76], Fernanda Amary[77], Samirkumar B. Amin[31,78,79], Brice Aminou[34], Ole Ammerpohl[80,47], Matthew J. Anderson[81], Yeng Ang[82], Davide Antonello[83], Pavana Anur[84], Samuel Aparicio[85], Elizabeth L. Appelbaum[86,87], Yasuhito Arai[45], Axel Aretz[88], Koji Arihiro[62], Shun-ichi Ariizumi[89], Joshua Armenia[90], Laurent Arnould[91], Sylvia Asa[92,93], Yassen Assenov[94], Gurnit Atwal[1,5,10], Sietse Aukema[47,95], J. Todd Auman[96], Miriam R. Aure[97], Philip Awadalla[1,10], Marta Aymerich[98], Gary D. Bader[10], Adrian Baez-Ortega[99], Matthew H. Bailey[86,100],

Peter J. Bailey[11], Miruna Balasundaram[72], Saianand Balu[25], Pratiti Bandopadhayay[20,101,102], Rosamonde E. Banks[103], Stefano Barbi[104], Andrew P. Barbour[105,106], Jonathan Barenboim[1], Jill Barnholtz-Sloan[107,108], Hugh Barr[109], Elisabet Barrera[110], John Bartlett[68,111], Javier Bartolome[112], Claudio Bassi[83], Oliver F. Bathe[113,114], Daniel Baumhoer[115], Prashant Bavi[116], Stephen B. Baylin[117,118], Wojciech Bazant[110], Duncan Beardsmore[119], Timothy A. Beck[120,121], Sam Behjati[15], Andreas Behren[122], Beifang Niu[123], Cindy Bell[124], Sergi Beltran[50,51], Christopher Benz[125], Andrew Berchuck[126], Anke K. Bergmann[127], Erik N. Bergstrom[70,71], Benjamin P. Berman[128,129,130], Daniel M. Berney[131], Stephan H. Bernhart[132,133,134], Rameen Beroukhim[20,19,21], Mario Berrios[135], Samantha Bersani[136], Johanna Bertl[137,138], Miguel Betancourt[139], Vinayak Bhandari[1,2], Shriram G. Bhosle[15], Andrew V. Biankin[11,12,13,14], Matthias Bieg[140,141], Darell Bigner[142], Hans Binder[132,133], Ewan Birney[110], Michael Birrer[143], Nidhan K. Biswas[144], Bodil Bjerkehagen[115,145], Tom Bodenheimer[25], Lori Boice[146], Giada Bonizzato[147], Johann S. De Bono[148], Arnoud Boot[149,150], Moiz S. Bootwalla[135], Ake Borg[151], Arndt Borkhardt[152], Keith A. Boroevich[153,154], Ivan Borozan[1], Christoph Borst[155], Marcus Bosenberg[156], Mattia Bosio[112,51,157], Jacqueline Boultwood[158], Guillaume Bourque[159,160], Paul C. Boutros[1,2,3,4], G. Steven Bova[161], David T. Bowen[15,162], Reanne Bowlby[72], David D. L. Bowtell[73], Sandrine Boyault[163], Rich Boyce[110], Jeffrey Boyd[164], Alvis Brazma[110], Paul Brennan[165], Daniel S. Brewer[166,167], Arie B. Brinkman[168], Robert G. Bristow[2,169,170,171,172], Russell R. Broaddus[30], Jane E. Brock[173], Malcolm Brock[174], Annegien Broeks[175], Angela N. Brooks[20,19,176,177], Denise Brooks[72], Benedikt Brors[178,179,180], Søren Brunak[181,182], Timothy J. C. Bruxner[81,183], Alicia L. Bruzos[74,75,76], Alex Buchanan[184], Ivo Buchhalter[141,185,186], Christiane Buchholz[187], Susan Bullman[20,19], Hazel Burke[188], Birgit Burkhardt[189], Kathleen H. Burns[190,191], John Busanovich[20,192], Carlos D. Bustamante[193,194], Adam P. Butler[15], Atul J. Butte[195], Niall J. Byrne[34], Anne-Lise Børresen-Dale[97,196], Samantha J. Caesar-Johnson[197], Andy Cafferkey[110], Declan Cahill[198], Claudia Calabrese[110,199], Carlos Caldas[200,53], Fabien Calvo[201], Niedzica Camacho[148], Peter J. Campbell[15,16], Elias Campo[202,203], Cinzia Cantù[147], Shaolong Cao[63], Thomas E. Carey[204], Joana Carlevaro-Fita[40,205,206], Rebecca Carlsen[72], Ivana Cataldo[136,147], Mario Cazzola[207], Jonathan Cebon[122], Robert Cerfolio[208], Dianne E. Chadwick[209], Dimple Chakravarty[210], Don Chalmers[211], Calvin Wing Yiu Chan[185,212], Kin Chan[213], Michelle Chan-Seng-Yue[116], Vishal S. Chandan[214], David K. Chang[11,12], Stephen J. Chanock[215], Lorraine A. Chantrill[12,216], Aurélien Chateigner[34,217], Nilanjan Chatterjee[117,218], Kazuaki Chayama[62], Hsiao-Wei Chen[82,90], Jieming Chen[195], Ken Chen[64], Yiwen Chen[63], Zhaohong Chen[219], Andrew D. Cherniack[20,19], Jeremy Chien[220], Yoke-Eng Chiew[221,222], Suet-Feung Chin[200,53], Juok Cho[20], Sunghoon Cho[223], Jung Kyoon Choi[224], Wan Choi[225], Christine Chomienne[226], Zechen Chong[227], Su Pin Choo[228], Angela Chou[12,221], Angelika N. Christ[81], Elizabeth L. Christie[73], Eric Chuah[72], Carrie Cibulskis[20], Kristian Cibulskis[20], Sara Cingarlini[229], Peter Clapham[15], Alexander Claviez[230], Sean Cleary[116,231], Nicole Cloonan[232], Marek Cmero[233,234,235], Colin C. Collins[236], Ashton A. Connor[231,237], Susanna L. Cooke[11], Colin S. Cooper[148,167,238], Leslie Cope[117], Vincenzo Corbo[104,147], Matthew G. Cordes[86,239], Stephen M. Cordner[240], Isidro Cortés-Ciriano[241,242,243], Kyle Covington[244], Prue A. Cowin[245], Brian Craft[177], David Craft[20,246], Chad J. Creighton[247], Yupeng Cun[248], Erin Curley[249], Ioana Cutcutache[149,150], Karolina Czajka[250], Bogdan Czerniak[30,251], Rebecca A. Dagg[252], Ludmila Danilova[117], Maria Vittoria Davi[253], Natalie R. Davidson[18,254,255,256,257], Helen Davies[15,35,36], Ian J. Davis[258], Brandi N. Davis-Dusenbery[259], Kevin J. Dawson[15], Francisco M. De La Vega[193,194,260], Ricardo De Paoli-Iseppi[188], Timothy Defreitas[20], Angelo P. Dei Tos[261], Olivier Delaneau[262,263,264], John A. Demchok[197], Jonas Demeulemeester[265,266], German M. Demidov[51,157,267], Deniz Demircioğlu[268,269], Nening M. Dennis[198], Robert E. Denroche[116], Stefan C. Dentro[15,265,270], Nikita Desai[34], Vikram Deshpande[143], Amit G. Deshwar[271], Christine Desmedt[272,273],

Jordi Deu-Pons[274,275], Noreen Dhalla[72], Neesha C. Dhani[276], Priyanka Dhingra[277,278], Rajiv Dhir[279], Anthony DiBiase[280], Klev Diamanti[281], Li Ding[86,100,282], Shuai Ding[283], Huy Q. Dinh[128], Luc Dirix[284], HarshaVardhan Doddapaneni[244], Nilgun Donmez[236,285], Michelle T. Dow[219], Ronny Drapkin[286], Oliver Drechsel[51,157], Ruben M. Drews[53], Serge Serge[15], Tim Dudderidge[118,198], Ana Dueso-Barroso[112], Andrew J. Dunford[20], Michael Dunn[287], Lewis Jonathan Dursi[1,288], Fraser R. Duthie[11,289], Ken Dutton-Regester[290], Jenna Eagles[250], Douglas F. Easton[291,292], Stuart Edmonds[293], Paul A. Edwards[53,294], Sandra E. Edwards[148], Rosalind A. Eeles[148,198], Anna Ehinger[295], Juergen Eils[296,297], Roland Eils[185,186,296,297], Adel El-Naggar[30,251], Matthew Eldridge[53], Kyle Ellrott[184], Serap Erkek[199], Georgia Escaramis[157,298,299], Shadrielle M. G. Espiritu[1], Xavier Estivill[157,300], Dariush Etemadmoghadam[73], Jorunn E. Eyfjord[301], Bishoy M. Faltas[18], Daiming Fan[302], Yu Fan[63], William C. Faquin[143], Claudiu Farcas[219], Matteo Fassan[303], Aquila Fatima[304], Francesco Favero[305], Nodirjon Fayzullaev[34], Ina Felau[197], Sian Fereday[73], Martin L. Ferguson[306], Vincent Ferretti[34,307], Lars Feuerbach[178], Matthew A. Field[308], J. Lynn Fink[81,112], Gaetano Finocchiaro[309], Cyril Fisher[198], Matthew W. Fittall[265], Anna Fitzgerald[310], Rebecca C. Fitzgerald[36], Adrienne M. Flanagan[311], Neil E. Fleshner[312], Paul Flicek[110], John A. Foekens[313], Kwun M. Fong[314], Nuno A. Fonseca[110,315], Christopher S. Foster[316,317], Natalie S. Fox[1], Michael Fraser[1], Scott Frazer[20], Milana Frenkel-Morgenstern[318], William Friedman[319], Joan Frigola[274], Catrina C. Fronick[86,239], Akihiro Fujimoto[154], Masashi Fujita[154], Masashi Fukayama[320], Lucinda A. Fulton[86], Robert S. Fulton[86,100,282], Mayuko Furuta[154], P. Andrew Futreal[321], Anja Füllgrabe[110], Stacey B. Gabriel[20], Steven Gallinger[116,231,237], Carlo Gambacorti-Passerini[322], Jianjiong Gao[90], Shengjie Gao[323], Levi Garraway[19], Øystein Garred[324], Erik Garrison[15], Dale W. Garsed[73], Nils Gehlenborg[20,325], Josep L. L. Gelpi[112,326], Joshy George[79], Daniela S. Gerhard[327], Clarissa Gerhauser[328], Jeffrey E. Gershenwald[329,330], Mark Gerstein[331,332,333], Moritz Gerstung[110,199], Gad Getz[20,21,22,23], Mohammed Ghori[15], Ronald Ghossein[334], Nasra H. Giama[335], Richard A. Gibbs[244], Anthony J. Gill[12,336], Pelvender Gill[337], Dilip D. Giri[334], Dominik Glodzik[15], Vincent J. Gnanapragasam[338,339], Maria Elisabeth Goebler[340], Mary J. Goldman[177], Carmen Gomez[341], Santiago Gonzalez[110,199], Abel Gonzalez-Perez[274,275,342], Dmitry A. Gordenin[343], James Gossage[344], Kunihito Gotoh[345], Ramaswamy Govindan[100], Dorthe Grabau[346], Janet S. Graham[11,347], Robert C. Grant[116,237], Anthony R. Green[294], Eric Green[348], Liliana Greger[110], Nicola Grehan[36], Sonia Grimaldi[147], Sean M. Grimmond[24], Robert L. Grossman[349], Adam Grundhoff[67,350], Gunes Gundem[57], Qianyun Guo[351], Manaswi Gupta[20], Shailja Gupta[352], Ivo G. Gut[50,51], Marta Gut[50,51], Jonathan Göke[268,353], Gavin Ha[20], Andrea Haake[80], David Haan[176], Siegfried Haas[155], Kerstin Haase[265], James E. Haber[354], Nina Habermann[199], Faraz Hach[236,355], Syed Haider[1], Natsuko Hama[45], Freddie C. Hamdy[337], Anne Hamilton[245], Mark P. Hamilton[356], Leng Han[357], George B. Hanna[358], Martin Hansmann[359], Nicholas J. Haradhvala[20,143], Olivier Harismendy[71,360], Ivon Harliwong[81], Arif O. Harmanci[333,361], Eoghan Harrington[362], Takanori Hasegawa[363], David Haussler[177,364], Steve Hawkins[53], Shinya Hayami[365], Shuto Hayashi[363], D. Neil Hayes[25,366,367], Stephen J. Hayes[368,369], Nicholas K. Hayward[188,290], Steven Hazell[198], Yao He[370], Allison P. Heath[371], Simon C. Heath[50,51], David Hedley[276], Apurva M. Hegde[372], David I. Heiman[20], Michael C. Heinold[185,186], Zachary Heins[57], Lawrence E. Heisler[120], Eva Hellstrom-Lindberg[373], Mohamed Helmy[374], Seong Gu Heo[375], Austin J. Hepperla[25], José María Heredia-Genestar[376], Carl Herrmann[185,186,377], Peter Hersey[188], Julian M. Hess[20,378], Holmfridur Hilmarsdottir[301], Jonathan Hinton[15], Satoshi Hirano[379], Nobuyoshi Hiraoka[380], Katherine A. Hoadley[25,26], Asger Hobolth[137,351], Ermin Hodzic[285], Jessica I. Hoell[152], Steve Hoffmann[132,133,134,380,381], Oliver Hofmann[382], Andrea Holbrook[135], Aliaksei Z. Holik[157], Michael A. Hollingsworth[383], Oliver Holmes[183,290], Robert A. Holt[72], Chen Hong[178,212], Eun Pyo Hong[375], Jongwhi H. Hong[384], Gerrit K. Hooijer[385], Henrik Hornshøj[138], Fumie Hosoda[45], Yong Hou[323,386],

Volker Hovestadt[387], William Howat[338], Alan P. Hoyle[25], Ralph H. Hruban[117], Jianhong Hu[244], Taobo Hu[388], Xing Hua[215], Kuan-lin Huang[86,389], Mei Huang[146], Mi Ni Huang[149,150], Vincent Huang[1], Yi Huang[390,391], Wolfgang Huber[199], Thomas J. Hudson[250,392], Michael Hummel[393], Jillian A. Hung[221,222], David Huntsman[394], Ted R. Hupp[395], Jason Huse[57], Matthew R. Huska[396], Barbara Hutter[141,180,397], Carolyn M. Hutter[348], Daniel Hübschmann[186,296,398,399,400], Christine A. Iacobuzio-Donahue[334], Charles David Imbusch[178], Marcin Imielinski[401,402], Seiya Imoto[363], William B. Isaacs[403], Keren Isaev[1,2], Shumpei Ishikawa[404], Murat Iskar[387], S. M. Ashiqul Islam[219], Michael Ittmann[405,406,407], Sinisa Ivkovic[259], Jose M. G. Izarzugaza[408], Jocelyne Jacquemier[409], Valerie Jakrot[188], Nigel B. Jamieson[11,14,410], Gun Ho Jang[116], Se Jin Jang[411], Joy C. Jayaseelan[244], Reyka Jayasinghe[86], Stuart R. Jefferys[25], Karine Jegalian[412], Jennifer L. Jennings[413], Seung-Hyup Jeon[225], Lara Jerman[199,414], Yuan Ji[415,416], Wei Jiao[1], Peter A. Johansson[290], Amber L. Johns[12], Jeremy Johns[250], Rory Johnson[205,417], Todd A. Johnson[153], Clemency Jolly[265], Yann Joly[418], Jon G. Jonasson[301], Corbin D. Jones[419], David R. Jones[15], David T. W. Jones[420,421], Nic Jones[422], Steven J. M. Jones[72], Jos Jonkers[175], Young Seok Ju[15,224], Hartmut Juhl[423], Jongsun Jung[424], Malene Juul[138], Randi Istrup Juul[138], Sissel Juul[362], Natalie Jäger[185], Rolf Kabbe[185], Andre Kahles[254,255,256,257,425], Abdullah Kahraman[426,427,428], Vera B. Kaiser[27], Hojabr Kakavand[188], Sangeetha Kalimuthu[116], Christof von Kalle[399], Koo Jeong Kang[60], Katalin Karaszi[337], Beth Karlan[429], Rosa Karlić[28], Dennis Karsch[430], Katayoon Kasaian[72], Karin S. Kassahn[81,431], Hitoshi Katai[432], Mamoru Kato[29], Hiroto Katoh[404], Yoshiiku Kawakami[62], Jonathan D. Kay[87], Stephen H. Kazakoff[183,290], Marat D. Kazanov[433,434,435], Maria Keays[110], Electron Kebebew[436,437], Richard F. Kefford[438], Manolis Kellis[20,439], James G. Kench[12,336,440], Catherine J. Kennedy[221,222], Jules N. A. Kerssemakers[185], David Khoo[251], Vincent Khoo[198], Narong Khuntikeo[83,441], Ekta Khurana[277,278,442,42], Helena Kilpinen[87], Hark Kyun Kim[443], Hyung-Lae Kim[444], Hyung-Yong Kim[409], Hyunghwan Kim[225], Jaegil Kim[20], Jihoon Kim[445], Jong K. Kim[446], Youngwook Kim[447,448], Tari A. King[449,450,451], Wolfram Klapper[95], Kortine Kleinheinz[185,186], Leszek J. Klimczak[452], Stian Knappskog[15,453], Michael Kneba[430], Bartha M. Knoppers[418], Youngil Koh[454,455], Jan Komorowski[281,456], Daisuke Komura[404], Mitsuhiro Komura[363], Gu Kong[409], Marcel Kool[420,457], Jan O. Korbel[110,199], Viktoriya Korchina[244], Andrey Korshunov[457], Michael Koscher[457], Roelof Koster[458], Zsofia Kote-Jarai[148], Antonios Koures[219], Milena Kovacevic[259], Barbara Kremeyer[15], Helene Kretzmer[133,134], Markus Kreuz[459], Savitri Krishnamurthy[30,460], Dieter Kube[461], Kiran Kumar[20], Pardeep Kumar[198], Sushant Kumar[332,333], Yogesh Kumar[388], Ritika Kundra[82,90], Kirsten Kübler[20,21,143], Ralf Küppers[462], Jesper Lagergren[373,463], Phillip H. Lai[135], Peter W. Laird[464], Sunil R. Lakhani[465], Christopher M. Lalansingh[1], Emilie Lalonde[1], Fabien C. Lamaze[1], Adam Lambert[337], Eric Lander[20], Pablo Landgraf[466,467], Luca Landoni[83], Anita Langerød[97], Andrés Lanzós[205,206,417], Denis Larsimont[468], Erik Larsson[469], Mark Lathrop[160], Loretta M. S. Lau[470], Chris Lawerenz[297], Rita T. Lawlor[147], Michael S. Lawrence[20,143,153], Alexander J. Lazar[30,31], Xuan Le[471], Darlene Lee[72], Donghoon Lee[333], Eunjung Alice Lee[472], Hee Jin Lee[411], Jake June-Koo Lee[241,243], Jeong-Yeon Lee[473], Juhee Lee[474], Ming Ta Michael Lee[321], Henry Lee-Six[15], Kjong-Van Lehmann[254,255,256,257,425], Hans Lehrach[475], Dido Lenze[393], Conrad R. Leonard[183,290], Daniel A. Leongamornlert[15,148], Ignaty Leshchiner[20], Louis Letourneau[476], Ivica Letunic[477], Douglas A. Levine[57,478], Lora Lewis[244], Tim Ley[479], Chang Li[323,386], Constance H. Li[1,2], Haiyan Irene Li[72], Jun Li[63], Lin Li[323], Shantao Li[333], Siliang Li[323,386], Xiaobo Li[323,386], Xiaotong Li[333], Xinyue Li[323], Yilong Li[15], Han Liang[63], Sheng-Ben Liang[209], Peter Lichter[387,397], Pei Lin[20], Ziao Lin[20,480], W. M. Linehan[481], Ole Christian Lingjærde[482], Dongbing Liu[323,386], Eric Minwei Liu[57,277,278], Fei-Fei Liu[172,483], Fenglin Liu[370,484], Jia Liu[58,109,485], Xingmin Liu[323,386], Julie Livingstone[1], Dimitri Livitz[20], Naomi Livni[198], Lucas Lochovsky[79,332,333], Markus Loeffler[459], Georgina V. Long[188], Armando Lopez-Guillermo[33], Shaoke Lou[332,333], David N. Louis[143],

Laurence B. Lovat[87], Yiling Lu[372], Yong-Jie Lu[131,486], Youyong Lu[487,488,489], Claudio Luchini[136], Ilinca Lungu[111,116], Xuemei Luo[120], Hayley J. Luxton[87], Andy G. Lynch[53,294,490], Lisa Lype[491], Cristina López[80,47], Carlos López-Otín[492], Eric Z. Ma[388], Yussanne Ma[72], Gaetan MacGrogan[493], Shona MacRae[494], Geoff Macintyre[53], Tobias Madsen[138], Kazuhiro Maejima[154], Andrea Mafficini[147], Dennis T. Maglinte[135,495], Arindam Maitra[144], Partha P. Majumder[144], Luca Malcovati[207], Salem Malikic[236,285], Giuseppe Malleo[83], Graham J. Mann[188,221,496], Luisa Mantovani-Löffler[497], Kathleen Marchal[498,499], Giovanni Marchegiani[83], Elaine R. Mardis[86,164,500], Adam A. Margolin[32], Maximillian G. Marin[176], Florian Markowetz[53,294], Julia Markowski[396], Jeffrey Marks[501], Tomas Marques-Bonet[50,376,502,503], Marco A. Marra[72], Luke Marsden[337], John W. M. Martens[313], Sancha Martin[15,54], Jose I. Martin-Subero[503,504], Iñigo Martincorena[15], Alexander Martinez-Fundichely[277,278,42], Yosef E. Maruvka[20,143,378], R. Jay Mashl[86,505], Charlie E. Massie[53], Thomas J. Matthew[176], Lucy Matthews[148], Erik Mayer[198,506], Simon Mayes[507], Michael Mayo[72], Faridah Mbabaali[250], Karen McCune[508], Ultan McDermott[15], Patrick D. McGillivray[332], Michael D. McLellan[86,100,282], John D. McPherson[116,250,509], John R. McPherson[149,150], Treasa A. McPherson[237], Samuel R. Meier[20], Alice Meng[510], Shaowu Meng[25], Andrew Menzies[15], Neil D. Merrett[83,511], Sue Merson[148], Matthew Meyerson[20,19,21], William Meyerson[333,512], Piotr A. Mieczkowski[513], George L. Mihaiescu[34], Sanja Mijalkovic[259], Ana Mijalkovic Mijalkovic-Lazic[259], Tom Mikkelsen[514], Michele Milella[229], Linda Mileshkin[73], Christopher A. Miller[86], David K. Miller[81,12], Jessica K. Miller[250], Gordon B. Mills[515], Ana Milovanovic[112], Sarah Minner[516], Marco Miotto[83], Gisela Mir Arnau[245], Lisa Mirabello[215], Chris Mitchell[73], Thomas J. Mitchell[15,294,338], Satoru Miyano[363], Naoki Miyoshi[363], Shinichi Mizuno[517], Fruzsina Molnár-Gábor[518], Malcolm J. Moore[276], Richard A. Moore[72], Sandro Morganella[15], Quaid D. Morris[5,483], Carl Morrison[519,520], Lisle E. Mose[25], Catherine D. Moser[335], Ferran Muiños[274,275], Loris Mularoni[274,275], Andrew J. Mungall[72], Karen Mungall[72], Elizabeth A. Musgrove[11], Ville Mustonen[521,522,523], David Mutch[524], Francesc Muyas[51,157,267], Donna M. Muzny[244], Alfonso Muñoz[110], Jerome Myers[525], Ola Myklebost[453], Peter Möller[526], Genta Nagae[58], Adnan M. Nagrial[12], Hardeep K. Nahal-Bose[34], Hitoshi Nakagama[527], Hidewaki Nakagawa[154], Hiromi Nakamura[45], Toru Nakamura[379], Kaoru Nakano[154], Tannistha Nandi[528], Jyoti Nangalia[15], Mia Nastic[259], Arcadi Navarro[50,376,502], Fabio C. P. Navarro[332], David E. Neal[53,338], Gerd Nettekoven[529], Felicity Newell[183,290], Steven J. Newhouse[110], Yulia Newton[176], Alvin Wei Tian Ng[530], Anthony Ng[531], Jonathan Nicholson[15], David Nicol[198], Yongzhan Nie[302,532], G. Petur Nielsen[143], Morten Muhlig Nielsen[138], Serena Nik-Zainal[15,35,36,37], Michael S. Noble[20], Katia Nones[183,290], Paul A. Northcott[533], Faiyaz Notta[116,534], Brian D. O'Connor[34,535], Peter O'Donnell[536], Maria O'Donovan[36], Sarah O'Meara[15], Brian Patrick O'Neill[537], J. Robert O'Neill[538], David Ocana[110], Angelica Ochoa[57], Layla Oesper[539], Christopher Ogden[198], Hideki Ohdan[62], Kazuhiro Ohi[363], Lucila Ohno-Machado[219], Karin A. Oien[519,540], Akinyemi I. Ojesina[541,542,543], Hidenori Ojima[544], Takuji Okusaka[545], Larsson Omberg[38], Chovon Kiat Ong[57,546], Stephan Ossowski[51,157,267], German Ott[547], B. F. Francis Ouellette[34,548], Christine P'ng[1], Marta Paczkowska[1], Salvatore Paiella[83], Chawalit Pairojkul[519], Marina Pajic[12], Qiang Pan-Hammarström[323,549], Elli Papaemmanuil[15], Irene Papatheodorou[110], Nagarajan Paramasivam[141,185], Ji Wan Park[375], Joong-Won Park[550], Keunchil Park[551,552], Kiejung Park[553], Peter J. Park[241,243], Joel S. Parker[513], Simon L. Parsons[93], Harvey Pass[554], Danielle Pasternack[250], Alessandro Pastore[254], Ann-Marie Patch[183,290], Iris Pauporté[226], Antonio Pea[83], John V. Pearson[183,290], Chandra Sekhar Pedamallu[20,19,21], Jakob Skou Pedersen[138,351], Paolo Pederzoli[83], Martin Peifer[248], Nathan A. Pennell[555], Charles M. Perou[96,513], Marc D. Perry[34,57], Gloria M. Petersen[556], Myron Peto[84], Nicholas Petrelli[557], Robert Petryszak[110], Stefan M. Pfister[420,457,558], Mark Phillips[418], Oriol Pich[274,275], Hilda A. Pickett[470], Todd D. Pihl[559], Nischalan Pillay[560], Sarah Pinder[561], Mark Pinese[12], Andreia V. Pinho[562],

Esa Pitkänen[199], Xavier Pivot[563], Elena Piñeiro-Yáñez[19], Laura Planko[529], Christoph Plass[328], Paz Polak[20,21,22], Tirso Pons[564], Irinel Popescu[565], Olga Potapova[566], Aparna Prasad[51], Shaun R. Preston[567], Manuel Prinz[185], Antonia L. Pritchard[290], Elena Provenzano[568], Xose S. Puente[492], Sonia Puig[146], Montserrat Puiggròs[112], Sergio Pulido-Tamayo[498,499], Gulietta M. Pupo[221], Colin A. Purdie[569], Michael C. Quinn[183,290], Raquel Rabionet[51,157,570], Janet S. Rader[571], Bernhard Radlwimmer[387], Petar Radovic[259], Benjamin Raeder[199], Keiran M. Raine[15], Manasa Ramakrishna[15], Kamna Ramakrishnan[15], Suresh Ramalingam[572], Benjamin J. Raphael[573], W. Kimryn Rathmell[574], Tobias Rausch[199], Guido Reifenberger[467], Jüri Reimand[1,2], Jorge Reis-Filho[334], Victor Reuter[334], Iker Reyes-Salazar[274], Matthew A. Reyna[573], Sheila M. Reynolds[491], Esther Rheinbay[20,21,143], Yasser Riazalhosseini[160], Andrea L. Richardson[304], Julia Richter[80,95], Matthew Ringel[575], Markus Ringnér[151], Yasushi Rino[576], Karsten Rippe[399], Jeffrey Roach[577], Lewis R. Roberts[335], Nicola D. Roberts[15], Steven A. Roberts[578], A. Gordon Robertson[72], Alan J. Robertson[81], Javier Bartolomé Rodriguez[112], Bernardo Rodriguez-Martin[74,75,76], F. Germán Rodríguez-González[313,579], Michael H. A. Roehrl[2,92,116,209,580,581], Marius Rohde[582], Hirofumi Rokutan[29], Gilles Romieu[583], Ilse Rooman[12], Tom Roques[239], Daniel Rosebrock[20], Mara Rosenberg[20,143], Philip C. Rosenstiel[584], Andreas Rosenwald[585], Edward W. Rowe[198,586], Romina Royo[112], Steven G. Rozen[149,150,587], Yulia Rubanova[5,588], Mark A. Rubin[417,41,589,43,44], Carlota Rubio-Perez[274,275,590], Vasilisa A. Rudneva[199], Borislav C. Rusev[147], Andrea Ruzzenente[591], Gunnar Rätsch[18,254,255,256,257,425], Radhakrishnan Sabarinathan[274,275,592], Veronica Y. Sabelnykova[1], Sara Sadeghi[72], S. Cenk Sahinalp[236,285,593], Natalie Saini[343], Mihoko Saito-Adachi[29], Gordon Saksena[20], Adriana Salcedo[1], Roberto Salgado[594], Leonidas Salichos[332,333], Richard Sallari[20], Charles Saller[595], Roberto Salvia[83], Michelle Sam[250], Jaswinder S. Samra[83,596], Francisco Sanchez-Vega[82,90], Chris Sander[254,597,598], Grant Sanders[25], Rajiv Sarin[599], Iman Sarrafi[236,285], Aya Sasaki-Oku[154], Torill Sauer[482], Guido Sauter[516], Robyn P. M. Saw[188], Maria Scardoni[136], Christopher J. Scarlett[112,600], Aldo Scarpa[147], Ghislaine Scelo[165], Dirk Schadendorf[397,601], Jacqueline E. Schein[72], Markus B. Schilhabel[584], Matthias Schlesner[185,602], Thorsten Schlomm[603,604], Heather K. Schmidt[86], Sarah-Jane Schramm[221], Stefan Schreiber[605], Nikolaus Schultz[90], Steven E. Schumacher[20,304], Roland F. Schwarz[110,396,399,606], Richard A. Scolyer[188,440,596], David Scott[422], Ralph Scully[607], Raja Seethala[608], Ayellet V. Segre[20,609], Iris Selander[237], Colin A. Semple[27], Yasin Senbabaoglu[254], Subhajit Sengupta[610], Elisabetta Sereni[83], Stefano Serra[580], Dennis C. Sgroi[143], Mark Shackleton[73], Nimish C. Shah[338], Sagedeh Shahabi[209], Catherine A. Shang[310], Ping Shang[188], Ofer Shapira[20,20,304], Troy Shelton[249], Ciyue Shen[597,598], Hui Shen[611], Rebecca Shepherd[15], Ruian Shi[483], Yan Shi[25], Yu-Jia Shiah[1], Tatsuhiro Shibata[45,612], Juliann Shih[20,19], Eigo Shimizu[363], Kiyo Shimizu[613], Seung Jun Shin[614], Yuichi Shiraishi[363], Tal Shmaya[260], Ilya Shmulevich[491], Solomon I. Shorser[1], Charles Short[110], Raunak Shrestha[236], Suyash S. Shringarpure[194], Craig Shriver[615], Shimin Shuai[1,10], Nikos Sidiropoulos[579], Reiner Siebert[47,10], Anieta M. Sieuwerts[313], Lina Sieverling[178,212], Sabina Signoretti[173,94], Katarzyna O. Sikora[147], Michele Simbolo[104], Ronald Simon[516], Janae V. Simons[25], Jared T. Simpson[1,588], Peter T. Simpson[465], Samuel Singer[83,450], Nasa Sinnott-Armstrong[20,194], Payal Sipahimalani[72], Tara J. Skelly[26], Marcel Smid[313], Jaclyn Smith[5], Karen Smith-McCune[508], Nicholas D. Socci[254], Heidi J. Sofia[348], Matthew G. Soloway[25], Lei Song[215], Anil K. Sood[616,617,618], Sharmila Sothi[619], Christos Sotiriou[219], Cameron M. Soulette[176], Paul N. Span[620], Paul T. Spellman[84], Nicola Sperandio[147], Andrew J. Spillane[188], Oliver Spiro[20], Jonathan Spring[621], Johan Staaf[151], Peter F. Stadler[132,133,134], Peter Staib[622], Stefan G. Stark[255,257,614,623], Lucy Stebbings[15], Ólafur Andri Stefánsson[624], Oliver Stegle[110,199,625], Lincoln D. Stein[1,10], Alasdair Stenhouse[626], Chip Stewart[20], Stephan Stilgenbauer[627], Miranda D. Stobbe[50,51], Michael R. Stratton[15], Jonathan R. Stretch[188], Adam J. Struck[32], Joshua M. Stuart[176,177], Henk G. Stunnenberg[386,628], Hong Su[323,386], Xiaoping Su[30], Ren X. Sun[1],

Stephanie Sungalee[199], Hana Susak[51,157], Akihiro Suzuki[58,629], Fred Sweep[630], Monika Szczepanowski[95], Holger Sültmann[180,631], Takashi Yugawa[613], Angela Tam[72], David Tamborero[274,275], Benita Kiat Tee Tan[632], Donghui Tan[513], Patrick Tan[150,528,587,633], Hiroko Tanaka[363], Hirokazu Taniguchi[612], Tomas J. Tanskanen[634], Maxime Tarabichi[15,265], Roy Tarnuzzer[197], Patrick Tarpey[635], Morgan L. Taschuk[120], Kenji Tatsuno[58], Simon Tavaré[53,636], Darrin F. Taylor[81], Amaro Taylor-Weiner[20], Jon W. Teague[15], Bin Tean Teh[150,587,633,637,638], Varsha Tembe[221], Javier Temes[74,75], Kevin Thai[34], Sarah P. Thayer[383], Nina Thiessen[72], Gilles Thomas[639], Sarah Thomas[198], Alan Thompson[198], Alastair M. Thompson[626], John F. Thompson[188], R. Houston Thompson[640], Heather Thorne[73], Leigh B. Thorne[146], Adrian Thorogood[418], Grace Tiao[20], Nebojsa Tijanic[259], Lee E. Timms[250], Roberto Tirabosco[641], Marta Tojo[76], Stefania Tommasi[642], Christopher W. Toon[12], Umut H. Toprak[186,643], David Torrents[112,502], Giampaolo Tortora[644,645], Jörg Tost[646], Yasushi Totoki[45], David Townend[647], Nadia Traficante[73], Isabelle Treilleux[648,649], Jean-Rémi Trotta[50], Lorenz H. P. Trümper[461], Ming Tsao[93,534], Tatsuhiko Tsunoda[153,650,651,652], Jose M. C. Tubio[74,75,76], Olga Tucker[653], Richard Turkington[654], Daniel J. Turner[507], Andrew Tutt[304], Masaki Ueno[365], Naoto T. Ueno[655], Christopher Umbricht[119,190,656], Husen M. Umer[281,657], Timothy J. Underwood[658], Lara Urban[110,199], Tomoko Urushidate[612], Tetsuo Ushiku[320], Liis Uusküla-Reimand[659,660], Alfonso Valencia[112,502], David J. Van Den Berg[135], Steven Van Laere[284], Peter Van Loo[265,266], Erwin G. Van Meir[661], Gert G. Van den Eynden[284], Theodorus Van der Kwast[92], Naveen Vasudev[103], Miguel Vazquez[112,662], Ravikiran Vedururu[245], Umadevi Veluvolu[513], Shankar Vembu[483,663], Lieven P. C. Verbeke[499,664], Peter Vermeulen[284], Clare Verrill[337,665], Alain Viari[147], David Vicente[112], Caterina Vicentini[147], K. Vijay Raghavan[352], Juris Viksna[666], Ricardo E. Vilain[667], Izar Villasante[112], Anne Vincent-Salomon[628], Tapio Visakorpi[161], Douglas Voet[20], Paresh Vyas[290,337], Ignacio Vázquez-García[15,668,669,670], Nick M. Waddell[183], Nicola Waddell[183,290], Claes Wadelius[671], Lina Wadi[1], Rabea Wagener[80,47], Jeremiah A. Wala[20,19,21], Jian Wang[323], Jiayin Wang[86,391,672], Linghua Wang[244], Qi Wang[457], Wenyi Wang[63], Yumeng Wang[63], Zhining Wang[197], Paul M. Waring[519], Hans-Jörg Warnatz[475], Jonathan Warrell[332,333], Anne Y. Warren[338,673], Sebastian M. Waszak[199], David C. Wedge[15,270,674], Dieter Weichenhan[328], Paul Weinberger[675], John N. Weinstein[372], Joachim Weischenfeldt[199,579,603], Daniel J. Weisenberger[135], Ian Welch[676], Michael C. Wendl[86,282,677], Johannes Werner[185,678], Justin P. Whalley[50,679], David A. Wheeler[244,680], Hayley C. Whitaker[87], Dennis Wigle[681], Matthew D. Wilkerson[513], Ashley Williams[219], James S. Wilmott[188], Gavin W. Wilson[1,116], Julie M. Wilson[116], Richard K. Wilson[86,682], Boris Winterhoff[683], Jeffrey A. Wintersinger[5,374,588], Maciej Wiznerowicz[684,685], Stephan Wolf[686], Bernice H. Wong[687], Tina Wong[72,86], Winghing Wong[688], Youngchoon Woo[225], Scott Wood[183,290], Bradly G. Wouters[2], Adam J. Wright[1], Derek W. Wright[11,97], Mark H. Wright[194], Chin-Lee Wu[143], Dai-Ying Wu[260], Guanming Wu[689], Jianmin Wu[12], Kui Wu[323,386], Yang Wu[149,150], Zhenggang Wu[388], Liu Xi[244], Tian Xia[690], Qian Xiang[34], Xiao Xiao[391], Rui Xing[489], Heng Xiong[323,386], Qinying Xu[183,290], Yanxun Xu[691], Hong Xue[388], Shinichi Yachida[45,692], Sergei Yakneen[199], Rui Yamaguchi[363], Takafumi N. Yamaguchi[1], Masakazu Yamamoto[89], Shogo Yamamoto[58], Hiroki Yamaue[365], Fan Yang[483], Huanming Yang[323], Jean Y. Yang[693], Liming Yang[197], Lixing Yang[694], Shanlin Yang[283], Tsun-Po Yang[248], Yang Yang[357], Xiaotong Yao[402,695], Marie-Laure Yaspo[475], Lucy Yates[15], Christina Yau[125], Chen Ye[323,386], Kai Ye[672,696], Venkata D. Yellapantula[282,669], Christopher J. Yoon[224], Sung-Soo Yoon[455], Jun Yu[697], Kaixian Yu[698], Willie Yu[699], Yingyan Yu[700], Ke Yuan[53,54,55], Yuan Yuan[63], Denis Yuen[1], Takashi Yugawa[613], Christina K. Yung[34], Olga Zaikova[701], Jorge Zamora[15,74,75,76], Marc Zapatka[387], Jean C. Zenklusen[197], Thorsten Zenz[180], Nikolajs Zeps[702,703], Cheng-Zhong Zhang[20,704], Fan Zhang[370], Hailei Zhang[20], Hongwei Zhang[486], Hongxin Zhang[90], Jiashan Zhang[197], Jing Zhang[333], Junjun Zhang[34], Xiuqing Zhang[323],

Xuanping Zhang[357,391], Yan Zhang[333,705,706], Zemin Zhang[370,707], Zhongming Zhao[708], Liangtao Zheng[370], Xiuqing Zheng[370], Wanding Zhou[611], Yong Zhou[323], Bin Zhu[215], Hongtu Zhu[698,709], Jingchun Zhu[177], Shida Zhu[323,386], Lihua Zou[710], Xueqing Zou[15], Anna deFazio[221,222,711], Nicholas van As[198], Carolien H. M. van Deurzen[712], Marc J. van de Vijver[519], L. van't Veer[713] & Christian von Mering[428,714]

[56]Applied Tumor Genomics Research Program, Research Programs Unit, University of Helsinki, Helsinki, Finland. [57]Memorial Sloan Kettering Cancer Center, New York, NY, USA. [58]Genome Science Division, Research Center for Advanced Science and Technology, University of Tokyo, Tokyo, Japan. [59]Department of Surgery, University of Chicago, Chicago, IL, USA. [60]Department of Surgery, Division of Hepatobiliary and Pancreatic Surgery, School of Medicine, Keimyung University Dongsan Medical Center, Daegu, South Korea. [61]Department of Oncology, Gil Medical Center, Gachon University, Incheon, South Korea. [62]Hiroshima University, Hiroshima, Japan. [63]Department of Bioinformatics and Computational Biology, The University of Texas MD Anderson Cancer Center, Houston, TX, USA. [64]University of Texas MD Anderson Cancer Center, Houston, TX, USA. [65]King Faisal Specialist Hospital and Research Centre, Al Maather, Riyadh, Saudi Arabia. [66]Bioinformatics Core Facility, University Medical Center Hamburg, Hamburg, Germany. [67]Heinrich Pette Institute, Leibniz Institute for Experimental Virology, Hamburg, Germany. [68]Ontario Tumour Bank, Ontario Institute for Cancer Research, Toronto, ON, Canada. [69]Laboratory of Pathology, Center for Cancer Research, National Cancer Institute, Bethesda, MD, USA. [70]Department of Cellular and Molecular Medicine and Department of Bioengineering, University of California San Diego, La Jolla, CA, USA. [71]UC San Diego Moores Cancer Center, San Diego, CA, USA. [72]Canada's Michael Smith Genome Sciences Centre, BC Cancer, Vancouver, BC, Canada. [73]Sir Peter MacCallum Department of Oncology, Peter MacCallum Cancer Centre, University of Melbourne, Melbourne, VIC, Australia. [74]Centre for Research in Molecular Medicine and Chronic Diseases (CiMUS), Universidade de Santiago de Compostela, Santiago de Compostela, Spain. [75]Department of Zoology, Genetics and Physical Anthropology, (CiMUS), Universidade de Santiago de Compostela, Santiago de Compostela, Spain. [76]The Biomedical Research Centre (CINBIO), Universidade de Vigo, Vigo, Spain. [77]Royal National Orthopaedic Hospital - Bolsover, London, UK. [78]Quantitative and Computational Biosciences Graduate Program, Baylor College of Medicine, Houston, TX, USA. [79]The Jackson Laboratory for Genomic Medicine, Farmington, CT, USA. [80]Institute of Human Genetics, Christian-Albrechts-University, Kiel, Germany. [81]Queensland Centre for Medical Genomics, Institute for Molecular Bioscience, University of Queensland, St. Lucia, Brisbane, QLD, Australia. [82]Salford Royal NHS Foundation Trust, Salford, UK. [83]Department of Surgery, Pancreas Institute, University and Hospital Trust of Verona, Verona, Italy. [84]Molecular and Medical Genetics, OHSU Knight Cancer Institute, Oregon Health and Science University, Portland, OR, USA. [85]Department of Molecular Oncology, BC Cancer Research Centre, Vancouver, BC, Canada. [86]The McDonnell Genome Institute at Washington University, St. Louis, MO, USA. [87]University College London, London, UK. [88]DLR Project Management Agency, Bonn, Germany. [89]Tokyo Women's Medical University, Tokyo, Japan. [90]Center for Molecular Oncology, Memorial Sloan Kettering Cancer Center, New York, NY, USA. [91]Los Alamos National Laboratory, Los Alamos, NM, USA. [92]Department of Pathology, University Health Network, Toronto General Hospital, Toronto, ON, Canada. [93]Nottingham University Hospitals NHS Trust, Nottingham, UK. [94]Epigenomics and Cancer Risk Factors, German Cancer Research Center (DKFZ), Heidelberg, Germany. [95]Hematopathology Section, Institute of Pathology, Christian-Albrechts-University, Kiel, Germany. [96]Department of Pathology and Laboratory Medicine, School of Medicine, University of North Carolina at Chapel Hill, Chapel Hill, NC, USA. [97]Department of Cancer Genetics, Institute for Cancer Research, Oslo University Hospital, The Norwegian Radium Hospital, Oslo, Norway. [98]Pathology, Hospital Clinic, Institut d'Investigacions Biomediques August Pi i Sunyer (IDIBAPS), University of Barcelona, Barcelona, Spain. [99]Department of Veterinary Medicine, Transmissible Cancer Group, University of Cambridge, Cambridge, UK. [100]Alvin J. Siteman Cancer Center, Washington University School of Medicine, St. Louis, MO, USA. [101]Dana-Farber/Boston Children's Cancer and Blood Disorders Center, Boston, MA, USA. [102]Department of Pediatrics, Harvard Medical School, Boston, MA, USA. [103]Leeds Institute of Medical Research @ St. James's University of Leeds, St. James's University Hospital, Leeds, UK. [104]Department of Pathology and Diagnostics, University and Hospital Trust of Verona, Verona, Italy. [105]Department of Surgery, Princess Alexandra Hospital, Brisbane, QLD, Australia. [106]Surgical Oncology Group, Diamantina Institute, University of Queensland, Brisbane, QLD, Australia. [107]Department of Population and Quantitative Health Sciences, Case Western Reserve University School of Medicine, Cleveland, OH, USA. [108]Research Health Analytics and Informatics, University Hospitals Cleveland Medical Center, Cleveland, OH, USA. [109]Gloucester Royal Hospital, Gloucester, UK. [110]European Molecular Biology Laboratory, European Bioinformatics Institute (EMBL-EBI), Cambridge, UK. [111]Diagnostic Development, Ontario Institute for Cancer Research, Toronto, ON, Canada. [112]Barcelona Supercomputing Center (BSC), Barcelona, Spain. [113]Arnie Charbonneau Cancer Institute, University of Calgary, Calgary, AB, Canada. [114]Departments of Surgery and Oncology, University of Calgary, Calgary, AB, Canada. [115]Department of Pathology, Oslo University Hospital, The Norwegian Radium Hospital, Oslo, Norway. [116]PanCuRx Translational Research Initiative, Ontario Institute for Cancer Research, Toronto, ON, Canada. [117]Department of Oncology, Sidney Kimmel Comprehensive Cancer Center at Johns Hopkins University School of Medicine, Baltimore, MD, USA. [118]University Hospital Southampton NHS Foundation Trust, Southampton, UK. [119]Royal Stoke University Hospital, Stoke-on-Trent, UK. [120]Genome Sequence Informatics, Ontario Institute for Cancer Research, Toronto, ON, Canada. [121]Human Longevity Inc, San Diego, CA, USA. [122]Olivia Newton-John Cancer Research Institute, La Trobe University, Heidelberg, VIC, Australia. [123]Computer Network Information Center, Chinese Academy of Sciences, Beijing, China. [124]Genome Canada, Ottawa, ON, Canada. [125]Buck Institute for Research on Aging, Novato, CA, USA. [126]Duke University Medical Center, Durham, NC, USA. [127]Department of Human Genetics, Hannover Medical School, Hannover, Germany. [128]Center for Bioinformatics and Functional Genomics, Cedars-Sinai Medical Center, Los Angeles, CA, USA. [129]Department of Biomedical Sciences, Cedars-Sinai Medical Center, Los Angeles, CA, USA. [130]The Hebrew University Faculty of Medicine, Jerusalem, Israel. [131]Barts Cancer Institute, Barts and the London School of Medicine and Dentistry, Queen Mary University of London, London, UK. [132]Department of Computer Science, Bioinformatics Group, University of Leipzig, Leipzig, Germany. [133]Interdisciplinary Center for Bioinformatics, University of Leipzig, Leipzig, Germany. [134]Transcriptome Bioinformatics, LIFE Research Center for Civilization Diseases, University of Leipzig, Leipzig, Germany. [135]USC Norris Comprehensive Cancer Center, University of Southern California, Los Angeles, CA, USA. [136]Department of Diagnostics and Public Health, University and Hospital Trust of Verona, Verona, Italy. [137]Department of Mathematics, Aarhus University, Aarhus, Denmark. [138]Department of Molecular Medicine (MOMA), Aarhus University Hospital, Aarhus N, Denmark. [139]Instituto Carlos Slim de la Salud, Mexico City, Mexico. [140]Center for Digital Health, Berlin Institute of Health and Charitè - Universitätsmedizin Berlin, Berlin, Germany. [141]Heidelberg Center for Personalized Oncology (DKFZ-HIPO), German Cancer Research Center (DKFZ), Heidelberg, Germany. [142]The Preston Robert Tisch Brain Tumor Center, Duke University Medical Center, Durham, NC, USA. [143]Massachusetts General Hospital, Boston, MA, USA. [144]National Institute of Biomedical Genomics, Kalyani, West Bengal, India. [145]Institute of Clinical Medicine and Institute of Oral Biology, University of Oslo, Oslo, Norway. [146]University of North Carolina at Chapel Hill, Chapel Hill, NC, USA. [147]ARC-Net Centre for Applied Research on Cancer, University and Hospital Trust of Verona, Verona, Italy. [148]The Institute of Cancer Research, London, UK. [149]Centre for Computational Biology, Duke-NUS Medical School,

Singapore, Singapore. [150]Programme in Cancer and Stem Cell Biology, Duke-NUS Medical School, Singapore, Singapore. [151]Division of Oncology and Pathology, Department of Clinical Sciences Lund, Lund University, Lund, Sweden. [152]Department of Pediatric Oncology, Hematology and Clinical Immunology, Heinrich-Heine-University, Düsseldorf, Germany. [153]Laboratory for Medical Science Mathematics, RIKEN Center for Integrative Medical Sciences, Yokohama, Japan. [154]RIKEN Center for Integrative Medical Sciences, Yokohama, Japan. [155]Department of Internal Medicine/Hematology, Friedrich-Ebert-Hospital, Neumünster, Germany. [156]Departments of Dermatology and Pathology, Yale University, New Haven, CT, USA. [157]Centre for Genomic Regulation (CRG), The Barcelona Institute of Science and Technology, Barcelona, Spain. [158]Radcliffe Department of Medicine, University of Oxford, Oxford, UK. [159]Canadian Center for Computational Genomics, McGill University, Montreal, QC, Canada. [160]Department of Human Genetics, McGill University, Montreal, QC, Canada. [161]Faculty of Medicine and Health Technology, Tampere University and Tays Cancer Center, Tampere University Hospital, Tampere, Finland. [162]Haematology, Leeds Teaching Hospitals NHS Trust, Leeds, UK. [163]Translational Research and Innovation, Centre Léon Bérard, Lyon, France. [164]Fox Chase Cancer Center, Philadelphia, PA, USA. [165]International Agency for Research on Cancer, World Health Organization, Lyon, France. [166]Earlham Institute, Norwich, UK. [167]Norwich Medical School, University of East Anglia, Norwich, UK. [168]Department of Molecular Biology, Faculty of Science, Radboud Institute for Molecular Life Sciences, Radboud University, Nijmegen, HB, The Netherlands. [169]CRUK Manchester Institute and Centre, Manchester, UK. [170]Department of Radiation Oncology, University of Toronto, Toronto, ON, Canada. [171]Division of Cancer Sciences, Manchester Cancer Research Centre, University of Manchester, Manchester, UK. [172]Radiation Medicine Program, Princess Margaret Cancer Centre, Toronto, ON, Canada. [173]Department of Pathology, Brigham and Women's Hospital, Harvard Medical School, Boston, MA, USA. [174]Department of Surgery, Division of Thoracic Surgery, The Johns Hopkins University School of Medicine, Baltimore, MD, USA. [175]Division of Molecular Pathology, The Netherlands Cancer Institute, Oncode Institute, Amsterdam, CX, The Netherlands. [176]Department of Biomolecular Engineering, University of California Santa Cruz, Santa Cruz, CA, USA. [177]UC Santa Cruz Genomics Institute, University of California Santa Cruz, Santa Cruz, CA, USA. [178]Division of Applied Bioinformatics, German Cancer Research Center (DKFZ), Heidelberg, Germany. [179]German Cancer Genome Consortium (DKTK), Heidelberg, Germany. [180]National Center for Tumor Diseases (NCT) Heidelberg, Heidelberg, Germany. [181]Center for Biological Sequence Analysis, Department of Bio and Health Informatics, Technical University of Denmark, Lyngby, Denmark. [182]Novo Nordisk Foundation Center for Protein Research, University of Copenhagen, Copenhagen, Denmark. [183]Institute for Molecular Bioscience, University of Queensland, St. Lucia, Brisbane, QLD, Australia. [184]Biomedical Engineering, Oregon Health and Science University, Portland, OR, USA. [185]Division of Theoretical Bioinformatics, German Cancer Research Center (DKFZ), Heidelberg, Germany. [186]Institute of Pharmacy and Molecular Biotechnology and BioQuant, Heidelberg University, Heidelberg, Germany. [187]Federal Ministry of Education and Research, Berlin, Germany. [188]Melanoma Institute Australia, University of Sydney, Sydney, NSW, Australia. [189]Pediatric Hematology and Oncology, University Hospital Muenster, Muenster, Germany. [190]Department of Pathology, Johns Hopkins University School of Medicine, Baltimore, MD, USA. [191]McKusick-Nathans Institute of Genetic Medicine, Sidney Kimmel Comprehensive Cancer Center at Johns Hopkins University School of Medicine, Baltimore, MD, USA. [192]Foundation Medicine, Inc, Cambridge, MA, USA. [193]Department of Biomedical Data Science, Stanford University School of Medicine, Stanford, CA, USA. [194]Department of Genetics, Stanford University School of Medicine, Stanford, CA, USA. [195]Bakar Computational Health Sciences Institute and Department of Pediatrics, University of California, San Francisco, CA, USA. [196]Institute of Clinical Medicine, Faculty of Medicine, University of Oslo, Oslo, Norway. [197]National Cancer Institute, National Institutes of Health, Bethesda, MD, USA. [198]Royal Marsden NHS Foundation Trust, Sutton, London, UK. [199]Genome Biology Unit, European Molecular Biology Laboratory (EMBL), Heidelberg, Germany. [200]Department of Oncology, University of Cambridge, Cambridge, UK. [201]Institut Gustave Roussy, Villejuif, France. [202]Anatomia Patológica, Hospital Clinic, Institut d'Investigacions Biomèdiques August Pi i Sunyer (IDIBAPS), University of Barcelona, Barcelona, Spain. [203]Spanish Ministry of Science and Innovation, Madrid, Spain. [204]University of Michigan Comprehensive Cancer Center, Ann Arbor, MI, USA. [205]Department of Medical Oncology, Inselspital, University Hospital and University of Bern, Bern, Switzerland. [206]Graduate School for Cellular and Biomedical Sciences, University of Bern, Bern, Switzerland. [207]University of Pavia, Pavia, Italy. [208]University of Alabama at Birmingham, Birmingham, AL, USA. [209]UHN Program in BioSpecimen Sciences, Toronto General Hospital, Toronto, ON, Canada. [210]Department of Urology, Icahn School of Medicine at Mount Sinai, New York, NY, USA. [211]Centre for Law and Genetics, University of Tasmania, Sandy Bay Campus, Hobart, TAS, Australia. [212]Faculty of Biosciences, Heidelberg University, Heidelberg, Germany. [213]Department of Biochemistry, Microbiology and Immunology, Faculty of Medicine, University of Ottawa, Ottawa, ON, Canada. [214]Division of Anatomic Pathology, Mayo Clinic, Rochester, MN, USA. [215]Division of Cancer Epidemiology and Genetics, National Cancer Institute, National Institutes of Health, Bethesda, MD, USA. [216]Illawarra Shoalhaven Local Health District L3 Illawarra Cancer Care Centre, Wollongong Hospital, Wollongong, NSW, Australia. [217]BioForA, French National Institute for Agriculture, Food, and Environment (INRAE), ONF, Orléans, France. [218]Department of Biostatistics, Bloomberg School of Public Health, Johns Hopkins University, Baltimore, MD, USA. [219]University of California San Diego, San Diego, CA, USA. [220]Division of Experimental Pathology, Mayo Clinic, Rochester, MN, USA. [221]Centre for Cancer Research, The Westmead Institute for Medical Research, University of Sydney, Sydney, NSW, Australia. [222]Department of Gynaecological Oncology, Westmead Hospital, Sydney, NSW, Australia. [223]PDXen Biosystems Inc, Seoul, South Korea. [224]Korea Advanced Institute of Science and Technology, Daejeon, South Korea. [225]Electronics and Telecommunications Research Institute, Daejeon, South Korea. [226]Institut National du Cancer (INCA), Boulogne-Billancourt, France. [227]Department of Genetics, Informatics Institute, University of Alabama at Birmingham, Birmingham, AL, USA. [228]Division of Medical Oncology, National Cancer Centre, Singapore, Singapore. [229]Medical Oncology, University and Hospital Trust of Verona, Verona, Italy. [230]Department of Pediatrics, University Hospital Schleswig-Holstein, Kiel, Germany. [231]Hepatobiliary/Pancreatic Surgical Oncology Program, University Health Network, Toronto, ON, Canada. [232]School of Biological Sciences, University of Auckland, Auckland, New Zealand. [233]Department of Surgery, University of Melbourne, Parkville, VIC, Australia. [234]The Murdoch Children's Research Institute, Royal Children's Hospital, Parkville, VIC, Australia. [235]Walter and Eliza Hall Institute, Parkville, VIC, Australia. [236]Vancouver Prostate Centre, Vancouver, BC, Canada. [237]Lunenfeld-Tanenbaum Research Institute, Mount Sinai Hospital, Toronto, ON, Canada. [238]University of East Anglia, Norwich, UK. [239]Norfolk and Norwich University Hospital NHS Trust, Norwich, UK. [240]Victorian Institute of Forensic Medicine, Southbank, VIC, Australia. [241]Department of Biomedical Informatics, Harvard Medical School, Boston, MA, USA. [242]Department of Chemistry, Centre for Molecular Science Informatics, University of Cambridge, Cambridge, UK. [243]Ludwig Center at Harvard Medical School, Boston, MA, USA. [244]Human Genome Sequencing Center, Baylor College of Medicine, Houston, TX, USA. [245]Peter MacCallum Cancer Centre, University of Melbourne, Melbourne, VIC, Australia. [246]Physics Division, Optimization and Systems Biology Lab, Massachusetts General Hospital, Boston, MA, USA. [247]Department of Medicine, Baylor College of Medicine, Houston, TX, USA. [248]University of Cologne, Cologne, Germany. [249]International Genomics Consortium, Phoenix, AZ, USA. [250]Genomics Research Program, Ontario Institute for Cancer Research, Toronto, ON, Canada. [251]Barking Havering and Redbridge University Hospitals NHS Trust, Romford, UK. [252]Children's Hospital at Westmead, University of Sydney, Sydney, NSW, Australia. [253]Section of Endocrinology, Department of Medicine, University and Hospital Trust of Verona, Verona, Italy. [254]Computational Biology Center, Memorial Sloan Kettering Cancer Center, New York, NY, USA. [255]Department of Biology, ETH Zurich, Zürich, Switzerland. [256]Department of Computer Science, ETH Zurich, Zurich, Switzerland. [257]SIB Swiss Institute of Bioinformatics, Lausanne, Switzerland. [258]Departments of Pediatrics and Genetics, University of North Carolina at Chapel Hill, Chapel Hill, NC, USA. [259]Seven Bridges Genomics, Charlestown, MA, USA. [260]Annai

Systems, Inc, Carlsbad, CA, USA. [261]Department of Pathology, General Hospital of Treviso, Department of Medicine, University of Padua, Treviso, Italy. [262]Department of Computational Biology, University of Lausanne, Lausanne, Switzerland. [263]Department of Genetic Medicine and Development, University of Geneva Medical School, Geneva, CH, Switzerland. [264]Swiss Institute of Bioinformatics, University of Geneva, Geneva, CH, Switzerland. [265]The Francis Crick Institute, London, UK. [266]University of Leuven, Leuven, Belgium. [267]Institute of Medical Genetics and Applied Genomics, University of Tübingen, Tübingen, Germany. [268]Computational and Systems Biology, Genome Institute of Singapore, Singapore, Singapore. [269]School of Computing, National University of Singapore, Singapore, Singapore. [270]Big Data Institute, Li Ka Shing Centre, University of Oxford, Oxford, UK. [271]The Edward S. Rogers Sr. Department of Electrical and Computer Engineering, University of Toronto, Toronto, ON, Canada. [272]Breast Cancer Translational Research Laboratory JC Heuson, Institut Jules Bordet, Brussels, Belgium. [273]Department of Oncology, Laboratory for Translational Breast Cancer Research, KU Leuven, Leuven, Belgium. [274]Institute for Research in Biomedicine (IRB Barcelona), The Barcelona Institute of Science and Technology, Barcelona, Spain. [275]Research Program on Biomedical Informatics, Universitat Pompeu Fabra, Barcelona, Spain. [276]Division of Medical Oncology, Princess Margaret Cancer Centre, Toronto, ON, Canada. [277]Department of Physiology and Biophysics, Weill Cornell Medicine, New York, NY, USA. [278]Institute for Computational Biomedicine, Weill Cornell Medicine, New York, NY, USA. [279]Department of Pathology, UPMC Shadyside, Pittsburgh, PA, USA. [280]Independent Consultant, Wellesley, USA. [281]Department of Cell and Molecular Biology, Science for Life Laboratory, Uppsala University, Uppsala, Sweden. [282]Department of Medicine and Department of Genetics, Washington University School of Medicine, St. Louis, St. Louis, MO, USA. [283]Hefei University of Technology, Anhui, China. [284]Translational Cancer Research Unit, GZA Hospitals St.-Augustinus, Center for Oncological Research, Faculty of Medicine and Health Sciences, University of Antwerp, Antwerp, Belgium. [285]Simon Fraser University, Burnaby, BC, Canada. [286]University of Pennsylvania, Philadelphia, PA, USA. [287]The Wellcome Trust, London, UK. [288]The Hospital for Sick Children, Toronto, ON, Canada. [289]Department of Pathology, Queen Elizabeth University Hospital, Glasgow, UK. [290]Department of Genetics and Computational Biology, QIMR Berghofer Medical Research Institute, Brisbane, QLD, Australia. [291]Department of Oncology, Centre for Cancer Genetic Epidemiology, University of Cambridge, Cambridge, UK. [292]Department of Public Health and Primary Care, Centre for Cancer Genetic Epidemiology, University of Cambridge, Cambridge, UK. [293]Prostate Cancer Canada, Toronto, ON, Canada. [294]University of Cambridge, Cambridge, UK. [295]Department of Laboratory Medicine, Translational Cancer Research, Lund University Cancer Center at Medicon Village, Lund University, Lund, Sweden. [296]Heidelberg University, Heidelberg, Germany. [297]New BIH Digital Health Center, Berlin Institute of Health (BIH) and Charité - Universitätsmedizin Berlin, Berlin, Germany. [298]CIBER Epidemiología y Salud Pública (CIBERESP), Madrid, Spain. [299]Research Group on Statistics, Econometrics and Health (GRECS), UdG, Barcelona, Spain. [300]Quantitative Genomics Laboratories (qGenomics), Barcelona, Spain. [301]Icelandic Cancer Registry, Icelandic Cancer Society, Reykjavik, Iceland. [302]State Key Laboratory of Cancer Biology, and Xijing Hospital of Digestive Diseases, Fourth Military Medical University, Shaanxi, China. [303]Department of Medicine (DIMED), Surgical Pathology Unit, University of Padua, Padua, Italy. [304]Department of Cancer Biology, Dana-Farber Cancer Institute, Boston, MA, USA. [305]Rigshospitalet, Copenhagen, Denmark. [306]Center for Cancer Genomics, National Cancer Institute, National Institutes of Health, Bethesda, MD, USA. [307]Department of Biochemistry and Molecular Medicine, University of Montreal, Montreal, QC, Canada. [308]Australian Institute of Tropical Health and Medicine, James Cook University, Douglas, QLD, Australia. [309]Department of Neuro-Oncology, Istituto Neurologico Besta, Milano, Italy. [310]Bioplatforms Australia, North Ryde, NSW, Australia. [311]Department of Pathology (Research), University College London Cancer Institute, London, UK. [312]Department of Surgical Oncology, Princess Margaret Cancer Centre, Toronto, ON, Canada. [313]Department of Medical Oncology, Josephine Nefkens Institute and Cancer Genomics Centre, Erasmus Medical Center, Rotterdam, CN, The Netherlands. [314]The University of Queensland Thoracic Research Centre, The Prince Charles Hospital, Brisbane, QLD, Australia. [315]CIBIO/InBIO - Research Center in Biodiversity and Genetic Resources, Universidade do Porto, Vairão, Portugal. [316]HCA Laboratories, London, UK. [317]University of Liverpool, Liverpool, UK. [318]The Azrieli Faculty of Medicine, Bar-Ilan University, Safed, Israel. [319]Department of Neurosurgery, University of Florida, Gainesville, FL, USA. [320]Department of Pathology, Graduate School of Medicine, University of Tokyo, Tokyo, Japan. [321]National Genotyping Center, Institute of Biomedical Sciences, Academia Sinica, Taipei, Taiwan. [322]University of Milano Bicocca, Monza, Italy. [323]BGI-Shenzhen, Shenzhen, China. [324]Department of Pathology, Oslo University Hospital Ulleval, Oslo, Norway. [325]Center for Biomedical Informatics, Harvard Medical School, Boston, MA, USA. [326]Department Biochemistry and Molecular Biomedicine, University of Barcelona, Barcelona, Spain. [327]Office of Cancer Genomics, National Cancer Institute, National Institutes of Health, Bethesda, MD, USA. [328]Cancer Epigenomics, German Cancer Research Center (DKFZ), Heidelberg, Germany. [329]Department of Cancer Biology, The University of Texas MD Anderson Cancer Center, Houston, TX, USA. [330]Department of Surgical Oncology, The University of Texas MD Anderson Cancer Center, Houston, TX, USA. [331]Department of Computer Science, Yale University, New Haven, CT, USA. [332]Department of Molecular Biophysics and Biochemistry, Yale University, New Haven, CT, USA. [333]Program in Computational Biology and Bioinformatics, Yale University, New Haven, CT, USA. [334]Department of Pathology, Memorial Sloan Kettering Cancer Center, New York, NY, USA. [335]Division of Gastroenterology and Hepatology, Mayo Clinic, Rochester, MN, USA. [336]University of Sydney, Sydney, NSW, Australia. [337]University of Oxford, Oxford, UK. [338]Cambridge University Hospitals NHS Foundation Trust, Cambridge, UK. [339]Department of Surgery, Academic Urology Group, University of Cambridge, Cambridge, UK. [340]Department of Medicine II, University of Würzburg, Wuerzburg, Germany. [341]Sylvester Comprehensive Cancer Center, University of Miami, Miami, FL, USA. [342]Institut Hospital del Mar d'Investigacions Mèdiques (IMIM), Barcelona, Spain. [343]Genome Integrity and Structural Biology Laboratory, National Institute of Environmental Health Sciences (NIEHS), Durham, NC, USA. [344]St. Thomas's Hospital, London, UK. [345]Osaka International Cancer Center, Osaka, Japan. [346]Department of Pathology, Skåne University Hospital, Lund University, Lund, Sweden. [347]Department of Medical Oncology, Beatson West of Scotland Cancer Centre, Glasgow, UK. [348]National Human Genome Research Institute, National Institutes of Health, Bethesda, MD, USA. [349]Department of Medicine, Section of Hematology/Oncology, University of Chicago, Chicago, IL, USA. [350]German Center for Infection Research (DZIF), Partner Site Hamburg-Borstel-Lübeck-Riems, Hamburg, Germany. [351]Bioinformatics Research Centre (BiRC), Aarhus University, Aarhus, Denmark. [352]Department of Biotechnology, Ministry of Science and Technology, Government of India, New DelhiDelhi, India. [353]National Cancer Centre Singapore, Singapore, Singapore. [354]Brandeis University, Waltham, MA, USA. [355]Department of Urologic Sciences, University of British Columbia, Vancouver, BC, Canada. [356]Department of Internal Medicine, Stanford University, Stanford, CA, USA. [357]The University of Texas Health Science Center at Houston, Houston, TX, USA. [358]Imperial College NHS Trust, Imperial College, London, INY, UK. [359]Senckenberg Institute of Pathology, University of Frankfurt Medical School, Frankfurt, Germany. [360]Division of Biomedical Informatics, Department of Medicine, UC San Diego School of Medicine, San Diego, CA, USA. [361]Center for Precision Health, School of Biomedical Informatics, The University of Texas Health Science Center, Houston, TX, USA. [362]Oxford Nanopore Technologies, New York, NY, USA. [363]Institute of Medical Science, University of Tokyo, Tokyo, Japan. [364]Howard Hughes Medical Institute, University of California Santa Cruz, Santa Cruz, CA, USA. [365]Wakayama Medical University, Wakayama, Japan. [366]Division of Medical Oncology, Department of Internal Medicine, Lineberger Comprehensive Cancer Center, University of North Carolina at Chapel Hill, Chapel Hill, NC, USA. [367]University of Tennessee Health Science Center for Cancer Research, Memphis, TN, USA. [368]Department of Histopathology, Salford Royal NHS Foundation Trust, Salford, UK. [369]Faculty of Biology, Medicine and Health, University of Manchester, Manchester, UK. [370]Peking University, Beijing, China. [371]Children's Hospital of Philadelphia, Philadelphia, PA, USA. [372]Department of Bioinformatics and Computational Biology and Department of Systems Biology, The University of Texas MD Anderson Cancer Center, Houston,

TX, USA. [373]Karolinska Institute, Stockholm, Sweden. [374]The Donnelly Centre, University of Toronto, Toronto, ON, Canada. [375]Department of Medical Genetics, College of Medicine, Hallym University, Chuncheon, South Korea. [376]Department of Experimental and Health Sciences, Institute of Evolutionary Biology (UPF-CSIC), Universitat Pompeu Fabra, Barcelona, Spain. [377]Health Data Science Unit, University Clinics, Heidelberg, Germany. [378]Massachusetts General Hospital Center for Cancer Research, Charlestown, MA, USA. [379]Hokkaido University, Sapporo, Japan. [380]Department of Pathology and Clinical Laboratory, National Cancer Center Hospital, Tokyo, Japan. [381]Computational Biology, Leibniz Institute on Aging - Fritz Lipmann Institute (FLI), Jena, Germany. [382]University of Melbourne Centre for Cancer Research, Melbourne, VIC, Australia. [383]University of Nebraska Medical Center, Omaha, NE, USA. [384]Syntekabio Inc, Daejeon, South Korea. [385]Department of Pathology, Academic Medical Center, Amsterdam, AZ, The Netherlands. [386]China National GeneBank-Shenzhen, Shenzhen, China. [387]Division of Molecular Genetics, German Cancer Research Center (DKFZ), Heidelberg, Germany. [388]Division of Life Science and Applied Genomics Center, Hong Kong University of Science and Technology, Clear Water Bay, Hong Kong, China. [389]Icahn School of Medicine at Mount Sinai, New York, NY, USA. [390]Geneplus-Shenzhen, Shenzhen, China. [391]School of Computer Science and Technology, Xi'an Jiaotong University, Xi'an, China. [392]AbbVie, North Chicago, IL, USA. [393]Institute of Pathology, Charité – University Medicine Berlin, Berlin, Germany. [394]Centre for Translational and Applied Genomics, British Columbia Cancer Agency, Vancouver, BC, Canada. [395]Edinburgh Royal Infirmary, Edinburgh, UK. [396]Berlin Institute for Medical Systems Biology, Max Delbrück Center for Molecular Medicine, Berlin, Germany. [397]German Cancer Consortium (DKTK), Heidelberg, Germany. [398]Department of Pediatric Immunology, Hematology and Oncology, University Hospital, Heidelberg, Germany. [399]German Cancer Research Center (DKFZ), Heidelberg, Germany. [400]Heidelberg Institute for Stem Cell Technology and Experimental Medicine (HI-STEM), Heidelberg, Germany. [401]Institute for Computational Biomedicine, Weill Cornell Medical College, New York, NY, USA. [402]New York Genome Center, New York, NY, USA. [403]Department of Urology, James Buchanan Brady Urological Institute, Johns Hopkins University School of Medicine, Baltimore, MD, USA. [404]Department of Preventive Medicine, Graduate School of Medicine, The University of Tokyo, Tokyo, Japan. [405]Department of Molecular and Cellular Biology, Baylor College of Medicine, Houston, TX, USA. [406]Department of Pathology and Immunology, Baylor College of Medicine, Houston, TX, USA. [407]Michael E. DeBakey Veterans Affairs Medical Center, Houston, TX, USA. [408]Technical University of Denmark, Lyngby, Denmark. [409]Department of Pathology, College of Medicine, Hanyang University, Seoul, South Korea. [410]Academic Unit of Surgery, School of Medicine, College of Medical, Veterinary and Life Sciences, University of Glasgow, Glasgow Royal Infirmary, Glasgow, UK. [411]Department of Pathology, Asan Medical Center, College of Medicine, Ulsan University, Songpa-gu, Seoul, South Korea. [412]Science Writer, Garrett Park, MD, USA. [413]International Cancer Genome Consortium (ICGC)/ICGC Accelerating Research in Genomic Oncology (ARGO) Secretariat, Ontario Institute for Cancer Research, Toronto, ON, Canada. [414]University of Ljubljana, Ljubljana, Slovenia. [415]Department of Public Health Sciences, University of Chicago, Chicago, IL, USA. [416]Research Institute, NorthShore University HealthSystem, Evanston, IL, USA. [417]Department for Biomedical Research, University of Bern, Bern, Switzerland. [418]Centre of Genomics and Policy, McGill University and Génome Québec Innovation Centre, Montreal, QC, Canada. [419]Carolina Center for Genome Sciences, University of North Carolina at Chapel Hill, Chapel Hill, NC, USA. [420]Hopp Children's Cancer Center (KiTZ), Heidelberg, Germany. [421]Pediatric Glioma Research Group, German Cancer Research Center (DKFZ), Heidelberg, Germany. [422]Cancer Research UK, London, UK. [423]Indivumed GmbH, Hamburg, Germany. [424]Genome Integration Data Center, Syntekabio, Inc, Daejeon, South Korea. [425]University Hospital Zurich, Zurich, Switzerland. [426]Clinical Bioinformatics, Swiss Institute of Bioinformatics, Geneva, Switzerland. [427]Institute for Pathology and Molecular Pathology, University Hospital Zurich, Zurich, Switzerland. [428]Institute of Molecular Life Sciences, University of Zurich, Zurich, Switzerland. [429]Women's Cancer Program at the Samuel Oschin Comprehensive Cancer Institute, Cedars-Sinai Medical Center, Los Angeles, CA, USA. [430]Department for Internal Medicine II, University Hospital Schleswig-Holstein, Kiel, Germany. [431]Genetics and Molecular Pathology, SA Pathology, Adelaide, SA, Australia. [432]Department of Gastric Surgery, National Cancer Center Hospital, Tokyo, Japan. [433]A.A. Kharkevich Institute of Information Transmission Problems, Moscow, Russia. [434]Oncology and Immunology, Dmitry Rogachev National Research Center of Pediatric Hematology, Moscow, Russia. [435]Skolkovo Institute of Science and Technology, Moscow, Russia. [436]Department of Surgery, The George Washington University, School of Medicine and Health Science, Washington, DC, USA. [437]Endocrine Oncology Branch, Center for Cancer Research, National Cancer Institute, National Institutes of Health, Bethesda, MD, USA. [438]Melanoma Institute Australia, Macquarie University, Sydney, NSW, Australia. [439]MIT Computer Science and Artificial Intelligence Laboratory, Massachusetts Institute of Technology, Cambridge, MA, USA. [440]Tissue Pathology and Diagnostic Oncology, Royal Prince Alfred Hospital, Sydney, NSW, Australia. [441]Cholangiocarcinoma Screening and Care Program and Liver Fluke and Cholangiocarcinoma Research Centre, Faculty of Medicine, Khon Kaen University, Khon Kaen, Thailand. [442]Controlled Department and Institution, New York, NY, USA. [443]National Cancer Center, Gyeonggi, South Korea. [444]Department of Biochemistry, College of Medicine, Ewha Womans University, Seoul, South Korea. [445]Health Sciences Department of Biomedical Informatics, University of California San Diego, La Jolla, CA, USA. [446]Research Core Center, National Cancer Centre Korea, Goyang-si, South Korea. [447]Department of Health Sciences and Technology, Sungkyunkwan University School of Medicine, Seoul, South Korea. [448]Samsung Genome Institute, Seoul, South Korea. [449]Breast Oncology Program, Dana-Farber/Brigham and Women's Cancer Center, Boston, MA, USA. [450]Department of Surgery, Memorial Sloan Kettering Cancer Center, New York, NY, USA. [451]Division of Breast Surgery, Brigham and Women's Hospital, Boston, MA, USA. [452]Integrative Bioinformatics Support Group, National Institute of Environmental Health Sciences (NIEHS), Durham, NC, USA. [453]Department of Clinical Science, University of Bergen, Bergen, Norway. [454]Center For Medical Innovation, Seoul National University Hospital, Seoul, South Korea. [455]Department of Internal Medicine, Seoul National University Hospital, Seoul, South Korea. [456]Institute of Computer Science, Polish Academy of Sciences, Warsawa, Poland. [457]Functional and Structural Genomics, German Cancer Research Center (DKFZ), Heidelberg, Germany. [458]Laboratory of Translational Genomics, Division of Cancer Epidemiology and Genetics, National Cancer Institute, National Institutes of Health, Bethesda, MD, USA. [459]Institute for Medical Informatics Statistics and Epidemiology, University of Leipzig, Leipzig, Germany. [460]Morgan Welch Inflammatory Breast Cancer Research Program and Clinic, The University of Texas MD Anderson Cancer Center, Houston, TX, USA. [461]Department of Hematology and Oncology, Georg-Augusts-University of Göttingen, Göttingen, Germany. [462]Institute of Cell Biology (Cancer Research), University of Duisburg-Essen, Essen, Germany. [463]King's College London and Guy's and St. Thomas' NHS Foundation Trust, London, UK. [464]Center for Epigenetics, Van Andel Research Institute, Grand Rapids, MI, USA. [465]The University of Queensland Centre for Clinical Research, Royal Brisbane and Women's Hospital, Herston, QLD, Australia. [466]Department of Pediatric Oncology and Hematology, University of Cologne, Cologne, Germany. [467]University of Düsseldorf, Düsseldorf, Germany. [468]Department of Pathology, Institut Jules Bordet, Brussels, Belgium. [469]Institute of Biomedicine, Sahlgrenska Academy at University of Gothenburg, Gothenburg, Sweden. [470]Children's Medical Research Institute, Sydney, NSW, Australia. [471]ILSbio, LLC Biobank, Chestertown, MD, USA. [472]Division of Genetics and Genomics, Boston Children's Hospital, Harvard Medical School, Boston, MA, USA. [473]Institute for Bioengineering and Biopharmaceutical Research (IBBR), Hanyang University, Seoul, South Korea. [474]Department of Statistics, University of California Santa Cruz, Santa Cruz, CA, USA. [475]Department of Vertebrate Genomics/Otto Warburg Laboratory Gene Regulation and Systems Biology of Cancer, Max Planck Institute for Molecular Genetics, Berlin, Germany. [476]McGill University and Genome Quebec Innovation Centre, Montreal, QC, Canada. [477]biobyte solutions GmbH, Heidelberg, Germany. [478]Gynecologic Oncology, NYU Laura and Isaac Perlmutter Cancer Center, New York University, New York, NY, USA. [479]Division of Oncology, Stem Cell Biology Section, Washington University School of Medicine, St. Louis, MO, USA. [480]Harvard University,

Cambridge, MA, USA. [481]Urologic Oncology Branch, Center for Cancer Research, National Cancer Institute, National Institutes of Health, Bethesda, MD, USA. [482]University of Oslo, Oslo, Norway. [483]University of Toronto, Toronto, ON, Canada. [484]School of Life Sciences, Peking University, Beijing, China. [485]Leidos Biomedical Research, Inc, McLean, VA, USA. [486]Second Military Medical University, Shanghai, China. [487]Chinese Cancer Genome Consortium, Shenzhen, China. [488]Department of Medical Oncology, Beijing Hospital, Beijing, China. [489]Laboratory of Molecular Oncology, Key Laboratory of Carcinogenesis and Translational Research (Ministry of Education), Peking University Cancer Hospital and Institute, Beijing, China. [490]School of Medicine/School of Mathematics and Statistics, University of St. Andrews, St, Andrews, Fife, UK. [491]Institute for Systems Biology, Seattle, WA, USA. [492]Department of Biochemistry and Molecular Biology, Faculty of Medicine, University Institute of Oncology-IUOPA, Oviedo, Spain. [493]Institut Bergonié, Bordeaux, France. [494]Cancer Unit, MRC University of Cambridge, Cambridge, UK. [495]Department of Pathology and Laboratory Medicine, Center for Personalized Medicine, Children's Hospital Los Angeles, Los Angeles, CA, USA. [496]John Curtin School of Medical Research, Canberra, ACT, Australia. [497]MVZ Department of Oncology, PraxisClinic am Johannisplatz, Leipzig, Germany. [498]Department of Information Technology, Ghent University, Ghent, Belgium. [499]Department of Plant Biotechnology and Bioinformatics, Ghent University, Ghent, Belgium. [500]Institute for Genomic Medicine, Nationwide Children's Hospital, Columbus, OH, USA. [501]Department of Surgery, Duke University, Durham, NC, USA. [502]Institució Catalana de Recerca i Estudis Avançats (ICREA), Barcelona, Spain. [503]Institut Català de Paleontologia Miquel Crusafont, Universitat Autònoma de Barcelona, Barcelona, Spain. [504]Institut d'Investigacions Biomèdiques August Pi i Sunyer (IDIBAPS), Barcelona, Spain. [505]Division of Oncology, Washington University School of Medicine, St. Louis, MO, USA. [506]Department of Surgery and Cancer, Imperial College, London, INY, UK. [507]Applications Department, Oxford Nanopore Technologies, Oxford, UK. [508]Department of Obstetrics, Gynecology and Reproductive Services, University of California San Francisco, San Francisco, CA, USA. [509]Department of Biochemistry and Molecular Medicine, University California at Davis, Sacramento, CA, USA. [510]STTARR Innovation Facility, Princess Margaret Cancer Centre, Toronto, ON, Canada. [511]Discipline of Surgery, Western Sydney University, Penrith, NSW, Australia. [512]Yale School of Medicine, Yale University, New Haven, CT, USA. [513]Department of Genetics, Lineberger Comprehensive Cancer Center, University of North Carolina at Chapel Hill, Chapel Hill, NC, USA. [514]Departments of Neurology and Neurosurgery, Henry Ford Hospital, Detroit, MI, USA. [515]Precision Oncology, OHSU Knight Cancer Institute, Oregon Health and Science University, Portland, OR, USA. [516]Institute of Pathology, University Medical Center Hamburg-Eppendorf, Hamburg, Germany. [517]Department of Health Sciences, Faculty of Medical Sciences, Kyushu University, Fukuoka, Japan. [518]Heidelberg Academy of Sciences and Humanities, Heidelberg, Germany. [519]Department of Clinical Pathology, University of Melbourne, Melbourne, VIC, Australia. [520]Department of Pathology, Roswell Park Cancer Institute, Buffalo, NY, USA. [521]Department of Computer Science, University of Helsinki, Helsinki, Finland. [522]Institute of Biotechnology, University of Helsinki, Helsinki, Finland. [523]Organismal and Evolutionary Biology Research Programme, University of Helsinki, Helsinki, Finland. [524]Department of Obstetrics and Gynecology, Division of Gynecologic Oncology, Washington University School of Medicine, St. Louis, MO, USA. [525]Penrose St. Francis Health Services, Colorado Springs, CO, USA. [526]Institute of Pathology, Ulm University and University Hospital of Ulm, Ulm, Germany. [527]National Cancer Center, Tokyo, Japan. [528]Genome Institute of Singapore, Singapore, Singapore. [529]German Cancer Aid, Bonn, Germany. [530]Programme in Cancer and Stem Cell Biology, Centre for Computational Biology, Duke-NUS Medical School, Singapore, Singapore. [531]The Chinese University of Hong Kong, Shatin, NT, Hong Kong, China. [532]Fourth Military Medical University, Shaanxi, China. [533]St. Jude Children's Research Hospital, Memphis, TN, USA. [534]University Health Network, Princess Margaret Cancer Centre, Toronto, ON, Canada. [535]Center for Biomolecular Science and Engineering, University of California Santa Cruz, Santa Cruz, CA, USA. [536]Department of Medicine, University of Chicago, Chicago, IL, USA. [537]Department of Neurology, Mayo Clinic, Rochester, MN, USA. [538]Cambridge Oesophagogastric Centre, Cambridge University Hospitals NHS Foundation Trust, Cambridge, UK. [539]Department of Computer Science, Carleton College, Northfield, MN, USA. [540]Institute of Cancer Sciences, College of Medical Veterinary and Life Sciences, University of Glasgow, Glasgow, UK. [541]Department of Epidemiology, University of Alabama at Birmingham, Birmingham, AL, USA. [542]HudsonAlpha Institute for Biotechnology, Huntsville, AL, USA. [543]O'Neal Comprehensive Cancer Center, University of Alabama at Birmingham, Birmingham, AL, USA. [544]Department of Pathology, Keio University School of Medicine, Tokyo, Japan. [545]Department of Hepatobiliary and Pancreatic Oncology, National Cancer Center Hospital, Tokyo, Japan. [546]Lymphoma Genomic Translational Research Laboratory, National Cancer Centre, Singapore, Singapore. [547]Department of Clinical Pathology, Robert-Bosch-Hospital, Stuttgart, Germany. [548]Department of Cell and Systems Biology, University of Toronto, Toronto, ON, Canada. [549]Department of Biosciences and Nutrition, Karolinska Institutet, Stockholm, Sweden. [550]Center for Liver Cancer, Research Institute and Hospital, National Cancer Center, Gyeonggi, South Korea. [551]Division of Hematology-Oncology, Samsung Medical Center, Sungkyunkwan University School of Medicine, Seoul, South Korea. [552]Samsung Advanced Institute for Health Sciences and Technology, Sungkyunkwan University School of Medicine, Seoul, South Korea. [553]Cheonan Industry-Academic Collaboration Foundation, Sangmyung University, Cheonan, South Korea. [554]NYU Langone Medical Center, New York, NY, USA. [555]Department of Hematology and Medical Oncology, Cleveland Clinic, Cleveland, OH, USA. [556]Department of Health Sciences Research, Mayo Clinic, Rochester, MN, USA. [557]Helen F. Graham Cancer Center at Christiana Care Health Systems, Newark, DE, USA. [558]Heidelberg University Hospital, Heidelberg, Germany. [559]CSRA Incorporated, Fairfax, VA, USA. [560]Research Department of Pathology, University College London Cancer Institute, London, UK. [561]Department of Research Oncology, Guy's Hospital, King's Health Partners AHSC, King's College London School of Medicine, London, UK. [562]Faculty of Medicine and Health Sciences, Macquarie University, Sydney, NSW, Australia. [563]University Hospital of Minjoz, INSERM UMR1098, Besançon, France. [564]Spanish National Cancer Research Centre, Madrid, Spain. [565]Center of Digestive Diseases and Liver Transplantation, Fundeni Clinical Institute, Bucharest, Romania. [566]Cureline, Inc, South San Francisco, CA, USA. [567]St. Luke's Cancer Centre, Royal Surrey County Hospital NHS Foundation Trust, Guildford, UK. [568]Cambridge Breast Unit, Addenbrooke's Hospital, Cambridge University Hospital NHS Foundation Trust and NIHR Cambridge Biomedical Research Centre, Cambridge, UK. [569]East of Scotland Breast Service, Ninewells Hospital, Aberdeen, UK. [570]Department of Genetics, Microbiology and Statistics, University of Barcelona, IRSJD, IBUB, Barcelona, Spain. [571]Department of Obstetrics and Gynecology, Medical College of Wisconsin, Milwaukee, WI, USA. [572]Hematology and Medical Oncology, Winship Cancer Institute of Emory University, Atlanta, GA, USA. [573]Department of Computer Science, Princeton University, Princeton, NJ, USA. [574]Vanderbilt Ingram Cancer Center, Vanderbilt University, Nashville, TN, USA. [575]Ohio State University College of Medicine and Arthur G. James Comprehensive Cancer Center, Columbus, OH, USA. [576]Department of Surgery, Yokohama City University Graduate School of Medicine, Kanagawa, Japan. [577]Research Computing Center, University of North Carolina at Chapel Hill, Chapel Hill, NC, USA. [578]School of Molecular Biosciences and Center for Reproductive Biology, Washington State University, Pullman, WA, USA. [579]Finsen Laboratory and Biotech Research and Innovation Centre (BRIC), University of Copenhagen, Copenhagen, Denmark. [580]Department of Laboratory Medicine and Pathobiology, University of Toronto, Toronto, ON, Canada. [581]Department of Pathology, Human Oncology and Pathogenesis Program, Memorial Sloan Kettering Cancer Center, New York, NY, USA. [582]University Hospital Giessen, Pediatric Hematology and Oncology, Giessen, Germany. [583]Oncologie Sénologie, ICM Institut Régional du Cancer, Montpellier, France. [584]Institute of Clinical Molecular Biology, Christian-Albrechts-University, Kiel, Germany. [585]Institute of Pathology, University of Wuerzburg, Wuerzburg, Germany. [586]Department of Urology, North Bristol NHS Trust, Bristol, UK. [587]SingHealth, Duke-NUS Institute of Precision Medicine, National Heart Centre Singapore, Singapore, Singapore. [588]Department of Computer Science, University of Toronto, Toronto, ON, Canada. [589]Englander Institute for Precision Medicine, Weill Cornell Medicine and New York Presbyterian Hospital, New York, NY, USA.

[590]Vall d'Hebron Institute of Oncology: VHIO, Barcelona, Spain. [591]General and Hepatobiliary-Biliary Surgery, Pancreas Institute, University and Hospital Trust of Verona, Verona, Italy. [592]National Centre for Biological Sciences, Tata Institute of Fundamental Research, Bangalore, India. [593]Indiana University, Bloomington, IN, USA. [594]Department of Pathology, GZA-ZNA Hospitals, Antwerp, Belgium. [595]Analytical Biological Services, Inc, Wilmington, DE, USA. [596]Sydney Medical School, University of Sydney, Sydney, NSW, Australia. [597]cBio Center, Dana-Farber Cancer Institute, Harvard Medical School, Boston, MA, USA. [598]Department of Cell Biology, Harvard Medical School, Boston, MA, USA. [599]Advanced Centre for Treatment Research and Education in Cancer, Tata Memorial Centre, Navi Mumbai, Maharashtra, India. [600]School of Environmental and Life Sciences, Faculty of Science, The University of Newcastle, Ourimbah, NSW, Australia. [601]Department of Dermatology, University Hospital of Essen, Essen, Germany. [602]Bioinformatics and Omics Data Analytics, German Cancer Research Center (DKFZ), Heidelberg, Germany. [603]Department of Urology, Charité Universitätsmedizin Berlin, Berlin, Germany. [604]Martini-Clinic, Prostate Cancer Center, University Medical Center Hamburg-Eppendorf, Hamburg, Germany. [605]Department of General Internal Medicine, University of Kiel, Kiel, Germany. [606]German Cancer Consortium (DKTK), Partner site Berlin, Berlin, Germany. [607]Cancer Research Institute, Beth Israel Deaconess Medical Center, Boston, MA, USA. [608]University of Pittsburgh, Pittsburgh, PA, USA. [609]Department of Ophthalmology and Ocular Genomics Institute, Massachusetts Eye and Ear, Harvard Medical School, Boston, MA, USA. [610]Center for Psychiatric Genetics, NorthShore University HealthSystem, Evanston, IL, USA. [611]Van Andel Research Institute, Grand Rapids, MI, USA. [612]Laboratory of Molecular Medicine, Human Genome Center, Institute of Medical Science, University of Tokyo, Tokyo, Japan. [613]Japan Agency for Medical Research and Development, Tokyo, Japan. [614]Korea University, Seoul, South Korea. [615]Murtha Cancer Center, Walter Reed National Military Medical Center, Bethesda, MD, USA. [616]Center for RNA Interference and Noncoding RNA, The University of Texas MD Anderson Cancer Center, Houston, TX, USA. [617]Department of Experimental Therapeutics, The University of Texas MD Anderson Cancer Center, Houston, TX, USA. [618]Department of Gynecologic Oncology and Reproductive Medicine, The University of Texas MD Anderson Cancer Center, Houston, TX, USA. [619]University Hospitals Coventry and Warwickshire NHS Trust, Coventry, UK. [620]Department of Radiation Oncology, Radboud University Nijmegen Medical Centre, Nijmegen, GA, The Netherlands. [621]Institute for Genomics and Systems Biology, University of Chicago, Chicago, IL, USA. [622]Clinic for Hematology and Oncology, St.-Antonius-Hospital, Eschweiler, Germany. [623]Computational and Systems Biology Program, Memorial Sloan Kettering Cancer Center, New York, NY, USA. [624]University of Iceland, Reykjavik, Iceland. [625]Division of Computational Genomics and Systems Genetics, German Cancer Research Center (DKFZ), Heidelberg, Germany. [626]Dundee Cancer Centre, Ninewells Hospital, Dundee, UK. [627]Department for Internal Medicine III, University of Ulm and University Hospital of Ulm, Ulm, Germany. [628]Institut Curie, INSERM Unit 830, Paris, France. [629]Department of Gastroenterology and Hepatology, Yokohama City University Graduate School of Medicine, Kanagawa, Japan. [630]Department of Laboratory Medicine, Radboud University Nijmegen Medical Centre, Nijmegen, GA, The Netherlands. [631]Division of Cancer Genome Research, German Cancer Research Center (DKFZ), Heidelberg, Germany. [632]Department of General Surgery, Singapore General Hospital, Singapore, Singapore. [633]Cancer Science Institute of Singapore, National University of Singapore, Singapore, Singapore. [634]Department of Medical and Clinical Genetics, Genome-Scale Biology Research Program, University of Helsinki, Helsinki, Finland. [635]East Anglian Medical Genetics Service, Cambridge University Hospitals NHS Foundation Trust, Cambridge, UK. [636]Irving Institute for Cancer Dynamics, Columbia University, New York, NY, USA. [637]Institute of Molecular and Cell Biology, Singapore, Singapore. [638]Laboratory of Cancer Epigenome, Division of Medical Science, National Cancer Centre Singapore, Singapore, Singapore. [639]Universite Lyon, INCa-Synergie, Centre Léon Bérard, Lyon, France. [640]Department of Urology, Mayo Clinic, Rochester, MN, USA. [641]Royal National Orthopaedic Hospital - Stanmore, Stanmore, Middlesex, UK. [642]Giovanni Paolo II/I.R.C.C.S. Cancer Institute, Bari, BA, Italy. [643]Neuroblastoma Genomics, German Cancer Research Center (DKFZ), Heidelberg, Germany. [644]Fondazione Policlinico Universitario Gemelli IRCCS, Rome, Italy, Rome, Italy. [645]University of Verona, Verona, Italy. [646]Centre National de Génotypage, CEA - Institute de Génomique, Evry, France. [647]CAPHRI Research School, Maastricht University, Maastricht, ER, The Netherlands. [648]Department of Biopathology, Centre Léon Bérard, Lyon, France. [649]Université Claude Bernard Lyon1, Villeurbanne, France. [650]Core Research for Evolutional Science and Technology (CREST), JST, Tokyo, Japan. [651]Department of Biological Sciences, Laboratory for Medical Science Mathematics, Graduate School of Science, University of Tokyo, Yokohama, Japan. [652]Department of Medical Science Mathematics, Medical Research Institute, Tokyo Medical and Dental University (TMDU), Tokyo, Japan. [653]University Hospitals Birmingham NHS Foundation Trust, Birmingham, UK. [654]Centre for Cancer Research and Cell Biology, Queen's University, Belfast, UK. [655]Breast Medical Oncology, The University of Texas MD Anderson Cancer Center, Houston, TX, USA. [656]Department of Surgery, Johns Hopkins University School of Medicine, Baltimore, MD, USA. [657]Department of Oncology-Pathology, Science for Life Laboratory, Karolinska Institute, Stockholm, Sweden. [658]School of Cancer Sciences, Faculty of Medicine, University of Southampton, Southampton, UK. [659]Department of Gene Technology, Tallinn University of Technology, Tallinn, Estonia. [660]Genetics and Genome Biology Program, SickKids Research Institute, The Hospital for Sick Children, Toronto, ON, Canada. [661]Departments of Neurosurgery and Hematology and Medical Oncology, Winship Cancer Institute and School of Medicine, Emory University, Atlanta, GA, USA. [662]Department of Clinical and Molecular Medicine, Faculty of Medicine and Health Sciences, Norwegian University of Science and Technology, Trondheim, Norway. [663]Argmix Consulting, North Vancouver, BC, Canada. [664]Department of Information Technology, Ghent University, Interuniversitair Micro-Electronica Centrum (IMEC), Ghent, Belgium. [665]Nuffield Department of Surgical Sciences, John Radcliffe Hospital, University of Oxford, Oxford, UK. [666]Institute of Mathematics and Computer Science, University of Latvia, Riga, LV, Latvia. [667]Discipline of Pathology, Sydney Medical School, University of Sydney, Sydney, NSW, Australia. [668]Department of Applied Mathematics and Theoretical Physics, Centre for Mathematical Sciences, University of Cambridge, Cambridge, UK. [669]Department of Epidemiology and Biostatistics, Memorial Sloan Kettering Cancer Center, New York, NY, USA. [670]Department of Statistics, Columbia University, New York, NY, USA. [671]Department of Immunology, Genetics and Pathology, Science for Life Laboratory, Uppsala University, Uppsala, Sweden. [672]School of Electronic and Information Engineering, Xi'an Jiaotong University, Xi'an, China. [673]Department of Histopathology, Cambridge University Hospitals NHS Foundation Trust, Cambridge, UK. [674]Oxford NIHR Biomedical Research Centre, University of Oxford, Oxford, UK. [675]Georgia Regents University Cancer Center, Augusta, GA, USA. [676]Wythenshawe Hospital, Manchester, UK. [677]Department of Genetics, Washington University School of Medicine, St. Louis, MO, USA. [678]Department of Biological Oceanography, Leibniz Institute of Baltic Sea Research, Rostock, Germany. [679]Wellcome Centre for Human Genetics, University of Oxford, Oxford, UK. [680]Department of Molecular and Human Genetics, Baylor College of Medicine, Houston, TX, USA. [681]Thoracic Oncology Laboratory, Mayo Clinic, Rochester, MN, USA. [682]Institute for Genomic Medicine, Nationwide Children's Hospital, Columbus, OH, USA. [683]Division of Gynecologic Oncology, Department of Obstetrics and Gynecology, Mayo Clinic, Rochester, MN, USA. [684]International Institute for Molecular Oncology, Poznań, Poland. [685]Poznan University of Medical Sciences, Poznań, Poland. [686]Genomics and Proteomics Core Facility High Throughput Sequencing Unit, German Cancer Research Center (DKFZ), Heidelberg, Germany. [687]NCCS-VARI Translational Research Laboratory, National Cancer Centre Singapore, Singapore, Singapore. [688]Edison Family Center for Genome Sciences and Systems Biology, Washington University, St. Louis, MO, USA. [689]Department of Medical Informatics and Clinical Epidemiology, Division of Bioinformatics and Computational Biology, OHSU Knight Cancer Institute, Oregon Health and Science University, Portland, OR, USA. [690]School of Electronic Information and Communications, Huazhong University of Science and Technology, Wuhan, China. [691]Department of Applied Mathematics and Statistics, Johns Hopkins University, Baltimore, MD, USA. [692]Department of Cancer Genome Informatics, Graduate School of Medicine, Osaka University,

Osaka, Japan. [693]School of Mathematics and Statistics, University of Sydney, Sydney, NSW, Australia. [694]Ben May Department for Cancer Research and Department of Human Genetics, University of Chicago, Chicago, IL, USA. [695]Tri-Institutional PhD Program in Computational Biology and Medicine, Weill Cornell Medicine, New York, NY, USA. [696]The First Affiliated Hospital, Xi'an Jiaotong University, Xi'an, China. [697]Department of Medicine and Therapeutics, The Chinese University of Hong Kong, Shatin, NT, Hong Kong, China. [698]Department of Biostatistics, The University of Texas MD Anderson Cancer Center, Houston, TX, USA. [699]Duke-NUS Medical School, Singapore, Singapore. [700]Department of Surgery, Ruijin Hospital, Shanghai Jiaotong University School of Medicine, Shanghai, China. [701]Division of Orthopaedic Surgery, Oslo University Hospital, Oslo, Norway. [702]Eastern Clinical School, Monash University, Melbourne, VIC, Australia. [703]Epworth HealthCare, Richmond, VIC, Australia. [704]Department of Biostatistics and Computational Biology, Dana-Farber Cancer Institute and Harvard Medical School, Boston, MA, USA. [705]Department of Biomedical Informatics, College of Medicine, The Ohio State University, Columbus, OH, USA. [706]The Ohio State University Comprehensive Cancer Center (OSUCCC – James), Columbus, OH, USA. [707]BIOPIC, ICG and College of Life Sciences, Peking University, Beijing, China. [708]The University of Texas School of Biomedical Informatics (SBMI) at Houston, Houston, TX, USA. [709]Department of Biostatistics, University of North Carolina at Chapel Hill, Chapel Hill, NC, USA. [710]Department of Biochemistry and Molecular Genetics, Feinberg School of Medicine, Northwestern University, Chicago, IL, USA. [711]Faculty of Medicine and Health, University of Sydney, Sydney, NSW, Australia. [712]Department of Pathology, Erasmus Medical Center Rotterdam, Rotterdam, GD, The Netherlands. [713]Division of Molecular Carcinogenesis, The Netherlands Cancer Institute, Amsterdam, CX, The Netherlands. [714]Institute of Molecular Life Sciences and Swiss Institute of Bioinformatics, University of Zurich, Zurich, Switzerland.

