## [Peer Review File · Nature Communications]

Reviewers' comments:

Reviewer #1 (Remarks to the Author):

In their manuscript Li et al present a meticulous analysis of sex differences in cancer genomes highlighting differences in mutational burden, frequency of specific mutations, copy number variation and subclonal heterogeneity. These are very important and timely results as they will guide and support future studies of mechanisms by which these sex differences arise, and what their implications are for cancer biology and treatment. In general this is a well-presented and thorough analysis.

I have only one concern and that is a lack of clarity about the methods by which ancestry was included in this analysis. For multiple cancers, there are marked differences in rate as a function of self-reported race. This raises two issues for an analysis like this. First, self-reported race is a near meaningless parameter in genetic studies. It needs to be clear whether genetic ancestry was established for each patient, and if so, how. If self-reported race was used, then a justification for this and discussion of its limitations should be included. Second, it will be important to determine whether sex differences are equally extant across genetic ancestries. A cross genetic ancestry analysis may be beyond the scope of this study but it would be helpful if the authors commented on this in the context of their use of ancestry in their analyses.

Reviewer #2 (Remarks to the Author):

The authors have analyzed whole genome data in a sex-stratified way to look for sex-differences in mutational profiles across tumors. This is a timely analysis.

Overall the paper is clearly written, and flows well.

However, I have several questions that may affect the conclusions of the manuscript. In particular:

1. Is it appropriate to consider all tumors together?

For example, if liver cancer has a 4:1 female incidence. And one finds that males in liver cancer have 10% more CTNNB1 mutations than females, but no other tumor shows this pattern, if liver cancer is overrepresented in the number of tumor samples, by grouping all samples together, it could look like a global sex bias across all tumors that is driven by just a single cancer.

After reading the paper in full, this is indeed the case, and I caution the authors to be more direct up front with the limitations of a pan-cancer analysis for sex-differences.

2. Does it make sense to include an unequal number of male and female samples. Did the authors do any sort of permutation test to confirm that the proportions in males (for which there are more) are significant after permuting and downsampling to the same number as females. Overall, some of the differences could be due to sampling bias. I appreciate that the authors attempted to get around this by investigating proportions, but these can still be affected. Can the authors more thoroughly address the concerns of sampling bias and unequal sample sizes between males and females?

3. While the mutations that occurred in male or female tumors more often may reflect the true biology, did the authors also account for pre-existing germline mutations? That is, if the females, for whatever reason, were sampled in such a way that their germline already had the CTNNB1 mutation, then they wouldn't show up as a tumor-specific mutation in female tumors. But, they

would all have it. I'm just curious how pre-existing germline mutations could bias the estimates of sex differences. Perhaps the authors will consider this outside the scope of the paper, but worth mentioning, if there is space.

4. I don't know what Box-cox transformed mutation load is, and would need someone else to evaluate this statistical framework. How does this account for potential confounders?

5. The parenthetical reference to higher mutational prevalence in male-derived samples should be in a table. It is exceedingly difficult to parse in the very long comment, as it stands now (lines 95-99).

6. Why would a patient's age be more important than the age of the tumor? Or maybe they corrected for age of the tumor, but that seems unlikely.

7. How did the authors adjust for ancestry? Did they do ancestry analysis, or did they use reported race? The latter would be problematic. The former would be curious as to whether they used bins, or proportions of inferred ancestry, and relative to which populations... 1000 genomes?

8. For the 15 tumor types with at least 15 samples, do the authors mean 15 of each sex, or 15 total - please clarify in the main text.

9. I don't understand the rationale for the test, or the potential mechanism of a promoter mutation being associated with a globally increased codon mutation burden (but not genome-wide mutation burden). That is, why would it only lead to more mutations in codons, but not elsewhere in the genome? What is the possible mechanism? And what are the chances of observing this correlation by chance?

10. I do not understand what the "matched mutational timing data" is? Can the authors explain this, and how they can come to the conclusion that TERT promoter mutation frequency is associated with sex-biased coding mutational load because of it?

11. Were polyclonal versus monoclonal determined from one bulk tumor sample? This seems difficult at best to infer, without multi regional sampling. Also, is the age of the tumors known? Is it possible that female tumors were detected earlier, and so were more likely to be monoclonal?

12. Can the authors comment on why there are 4,285 gene gains in males across all tumor types, but none that are female-biased? This seems, on sheer numbers alone, to be a technical artifact of some sort.

13. The case of female-specific losses showing up in liver cancer, but no losses in the full analysis is a good argument for why the tumors shouldn't be lumped together.

14. The "Pan-cancer analysis of whole genomes" cited for us to find the genomics methodology is a 2017 preprint: <https://www.biorxiv.org/content/10.1101/162784v1>. Can the authors confirm that this is the most updated version? It notes there will be a complete version updated, but I couldn't immediately find it. Then, upon investigation, it is nearly impossible to find the exact pipeline that was run. Did the authors really align to version 37 of the human genome? Version 38 was released in 2013. There is no excuse for using version 37.

15. Did the authors look at whether there was any difference in the distribution of sequencing types used for males versus females? That is, how many male samples were run on the HiSeq 100bp versus 150bp? How many female samples for each? This could 100% explain discrepancies.

16. How did the male/female samples separate out among the 75 donors that were flagged for QA for "a variety of reasons"? Including an unexpectedly high fraction of paired reads mapping to

different chromosomes, unusual mutational signatures, and contamination?

Reviewer #3 (Remarks to the Author):

In this manuscript, Li et al. evaluate the genomic differences between cancers in males and females using pan-cancer whole-genome sequencing data from the ICGC Pan-Cancer Analysis of Whole Genomes (PCAWG) project. The paper largely focuses on statistical analysis and identification of driver genes, mutation load, tumor evolution, copy-number changes, and mutational signatures with differences between the sexes.

While this manuscript is largely descriptive, it does try to answer an important enigma in cancer epidemiology: why are sex differences observed in most cancers outside sex organs? Indeed, understanding the molecular mechanisms underlying such difference will have far-reaching implications. Unfortunately, I do not find this manuscript to be appropriately addressing this question or to be providing any significant insights beyond previous publications. Further, I do have major concerns with the performed (or at least the description and reproducibility of the performed) statistical analysis. At its current state, I cannot recommend this manuscript for acceptance as major work/re-analysis is required.

Concerns about Statistical Analyses:

The manuscript heavily relies on statistical analysis. However, this analysis is neither properly described nor particularly reproducible.

For example, in the section for driver gene, page 2 line 60 states: "adjust for tumour subtype, ancestry and age (Online Methods)". As far as I can tell, there is nothing in the Online methods discussing the approach for adjusting for these parameters. Further, derivation of these parameters is not described. What was the granularity of tumor subtype (e.g., did you separate breast cancer into multiple subtypes)? More importantly, how were sex and ancestry determined (self-reporting vs inferring from the available genomics data)?

It is also unclear whether these (or any adjustments) were made for the other sets of statistical analysis (e.g., tumor evolution, copy-number changes, etc.). I am not sure if Supplementary Table 1 is applicable to all analysis or just the ones in the driver gene section. Moreover, there are at least two additional parameters that need to be included in the adjustments: stage and treatment (i.e., prior chemotherapy; a number of the PCAWG samples are post chemotherapy).

The Box-Cox Transformation is a good approach to reduce non-additivity, non-normality and heteroscedasticity of data. However, the transformation only reduces these problems and it does not eliminate them. The authors should really use robust statistical approaches, e.g., robust regression instead of regression as well as robust logistic regression instead of logistic regression.

Lastly, a more minor note, I cannot seem to find the supplementary materials of this manuscript. Were these missed as part of the submission? Nevertheless, I was able to download them from the bioRxiv version of the manuscript. Also, there are a number of typos throughout the manuscript (e.g., page 6 line 246: "asociated").

Concerns about Reproducibility:

This paper relies solely on statistical analysis and it is essential to be able to peruse the actual procedures as well as to use the code to reproduce the reported results. Indeed, the authors can provide one table (or even an R data frame) with the information of the patient used in the analysis (sex, cancer type, age, ancestry, etc.) and another table(s) with the genomics features used for associating sex differences. Further, I would suggest posting all of their code in a

repository (e.g., GitLab) and, thus, allowing reviewers and readers to really examine their analysis as well as to easily run the code and reproduce the results.

Concerns about Novelty/Impact of Findings:

Even excluding the technical concerns, I am worried about the novelty of the findings. The authors use quite a lot of space for CTNNB1 but sex difference in CTNNB1 were previously reported in both reference 10 & 11 of the current manuscript. The sex difference in the TERT promoter were reported in some of the early melanoma manuscripts (PMID: 24463461) albeit not in thyroid cancer. I am both surprised and concerned about the finding of sex differences in ALB and PTCH1. The mutations in these genes are in the coding sequence, however, they were not identified in the twice larger (~4500 exome) TCGA paper on differences between males and females (PMID: 27165743; reference 10 in the current manuscript). How do the authors explain that? My concern is that the number of mutations in these genes is low and they might be an artifact of the technical methodology. Further, if these are correct associations, one would expect to observe them in the analysis including larger number of samples.

The analysis of genome instability and copy-number changes does not seem to bring much more compared to the prior TCGA study. Again, reference 10 had a large sample size but used whole-exome sequencing and copy-number arrays. Nevertheless, the whole-genomes do not seem to be bringing much in terms of findings as the results seem very similar to me. What are the novel findings compared to prior studies?

Finally, on the subject of mutational signatures, comparisons of the numbers of samples with a signature versus the numbers of samples without a signature is, unfortunately, not appropriate. To avoid overfitting, the PCAWG signature analysis is using a penalty for assigning signatures with each signature required to explain at least 5% of the overall observed pattern. This results in losing lower burden mutational signatures in samples with large numbers of mutations. For example, the authors report sex difference in total mutational burden of hepatocellular carcinoma. As observed, this results in certain signatures not being assigned to female hepatocellular carcinomas when compared to male hepatocellular carcinomas. At the very least, the authors should correct for total mutational burden in their sex analysis of mutational signatures. Preferably, the authors should compare the distributions of the numbers of mutations of the signatures between males and females when the signatures have been attributed to these samples (i.e., more than zero mutations).

We thank the reviewers and the editorial staff for their thoughtful commentary and thorough review. We have addressed all specific points below, and believe that many of these changes have significantly strengthened the manuscript. Key additions include:

- Updating our two-stage statistical framework to a three-stage one that incorporates down-sampling to prune hits that might be functions of imbalanced cohorts. This gives greater confidence in our results, albeit while modestly increasing our false negative rate. In particular, we repeatedly down-sampling on tumour subtype and on sex to assess effects of imbalances in these covariates, and reject hits biased by these imbalances.
- We have extended our multivariate modeling to include stage and/or grade wherever possible to evaluate sex-bias in relative to these covariates. We also now adjust for mutation-density where appropriate in statistical models, for example in analysis of mutational signature exposures.
- We have greatly expanded the Methods to fully outline both our own statistical and computational analyses, and the origins of the different aspects of data from within the PCAWG consortium.

Reviewer #1

In their manuscript Li et al present a meticulous analysis of sex differences in cancer genomes highlighting differences in mutational burden, frequency of specific mutations, copy number variation and subclonal heterogeneity. These are very important and timely results as they will guide and support future studies of mechanisms by which these sex differences arise, and what their implications are for cancer biology and treatment. In general this is a well-presented and thorough analysis.

I have only one concern and that is a lack of clarity about the methods by which ancestry was included in this analysis. For multiple cancers, there are marked differences in rate as a function of self-reported race. This raises two issues for an analysis like this. First, self-reported race is a near meaningless parameter in genetic studies. It needs to be clear whether genetic ancestry was established for each patient, and if so, how. If self-reported race was used, then a justification for this and discussion of its limitations should be included. Second, it will be important to determine whether sex differences are equally extant across genetic ancestries. A cross genetic ancestry analysis may be beyond the scope of this study but it would be helpful if the authors commented on this in the context of their use of ancestry in their analyses.

The reviewer has raised a set of truly critical questions. In this study, ancestry was imputed by the PCAWG-8 working group based on germline SNP profiles determined by whole-genome sequencing of the reference (typically blood) sample. Full detail is provided as an attachment to the reviewers. Briefly, ADMIXTURE (PMID 19648217) was used to estimate the ancestry of donors based on 4,235 ancestry-informative SNP markers. The markers were chosen to discriminate between European, East Asian, African, Native American, and South Asian populations. This method was validated using samples from the 1000 Genomes project. This is now outlined in the Methods on Lines 595-598 which reads:

“Covariate data includes genomically matched sex, age at diagnosis, and ancestry as imputed using ADMIXTURE²² by the PCAWG-8 working group based on germline SNP profiles determined by whole-genome sequencing of the reference sample.”

The authors agree that a cross genetic ancestry analysis of sex differences would be of great interest in future work. Unfortunately the PCAWG data is not well powered for this scale of analysis. We have added discussion on this topic starting at Lines 321-323 which reads:

“Increasing the diversity of donors will also allow the study of intriguing cross-variable questions such as investigating whether sex differences are universal across races, or if there are race-specific sex differences”

Reviewer #2

The authors have analyzed whole genome data in a sex-stratified way to look for sex-differences in mutational profiles across tumors. This is a timely analysis. Overall the paper is clearly written, and flows well. However, I have several questions that may affect the conclusions of the manuscript. In particular:

We thank the Reviewer for their discerning insight on our manuscript and analysis. To address the Reviewer's concerns on imbalances in the data, we have added repeated down-sampling analysis to assess the impact of uneven sex and tumour subtype data. We have also made a number of changes to the text to clarify our methods and better communicate our results. We respond to each specific comment below.

1. Is it appropriate to consider all tumors together? For example, if liver cancer has a 4:1 female incidence. And one finds that males in liver cancer have 10% more CTNNB1 mutations than females, but no other tumor shows this pattern, if liver cancer is overrepresented in the number of tumor samples, by grouping all samples together, it could look like a global sex bias across all tumors that is driven by just a single cancer. After reading the paper in full, this is indeed the case, and I caution the authors to be more direct up front with the limitations of a pan-cancer analysis for sex-differences.

The reviewer raises excellent points on the effects of overrepresented tumour types and sex-biased incidence. To address this we have both added or modified several different analyses. For each significant pan-cancer finding, we performed repeated down-sampling on tumour subtypes to balance the effect of overrepresented tumour subtypes like Liver-HCC (n=314) and Panc-AdenoCA (n=232). We down-sampled tumour subtypes with more samples than the median tumour subtype sample size (n=48) to this median, giving us a down-sampled pan-cancer dataset of n=835. We re-ran our univariate tests and repeated the down-sampling 10,000 times. From each run we recorded the male and female proportions (or medians) and the p-values from the proportions tests (or Mann-Whitney U-tests). Cases where the median p-value was below the threshold of $p=0.05$ were kept in the revised manuscript. These are presented in Supplementary Figure 2 and detailed below:

Driver-SNV analysis: removed

Our findings of sex-biased PTCH1, CTNNB1 and ALB were indeed found to be strongly influenced by individual tumour types. The median resampled p-values did not pass the $p=0.05$ threshold, and in many cases the recorded effects (proportion of affected tumours) were in the opposite direction as initially observed in the full dataset (Supplementary Figure 2a-c). The pan-cancer driver SNV driver analysis was removed from the revised manuscript.

SNV density analysis: retained

All effects were in the same direction as initially observed, with median male-derived SNV density consistently higher than median female-derived SNV density. The p-values were all <0.05 (Supplementary Figure 2d-f).

PGA analysis: retained

The median p-value passed the $p=0.05$ threshold and median PGA was male-biased in the majority of down-sampling tests (Supplementary Figure 2g).

CNA-gains: 1,373 genes removed; 2,912 genes retained

We performed repeated tumour subtype down-sampling analysis for each gene found to be sex-biased. Each density curve in Supplementary Figure 2h for CNA-gains reflects the p-value distribution for one gene. While most down-sampled proportions reflected our initial observation of higher male-derived proportions, we removed 1,373 hits where the median down-sampled p-value did not pass the $p=0.05$ threshold. We retained 2,912 hits that did meet this requirement. We added detail in the text, to Figure 2, and in Supplementary Table 7 to reflect these changes.

Mutation Signatures: ID1-attributed & ID8-positive removed

We found that the median down-sampled p-value for the proportion of ID1-attributed mutations did not meet the $p=0.05$ threshold (Supplementary Figures 2m & o). In addition, the effect sizes showed that the initially observed male-bias was inconsistent to tumour subtype-resampling. We removed this finding and the finding of sex-biased proportions of ID8-positive sampling for the same reason. Other mutation signatures were retained in the manuscript (Supplementary Figure 2j, k, l & n).

We added discussion on limitations of a pan-cancer analysis for sex differences at Lines 315-319:

“Finally, there are imbalances across covariate sample sizes, such as over-representation of some tumour types in pan-cancer analysis. We evaluated these imbalances using down-sampling analysis and rejected results that were biased by these imbalances. Nevertheless, pan-cancer analysis is dependent on the tumour subtypes included in the cohort and some findings may reflect subtype-specific trends rather than general characteristics across all cancers.”

As well as detail in the methods regarding this subsampling analysis at Lines 621-625:

“For pan-cancer findings, we evaluate the effect of unbalanced tumour subtype sample sizes by repeatedly and randomly down-sampling to the median subtype sample size ($n_{\text{median}}=48$). For each down-sampled dataset, we repeat the relevant univariate test and record the effects and p-value (Supplementary Figure 2). We repeat this 10,000 times for each finding and reject those where the median down-sampled p-value is greater than the $p=0.05$ threshold.”

2. Does it make sense to include an unequal number of male and female samples. Did the authors do any sort of permutation test to confirm that the proportions in males (for which there are more) are significant after permuting and downsampling to the same number as females. Overall, some of the differences could be due to sampling bias. I appreciate that the authors attempted to get around this by investigating proportions, but these can still be affected. Can the authors more thoroughly address the concerns of sampling bias and unequal sample sizes between males and females?

Yes, this is a good question. To address it we performed repeated down-sampling for a selection of analyses in tumour types with imbalanced sexes, or when either male or female donors accounted for >60% of samples in the cohort. We then randomly down-sampled the larger group to the size of the smaller size and performed the relevant univariate non-parametric test. We repeated this 10,000 times and saved the p-values and measures of effect sizes for comparison. Down-sampled proportions, medians and p-values are presented in Supplementary Figures 1, 3, 8, 9, 11-12.

In most cases, the median down-sampled p-value passed our $p=0.05$ threshold and the down-sampled effects (proportions or medians) agreed with initially observed effect directions. For example for sex-biased CTNNB1 mutation in Liver-HCC, we down-sampled male-derived samples from $n=226$ to $n=88$, matching the number of female-derived samples. We found that in all 10,000 down-sampling iterations, the proportion of male-derived samples with a CTNNB1 mutation was consistently higher than the proportion of affected female-derived samples (Supplementary Figure 1a). We therefore retained all sex-biased findings with some exceptions regarding CNA results.

Our repeated down-sampling on sex analysis for sex-biased CNAs was performed for each gene (Supplementary Figure 9). While the analysis showed that most sex-biased CNA findings remained significant, there were some cases where the median down-sampled p-value did not meet the $p=0.05$ threshold. Specifically, of 2,609 sex-biased losses in Liver-HCC, 2,078 genes passed the sex down-sampling p threshold and 531 genes failed. All 1,986 sex-biased gains in Kidney-RCC passed. And of 2,912 sex-biased genes in Pan-cancer analysis, 2,907 passed the p threshold and 5 failed. We have updated the CNA section, Figure 2 and Supplementary Tables 5 & 7 with these updated numbers. We have also updated Figure 2, Supplementary Tables 6 & 7 and methods at Lines 626-631:

“For both pan-cancer and tumour subtype-specific findings, we evaluate the effect of unbalanced sexes when either female or male donors account for $>60\%$ of samples. We down-sample to the smaller number of samples, repeat the relevant univariate test and record the effects and p-value (Supplementary Figures 2, 3, 8, 9, & 11). We repeat this 10,000 for each finding and reject those where the median down-sampled p-value is greater than the $p=0.05$ threshold. We present the median down-sampled p-values throughout Supplementary Tables 2-8”

3. While the mutations that occurred in male or female tumors more often may reflect the true biology, did the authors also account for pre-existing germline mutations? That is, if the females, for whatever reason, were sampled in such a way that their germline already had the CTNNB1 mutation, then they wouldn't show up as a tumor-specific mutation in female tumors. But, they would all have it. I'm just curious how pre-existing germline mutations could bias the estimates of sex differences. Perhaps the authors will consider this outside the scope of the paper, but worth mentioning, if there is space.

This is an excellent point. We looked for pre-existing germline mutations matching known driver variants as described in the PCAWG driver data (Ref 14; doi:<https://doi.org/10.1101/190330>) and pathogenic germline variants from a study of “Pathogenic Germline Variants in 10,389 Adult Cancers” (PMID: 29625052). We did not find driver germline variants in CTNNB1 (Liver-HCC) or TERT (Thy-AdenoCA) tumours. We added to Lines 80-82:

“We did not find pathogenic germline variants in TERT or CTNNB1 that might bias the detection of sex-associated somatic mutations in these genes.”

And added to Lines 327-329:

“We will also be able to also leverage germline data to assess whether there are sex-biases in inherited variants that affect the variants we observe in somatic mutation profiles.”

4. I don't know what Box-Cox transformed mutation load is, and would need someone else to evaluate this statistical framework. How does this account for potential confounders?

A Box-Cox transformation is a statistical technique used to transform non-normal data into a more normal-like shape. It is a family of power transformations that does not change the monotonicity of the data, but make it better suited for multivariable regression techniques like those we use here. We do not draw inferences at the Box-Cox step directly, and appreciate that this approach is not sufficiently clear for a general audience. We have updated the text to try clarify what the Box-Cox transformation is and how it was used in our study at Lines 91-92 which now reads:

“The Box-Cox transformation applies a power function to modify the shape of a variable’s distribution to better approximate a normal distribution. It preserves monotonicity and is often applied to make the data more suitable for regression analysis (Online Methods).”

5. The parenthetical reference to higher mutational prevalence in male-derived samples should be in a table. It is exceedingly difficult to parse in the very long comment, as it stands now (lines 95-99).

Full mutational prevalence results are now in Supplementary Table 3 and Lines 96-98 now read:

“Across all pan-cancer samples, we identified higher mutation prevalence in male-derived samples in all three contexts (coding LNR $q = 7.5 \times 10^{-4}$, non-coding LNR $q = 6.5 \times 10^{-4}$, overall LNR $q = 1.9 \times 10^{-6}$; Supplementary Table 3).”

6. Why would a patient's age be more important than the age of the tumor? Or maybe they corrected for age of the tumor, but that seems unlikely.

We do not have data describing age of the tumour, only for patient age at diagnosis. We adjust in these analyses for age at diagnosis, which is associated with myriad epidemiological and genomic characteristics (PMID:23612461), including mutation density (PMID:26384365), mutation signatures (PMID:24657537, doi:<https://doi.org/10.1101/322859>) and specific driver genes such as IDH1 in gliomas (PMID:28110298). By controlling for age at diagnosis we hope to better isolate the sex-biases.

7. How did the authors adjust for ancestry? Did they do ancestry analysis, or did they use reported race? The latter would be problematic. The former would be curious as to whether they used bins, or proportions of inferred ancestry, and relative to which populations... 1000 genomes?

Yes, Reviewer #1 had similar comments and on re-reading appreciate that this was fully unclear in our initial submission! Ancestry was imputed by the PCAWG-8 working group based on whole-genome sequencing of the reference (typically blood) sample. Briefly, ADMIXTURE (PMID:19648217) was used to estimate the ancestry of donors based on 4,235 ancestry-informative SNP markers. These markers were measured from the non-tumour reference sample sequenced for each patient in this study (typically blood). The markers were chosen to discriminate between European, East Asian, African, Native American, and South Asian populations. This method was validated using samples from the 1000 Genomes project. Full PCAWG methods are provided as an attachment to the reviewers and we have added detail on ancestry data to Methods at Lines 595-598:

“Covariate data includes genomically matched sex, age at diagnosis, and ancestry as imputed using ADMIXTURE²² by the PCAWG-8 working group based on germline SNP profiles determined by whole-genome sequencing of the reference sample.”

8. For the 15 tumor types with at least 15 samples, do the authors mean 15 of each sex, or 15 total - please clarify in the main text.

We have clarified the text at Lines 101-103 to now read:

“We also investigated somatic SNV burden in each of the 23 individual tumour subtypes with at least 15 samples ($n_{\text{male}} + n_{\text{female}} \geq 15$), applying the same statistical approach using tumour subtype-specific models (Supplementary Table 1).”

9. I don't understand the rationale for the test, or the potential mechanism of a promoter mutation being associated with a globally increased codon mutation burden (but not genome-wide mutation burden). That is, why would it only lead to more mutations in codons, but not elsewhere in the genome? What is the possible mechanism? And what are the chances of observing this correlation by chance?

We apologize for the lack of clarity. We have revised the analysis to look at global mutation burden, rather than solely coding mutation burden. We found that as the reviewer suspected, TERT promoter mutation was associated with overall SNV density. We have revised the Lines 118-123 to read:

“We therefore looked for associations between SNV burden with CTNNB1 mutation in hepatocellular cancer, and with TERT promoter mutation in thyroid cancer. We did not find a significant association between SNV burden and CTNNB1 mutation in hepatocellular cancer. In thyroid cancer however, TERT promoter mutation was associated with increased overall mutation burden ($\text{median}_{\text{TERT-wt}} = 0.32 \text{ mut/Mbp}$ vs $\text{median}_{\text{TERT-mut}} = 0.82 \text{ mut/Mbp}$, $u\text{-test } p = 7.9 \times 10^{-8}$).”

10. I do not understand what the "matched mutational timing data" is? Can the authors explain this, and how they can come to the conclusion that TERT promoter mutation frequency is associated with sex-biased coding mutational load because of it?

We used tumour subclonal reconstruction data generated by the PCAWG-11 working group, which includes mutation timing data describing whether a mutation event occurred clonally (i.e. in the trunk) or subclonally (i.e. in the branches). We added a citation referencing the PCAWG tumour evolution preprint (Ref 15; doi:<https://doi.org/10.1101/161562>) and have removed speculation on the association between TERT promoter mutation frequency and sex-biased coding mutational load.

11. Were polyclonal versus monoclonal determined from one bulk tumor sample? This seems difficult at best to infer, without multi regional sampling. Also, is the age of the tumors known? Is it possible that female tumors were detected earlier, and so were more likely to be monoclonal?

The subclonal reconstruction data is based on single-sample tumour WGS. The PCAWG reconstruction calls are based on a consensus strategy using six copy number callers and 11 subclonal reconstruction callers. The method is described in the source paper on the evolutionary history of PCAWG tumours (Ref 15; doi:<https://doi.org/10.1101/161562>). While multi-region sampling is certainly preferred to produce

reconstructions with greater resolution, single-region reconstruction is still able to produce phylogenies with meaningful clinical implications (PMID:29681457). However, we have added caveats to this analysis to the Discussion at Lines 324-327.

“Our results are based on single region sequencing, which can bias the clonal reconstruction for these tumours. Future work sampling multiple regions will allow us to detect sex differences in more precise reconstructions at a greater resolution.

We found that female-derived biliary cancers were more likely to be polyclonal than male-derived, which might suggest these female-derived tumours were detected later. While we do not know the age of the tumours, we do have tumour stage for this cancer subtype as a proxy. We do not find an association between tumour stage and sex in biliary cancers. Moreover, the tumour-specific model for biliary cancer includes pathologic stage, and sex remains significant after adjusting for this.

12. Can the authors comment on why there are 4,285 gene gains in males across all tumor types, but none that are female-biased? This seems, on sheer numbers alone, to be a technical artifact of some sort.

Yes, the Reviewer raises a very valid concern. We also thought the exclusively male-dominated hits were curious and investigated such artefacts. After incorporating the Reviewer’s suggestions by performing repeated down-sampling analysis on tumour subtype and sex to evaluate imbalances in the data, we reduced this set of significantly sex-biased CNAs to 2,907 male-dominated gains. We believe the large number of male-biased pan-cancer gains is likely a combination of real biology and statistical limitation:

Copy number changes are large structural events that often impact hundreds of genes. While our analysis is done at the gene level, the results represent a far smaller number of chromosome segment- or arm-level events. The results we observe here follow trends we observed in our previous TCGA study (Ref 11; PMID:30275052). In TCGA data, we identified 3,251 genes that were more frequently gained in male-derived samples and no female-dominated CNAs. The effect sizes of these significant sex-biases were similar between the TCGA and this study (up to 10% difference in frequency). This is despite differences in the tumour subtypes included and tumour subtype sample sizes between TCGA and PCAWG data.

Finally, CNAs that occur more frequently in female-derived samples may have smaller effect sizes that we are underpowered to detect. In our PCAWG pan-cancer analysis, the strongest female-biased occurred 1.6% more frequently in females than males (10.6% in females vs. 8.96% in males). In contrast, all male-biased CNAs were at least 3% more likely to occur in males.

Thus we believe our statistical approach has identified male-dominated gains that are robust to imbalances in tumour subtype and sex sample sizes. It is likely that our methods are too conservative to detect small effect sizes and we are underestimating the number of genes affected by sex-biased CNAs, including sex-biased losses and female-dominated CNAs.

13. The case of female-specific losses showing up in liver cancer, but no losses in the full analysis is a good argument for why the tumors shouldn't be lumped together.

Our methods are likely underpowered to detect female-dominated losses. The female-dominated losses in liver cancer are associated with up to 30% difference in frequency, but this effect is diluted when

combined with other tumour subtypes that do not have the same sex-difference in loss frequency for these genes. This is suitable behavior for our pan-cancer approach, since we want to capture sex-biases common across several tumour subtypes and not just one.

14. The "Pan-cancer analysis of whole genomes" cited for us to find the genomics methodology is a 2017 preprint: <https://www.biorxiv.org/content/10.1101/162784v1>. Can the authors confirm that this is the most updated version? It notes there will be a complete version updated, but I couldn't immediately find it. Then, upon investigation, it is nearly impossible to find the exact pipeline that was run. Did the authors really align to version 37 of the human genome? Version 38 was released in 2013. There is no excuse for using version 37.

The 'Pan-cancer analysis of whole genomes' marker paper is currently under review and apparently the Nature editors have asked for the preprint to not be updated. We have attached the latest version of the full marker paper for the reviewer's consideration. While we have updated our references to the most up-to-date PCAWG information, these papers are yet to be released in their final form.

The PCAWG project was initiated in 2014 when GRCh37 was still the prevailing reference build, and estimates of the project's completion-date were significantly optimistic in hindsight. As a result, GRCh37 was chosen by the PCAWG Steering Committee (Lincoln Stein, Gaddy Getz, Josh Stuart, Jan Korbel and Peter Campbell) because of the large amount of correlative data available. Current releases of PCAWG data are all based on GRCh37: this is true for the full set of papers in the PCAWG-basket, and thus we cannot avoid that older build. However the PCAWG technical team is now rerunning the core variant analysis pipeline on GRCh38 with public data release targeted for 2020.

15. Did the authors look at whether there was any difference in the distribution of sequencing types used for males versus females? That is, how many male samples were run on the HiSeq 100bp versus 150bp? How many female samples for each? This could 100% explain discrepancies.

We compared read lengths between sexes for all samples and did not find a sex-bias for any tumour subtype or across all pan-cancer samples (Supplementary Figure 13). Indeed, 100 bp paired end sequencing was used for the vast majority of samples. In addition, our assessment of sequencing quality control finds that male- and female-derived samples are of comparable sequencing quality across a number of metrics (Supplementary Figure 13, Supplementary Table 9).

16. How did the male/female samples separate out among the 75 donors that were flagged for QA for "a variety of reasons"? Including an unexpectedly high fraction of paired reads mapping to different chromosomes, unusual mutational signatures, and contamination?

Of 75 flagged donors, 51 donors contributed to projects included in our study – we excluded projects focusing on sex-specific tumours such as prostate and ovary cancer. There were 16 female and 35 male donors (31% vs. 69%). This is statistically indistinguishable from the sex breakdown of the donors in our study (39% female vs. 61% male; $p = 0.36$; proportion test).

Reviewer #3

In this manuscript, Li et al. evaluate the genomic differences between cancers in males and females using pan-cancer whole-genome sequencing data from the ICGC Pan-Cancer Analysis of Whole Genomes (PCAWG) project. The paper largely focuses on statistical analysis and identification of driver genes, mutation load, tumor evolution, copy-number changes, and mutational signatures with differences between the sexes.

While this manuscript is largely descriptive, it does try to answer an important enigma in cancer epidemiology: why are sex differences observed in most cancers outside sex organs? Indeed, understanding the molecular mechanisms underlying such difference will have far-reaching implications. Unfortunately, I do not find this manuscript to be appropriately addressing this question or to be providing any significant insights beyond previous publications. Further, I do have major concerns with the performed (or at least the description and reproducibility of the performed) statistical analysis. At its current state, I cannot recommend this manuscript for acceptance as major work/re-analysis is required.

We thank the Reviewer for their careful review, and we appreciate the importance of better understanding the limitations of our statistical approach. We have undertaken a series of analyses outlined below to attempt to improve the statistical analysis and its reporting, and greatly appreciate the Reviewer's insight.

Concerns about Statistical Analyses:

The manuscript heavily relies on statistical analysis. However, this analysis is neither properly described nor particularly reproducible.

For example, in the section for driver gene, page 2 line 60 states: "adjust for tumour subtype, ancestry and age (Online Methods)". As far as I can tell, there is nothing in the Online methods discussing the approach for adjusting for these parameters. Further, derivation of these parameters is not described. What was the granularity of tumor subtype (e.g., did you separate breast cancer into multiple subtypes)? More importantly, how were sex and ancestry determined (self-reporting vs inferring from the available genomics data)?

We apologize for the omission and lack of detail in Online Methods. We adjust for the parameters by including them as independent variables in multivariable linear/logistic regression for each analysis. We have added detail at Lines 604-609:

"At stage two, we further investigate these putative sex-biases by using multivariate linear or logistic modeling to account for potential confounders using bespoke models for each tumour subtype. Confounders were included as independent variables in each model. Supplementary Table 1 describes the model variables for each tumour context, as well as detail on when analyses included multivariate modeling. Variables were included based on availability of data (<15% missing), sufficient variability (at least two levels) and collinearity."

We have also updated Supplementary Table 1 to present a full description of model variables. The table presents a complete summary of all modeling performed across all genomic features and tumour subtypes. The first sheet presents the model descriptions and when multivariable model were used.

Subsequent sheets present the granularity and breakdown for each variable. Specifically, tumour subtype is separated by histological subtype for biliary (papillary cell, cholangiocarcinoma), kidney (clear cell, papillary cell), liver (hepatocellular, cholangiocarcinoma, fibrolamellar), non-Hodgkin's lymphoma (diffuse large B-cell, follicular, Burkitt lymphoma, marginal zone B-cell, post-transplant lymphoproliferative disorder, not otherwise specific) and thyroid (classical, follicular, columnar) cancers. There is no additional histological separation for the chronic lymphocytic leukemia and melanoma analyses.

Clinically reported sex was genomically-verified with the sequencing data, and ancestry was imputed by the PCAWG-8 working group based on germline SNP profiles determined by whole-genome sequencing of the reference sample. We have included the full ancestry imputation methods as an attachment to Reviewers and included detail in our Methods at Lines 595-598:

“Covariate data includes genomically matched sex, age at diagnosis, and ancestry as imputed using ADMIXTURE²² by the PCAWG-8 working group based on germline SNP profiles determined by whole-genome sequencing of the reference sample.”

It is also unclear whether these (or any adjustments) were made for the other sets of statistical analysis (e.g., tumor evolution, copy-number changes, etc.). I am not sure if Supplementary Table 1 is applicable to all analysis or just the ones in the driver gene section. Moreover, there are at least two additional parameters that need to be included in the adjustments: stage and treatment (i.e., prior chemotherapy; a number of the PCAWG samples are post chemotherapy).

We apply the same statistical framework and models for all sets of statistical analyses and have added information in Supplementary Table 1 and Online Methods to specify when multivariate regression was performed. In short, we always start with a non-parametric univariate test (Pearson's χ -squared proportion or Wilcoxon-Mann-Whitney test) and use a false discovery rate (Benjamini-Hochberg procedure) threshold of 10% to determine putative sex-biased events. We then use the same tumour subtype-specific model variables in our multivariable regression models (linear or logistic) to evaluate these putative sex-biases. We again use the Benjamini-Hochberg procedure to correct for multiple testing and a 10% FDR threshold to select statistically significant results. For some tumour subtypes, the multivariate step is never performed because there are no univariate hits to evaluate.

We added stage as a model variable when possible, including in models for Biliary cancer, chronic lymphocytic leukemia and melanoma, but most tumour subtypes lack any stage data, and others have a large number of NA values (>20%) which significantly complicates statistical modeling. The treatment variable has a similar problem where it is not available in the vast majority of cases. We there apply an extended model that incorporates stage and/or grade as additional variables when available. This results in a better specified model but its use is limited to the reduced number of samples with stage and/or grade data available. We present these results alongside all other results in Supplementary Tables 2-8. To summarize the results after extended modeling:

Driver-SNV analysis

Stage was available for 75% and grade was available for 80% of Liver-HCC patients, and were included in an extended model alongside variables sex, race, age, histological subtype and source project. Sex remained significantly associated with CTNNB1 mutation. Treatment data is not available for 44% of Liver-HCC tumours or is recorded as 'no treatment' (n=172),

compared with ‘chemotherapy’ (n=1) and ‘surgery’ (n=3). There is no stage, grade or treatment data for Thy-AdenoCA.

SNV density analysis:

We found sex-biased SNV density in three tumour types: Liver-HCC, Kidney-RCC and Thy-AdenoCA. The sex-association remained significant in the extended Liver-HCC model (including stage and grade), and as mentioned above, we did not extend the Thy-AdenoCA model. For Kidney-RCC, grade (available for 51% of samples) is highly correlated with stage (available for 40% of samples). Because grade is available for more samples, we included grade in the extended model. Treatment data was either not available or ‘no treatment’. The sex-association was significant in the extended Kidney-RCC model.

Polyclonality & SV timing

We found significant polyclonality and SV timing results for Biliary-AdenoCA. The Biliary-AdenoCA model already includes stage in its core analysis. Treatment data is also unknown or ‘no treatment’ for all except one sample.

PGA analysis

We found sex-biased PGA in Liver-HCC and Kidney-RCC. The sex-associations remained significant in the extended Liver-HCC model (including stage and grade) and Kidney (including grade) in all except one case. In Kidney-RCC, the association between sex and genomic instability on chromosome 12 was only trending at q-value=0.16. We have updated the text at Lines 171-174 to reflect this:

“On further scrutinizing these sex-PGA associations using extended models, we found that grade was a likely confounder in renal cell cancer, though the sex effect after correcting for this variable was still trending (extended MLR $q = 0.16$).”

CNAs

We found sex-biased CNAs in Liver-HCC and Kidney-RCC. Of 2,078 sex-biased losses we identified in Liver-HCC, 1,894 were significant after additional analysis using the extended model. Of 1,986 sex-biased gains in Kidney-RCC, 969 were significant ($q < 0.1$) and the 1,017 were trending with q-values < 0.17 . We have added text at Lines 191-193 to reflect this:

“Using an extended renal cell cancer model accounting for grade, we obtained a high confidence set of 969 genes altered by sex-biased gains (extended model $q < 0.1$), with the remaining 1,017 genes having a trending sex effect (extended model $q < 0.17$).”

And at Lines 201-202:

“Extended modeling in Liver-HCC incorporating stage and grade led to a list of 1,894 high confidence sex-biased genes (extended model $q < 0.1$).”

Mutation Signatures

We identified tumour subtype-specific sex-biased signatures in Liver-HCC, Lymph-BNHL and Lymph-CLL. We added stage (59% of samples) to the extended Lymph-BNHL model. Lymph-CLL included stage in its core model. All sex-biases remained significant with the exception of ID3-positive in Liver-HCC, where the effect was only trending $q = 0.12$. We have updated the text at Lines 258-260 to reflect this:

“Using our extended hepatocellular model to further scrutinize these signatures, we found that all remained sex-biased after accounting for these

variables except in ID3, where the effect was trending (extended model q-value = 0.12).”

Extending the models to include stage or grade results in a substantial reduction in usable sample size. Despite this, this analysis shows that the majority of our sex-biased findings remain significant after accounting for stage or grade. We added detail on limitations of our modeling at Lines 306-314:

“Secondly, the tumour subtype-specific results are bound by subtype sample size, and lack of annotation data restricts the ability to account for confounding variables. It is therefore important to consider these results within context of the multivariable models used, which do not directly capture characteristics such as tobacco smoking history. Many of our base multivariate regression models omit stage and grade due to a large number of missing values. We follow up this base regression with extended modeling as an additional level of scrutiny. While these extended models do include stage or grade, they are run on a much smaller (up to 50%) subset of the data and there is a corresponding loss of statistical power.”

The Box-Cox Transformation is a good approach to reduce non-additivity, non-normality and heteroscedasticity of data. However, the transformation only reduces these problems and it does not eliminate them. The authors should really use robust statistical approaches, e.g., robust regression instead of regression as well as robust logistic regression instead of logistic regression.

We have attempted to apply robust methods as recommended, but these tend to be highly sensitive to collinearity between variables and to allow them to converge we would have to omit key confounding factors like ancestry from our models. We explored implementations based on a robust M estimators implemented in R, including rlm (MASS), lmRob (robust) and glmrob (robustbase). We acknowledge the weaknesses of our regression-based approach and have added caveats at Lines 304-307:

“These results should be taken within context of a number of caveats. While we use techniques like the Box-Cox transformation to make the data better suited for our statistical methods, there are likely characteristics that our models are unable to account for. An alternate approach using robust modeling may be better suited for future analyses.”

Lastly, a more minor note, I cannot seem to find the supplementary materials of this manuscript. Were these missed as part of the submission? Nevertheless, I was able to download them from the bioRxiv version of the manuscript. Also, there are a number of typos throughout the manuscript (e.g., page 6 line 246: “asociated”).

We apologize for this omission. We provide all updated supplementary materials, and have revised and spellchecked text in this resubmission.

Concerns about Reproducibility:

This paper relies solely on statistical analysis and it is essential to be able to peruse the actual procedures as well as to use the code to reproduce the reported results. Indeed, the authors can provide one table (or even an R data frame) with the information of the patient used in the analysis

(sex, cancer type, age, ancestry, etc.) and another table(s) with the genomics features used for associating sex differences. Further, I would suggest posting all of their code in a repository (e.g., GitLab) and, thus, allowing reviewers and readers to really examine their analysis as well as to easily run the code and reproduce the results.

All data used in the analysis of this paper are available through <https://dcc.icgc.org/releases/PCAWG> and <https://www.synapse.org/> through the accession links provided in methods. It is PCAWG policy not to reproduce these in different versions, as requested by the Reviewer here, as that could lead to confusion. We have provided all the data resource links throughout the Methods with the stable releases. We use pre-existing R packages and functions which are described throughout and have not done algorithm development. We have also added detail on PCAWG data acquisition and methods at Lines 582-594:

“Somatic and germline variant calls, mutational signatures, subclonal reconstructions, transcript abundance, splice calls and other core data generated by the ICGC/TCGA Pan-cancer Analysis of Whole Genomes Consortium is described in the marker paper¹³ and available for download at <https://dcc.icgc.org/releases/PCAWG>. Additional information on accessing the data, including raw read files, can be found at <https://docs.icgc.org/pcawg/data/>. In accordance with the data access policies of the ICGC and TCGA projects, most molecular, clinical and specimen data are in an open tier which does not require access approval. To access potentially identification information, such as germline alleles and underlying sequencing data, researchers will need to apply to the TCGA Data Access Committee (DAC) via dbGaP (<https://dbgap.ncbi.nlm.nih.gov/aa/wga.cgi?page=login>) for access to the TCGA portion of the dataset, and to the ICGC Data Access Compliance Office (DACO; <http://icgc.org/daco>) for the ICGC portion. In addition, to access somatic single nucleotide variants derived from TCGA donors, researchers will also need to obtain dbGaP authorisation.”

And Lines 715-717:

“The core computational pipelines used by the PCAWG Consortium for alignment, quality control and variant calling are available to the public at <https://dockstore.org/search?search=pcawg> under the GNU General Public License v3.0, which allows for reuse and distribution.”

Concerns about Novelty/Impact of Findings:

Even excluding the technical concerns, I am worried about the novelty of the findings. The authors use quite a lot of space for CTNNB1 but sex difference in CTNNB1 were previously reported in both reference 10 & 11 of the current manuscript. The sex difference in the TERT promoter were reported in some of the early melanoma manuscripts (PMID: 24463461) albeit not in thyroid cancer. I am both surprised and concerned about the finding of sex differences in ALB and PTCH1. The mutations in these genes are in the coding sequence, however, they were not identified in the twice larger (~4500 exome) TCGA paper on differences between males and females (PMID: 27165743; reference 10 in the current manuscript). How do the authors explain that? My concern is that the number of mutations in these genes is low and they might be an artifact of the technical methodology. Further, if these are correct associations, one would expect to observe them in the analysis including larger number of samples.

We have removed the analysis of pan-cancer sex-biased driver mutations. They were indeed driven by specific overrepresented tumour subtypes (Liver-HCC and CNS-Medullo). We have also investigated the balance of tumour subtypes in our other findings and made changes to better reflect pan-cancer results. Broadly we believe that recapitulating these previously known findings gives confidence in our analysis, while the novel unexpected ones around mutational signatures are intriguing and will absolutely require validation in future large cohorts as the reviewer suggests. We now try to make this caveat of our study much more clear in the Discussion, which now reads at Lines 320-322:

“Future increases in sample size and robust associated annotation will allow for the detection of smaller effects and the control of more confounders. Such large datasets are critical in validating the preliminary findings we have described in this study.”

The analysis of genome instability and copy-number changes does not seem to bring much more compared to the prior TCGA study. Again, reference 10 had a large sample size but used whole-exome sequencing and copy-number arrays. Nevertheless, the whole-genomes do not seem to be bringing much in terms of findings as the results seem very similar to me. What are the novel findings compared to prior studies?

We investigate genome instability in greater granularity in this study by breaking down PGA by chromosome. This indicates that specific genomic regions are structurally altered or mis-repaired at different rates between the sexes. Our gene-specific analyses go on to examine the effect of these sex-biased CNAs. In addition, the CNA calls from TCGA and PCAWG are generated by fundamentally different technologies with distinct sensitivities and calling approaches. We are confident that confirmation of previous findings using data that is orthogonal in both sampling and generation provides validation that these are robust and reproducible biological findings.

There has been no previous pan-cancer-wide investigation of sex-differences in whole genome sequencing data. We present a complete landscape of all our results, including those validating previous findings, as well as hitherto undescribed ones. These include the first analyses of sex-biases in subclonal reconstruction and mutation signatures. These results suggest how and when sex differences may emerge in the mutational landscape. The mutations signatures results are especially intriguing and suggest ways of accounting for mutagenic processes in the absence of annotation data in future analyses of genomic sex differences.

Finally, on the subject of mutational signatures, comparisons of the numbers of samples with a signature versus the numbers of samples without a signature is, unfortunately, not appropriate. To avoid overfitting, the PCAWG signature analysis is using a penalty for assigning signatures with each signature required to explain at least 5% of the overall observed pattern. This results in losing lower burden mutational signatures in samples with large numbers of mutations. For example, the authors report sex difference in total mutational burden of hepatocellular carcinoma. As observed, this results in certain signatures not being assigned to female hepatocellular carcinomas when compared to male hepatocellular carcinomas. At the very least, the authors should correct for total mutational burden in their sex analysis of mutational signatures. Preferably, the authors should compare the distributions of the numbers of mutations of the signatures between males and females when the signatures have been attributed to these samples (i.e., more than zero mutations).

The reviewer brings up another excellent point. We have included mutation burden as a variable in all analyses of signature-attributed mutation, and also performed Kolmogorov–Smirnov tests to compare

their distributions. We added these steps and used repeated down-sampling analysis to identify results biased by uneven tumour subtype and sex sample sizes. We have revised our multivariate model results to reflect the updated models, and present the K-S p-values in Supplementary Table 8. In pan-cancer analysis, signatures SBS1, SBS17a, SBS17b and ID5 were found to be sex-biased. Signatures SBS16, SBS40, ID1 and ID8 were not significantly sex-biased and removed from our manuscript. In Liver-HCC, all signatures remained sex biased, though the q-values are higher than previously reported. These signatures are SBS1, SBS16, ID3, ID8, ID11 and ID1. In Lymph-BNHL, SBS17b remained sex-biased. The updated methods identified a sex-bias in SBS17a. We also include newly detected sex-biases in Lymph-CLL for signatures SBS40, which was more active in female-derived tumours.

Reviewers' comments:

Reviewer #1 (Remarks to the Author):

I think this is very strong and important work. I expect it will push the field forward considerably.

Josh Rubin

Reviewer #2 (Remarks to the Author):

I thank the authors for thoroughly addressing my comments. I think the paper is much stronger, and will serve as an excellent reference for people moving forward in the field, especially for including SABV in cancer genomics.

(This is to the editors, but I want to make sure the authors see it.) I also find that it is unconscionable for a journal/editors to not allow update of a preprint, and appreciate the authors sharing the updated version for review here.

Sincerely,
Melissa A. Wilson, Ph.D.
Arizona State University

Reviewer #3 (Remarks to the Author):

The authors have performed a significant modification of the original manuscript and addressed my original concerns. They have extended their analyses and greatly improved the manuscript. Nevertheless, I still have a number of concerns with the revised manuscript.

Statistical concerns: The authors state "We repeat this 10,000 times for each finding and reject those where the median down-sampled p-value is greater than the $p=0.05$ threshold." I do not understand why using "the median down-sampled p-value" for comparison is statistically sound. I do not think this will reflect having 5% of the results being false-positive (i.e., happening purely by chance); the authors can integrate the p-curve from the 10,000 repetitions and will get the actual false-positive rate, which I presume will be much higher. I think the authors should be using a more conservative approach: repeat this 10,000 times for each finding and reject those where $p>0.05$ occurs more than 500 times.

Novelty concerns: Figure 4 in the manuscript is very informative and useful. Unfortunately, it immediately raises the question of novelty. Everything with a "*" was previously known and authors should adjust "* Exome-based finding" to "* Previously known from exome studies". Additionally, some of the reported sex differences were previously known from genome studies. For example, liver gender bias for SBS16 (and CTNNB1 being caused by SBS16 resulting in gender bias for CTNNB1) was previously reported in PMID: 29101368. The authors should probably add these with "*** Previously known from genome studies" and check for other results in the WGS studies published for individual cancer types. Looking at Figure 4 with all the "*" and potential future "***", I am a bit concerned about the novelty of the manuscript compared to previous findings.

Reproducibility concerns: The authors state "All data used in the analysis of this paper are available through <https://dcc.icgc.org/releases/PCAWG> and <https://www.synapse.org/> through the accession links provided in methods. It is PCAWG policy not

to

reproduce these in different versions, as requested by the Reviewer here, as that could lead to confusion." Unfortunately, this is not particularly useful as the PCAWG release has 27 folders with thousands of files. Only for mutational signatures and driver mutations, there are multiple versions of results from the different tools that were applied to the PCAWG data. I understand that the authors cannot reproduce data in supplementary. However, can they provide in a Supplementary note and/or table a set of links to the files that they used in their analysis. For example:

Age information: <https://dcc.icgc.org/releases/PCAWG/XXX/file1.txt>

Ancestry information: <https://dcc.icgc.org/releases/PCAWG/XXX/file2.txt>

Mutational signatures: <https://dcc.icgc.org/releases/PCAWG/XXX/file3.txt>

Driver mutations: <https://dcc.icgc.org/releases/PCAWG/XXX/file4.txt>

We thank the reviewers and the editorial staff for their thoughtful review of our revised manuscript. We have strengthened our down-sampling method, addressed data accession concerns and submit an improved manuscript. We have addressed additional specific reviewer's comments below, with the key changes being:

- We continue to use a down-sampling method to prune hits that might be functions of imbalanced cohorts. In this resubmission, we use a bootstrapping down-sampling approach to obtain better estimates of the median/proportion. This allows us to formally use the distribution of effects to test the robustness of the sex-bias. We use both this effects-based approach and a p-value filter to reject hits that are biased by these imbalances.
- We added detailed data access information in methods and Supplementary Table 10 to point directly at files used in our analysis. We also provide a greatly-expanded description of these data in the Online Methods.

Reviewer #3

The authors have performed a significant modification of the original manuscript and addressed my original concerns. They have extended their analyses and greatly improved the manuscript. Nevertheless, I still have a number of concerns with the revised manuscript.

We thank the Reviewer for their review of our revised manuscript. We have added changes to our submission in response to the Reviewer's statistical, novelty and reproducibility concerns and appreciate the Reviewer's insight.

Statistical concerns: The authors state "We repeat this 10,000 times for each finding and reject those where the median down-sampled p-value is greater than the $p=0.05$ threshold." I do not understand why using "the median down-sampled p-value" for comparison is statistically sound. I do not think this will reflect having 5% of the results being false-positive (i.e., happening purely by chance); the authors can integrate the p-curve from the 10,000 repetitions and will get the actual false-positive rate, which I presume will be much higher. I think the authors should be using a more conservative approach: repeat this 10,000 times for each finding and reject those where $p>0.05$ occurs more than 500 times.

We have updated our down-sampling approach from a jackknife to a bootstrap approach and use two criteria to evaluate the robustness of the sex-bias to tumour subtype and sex imbalances. Sampling with replacement gives a better estimate of the differences between male and female medians/proportions. This allows us to formally consider the effect sizes in this down-sampling analysis by generating a distribution of male-female differences and testing the null hypothesis that there is no difference between male- and female-medians/proportions. We calculate the 95% confidence interval, indicated as green regions in Supplementary Figures 1-3, 8, 9, 11 & 12. No overlap of the confidence interval with $x=0$ is evidence to reject the null hypothesis.

We also use the p-value distribution from the relevant proportion or Mann-Whitney U-test as these tests form the basis of our statistical approach. We continue using the condition that the median bootstrap p-value is less than 0.05, which suggests that the sex-bias is statistically significant in at least half of the down-sampled tests. However, we acknowledge the limitations of embedding a statistical test within a bootstrap approach with respect to power, changes in the effect size and decreased sample numbers.

Therefore, we only retain findings where (1) the median bootstrap proportion/Mann-Whitney U-test p-value is less than 0.05, and (2) there is no overlap of 0 within the bootstrap confidence interval. This indicates that not only do we observe a significant sex-bias in at least half of the tests performed, we also consistently observe a sex-bias across all down-sampled data and the finding is robust to tumour subtype or sex imbalances. We also would emphasize that this approach is inherently quite conservative, as our procedure requires each candidate to pass three separate statistical assessments:

- Univariable modeling with multiple-testing correction
- Multivariable modeling
- Assessment of skew

To our knowledge, no previous pan-cancer study has involved this stringency of assessment, which we believe greatly strengthens the confidence that can be drawn in the conclusions raised here.

All changes to significant sex-biases reported in this revised manuscript are as follows:

CNA s

The number of pan-cancer sex-biased losses decreased from 2,907 to 2,502 genes.
The number of Liver-HCC sex-biased gains decreased from 1,894 to 1,797.

Mutation Signatures

The sex-bias in signature SBS40 in Lymph-CLL failed both the confidence interval and p-value test and was rejected (Supplementary Figure 12C). We removed mention of this result from the text and Figure 3.

Novelty concerns: Figure 4 in the manuscript is very informative and useful. Unfortunately, it immediately raises the question of novelty. Everything with a “*” was previously known and authors should adjust “* Exome-based finding” to “* Previously known from exome studies”. Additionally, some of the reported sex differences were previously known from genome studies. For example, liver gender bias for SBS16 (and CTNNB1 being caused by SBS16 resulting in gender bias for CTNNB1) was previously reported in PMID: 29101368. The authors should probably add these with “*** Previously known from genome studies” and check for other results in the WGS studies published for individual cancer types. Looking at Figure 4 with all the “*” and potential future “***”, I am a bit concerned about the novelty of the manuscript compared to previous findings.

We have performed a literature search and updated Figure 4 as suggested by the Reviewer. We also thank the reviewer for bringing to our attention this relevant publication, which we have now referenced at Lines 248-251:

“A previous study²¹ described this sex-biased signature and an association between more CTNNB1 mutations and higher activity of SBS16 in an independent dataset; these findings agree with what we report here for PCAWG data.”

We believe that our findings of previously-reported results add to the credibility of both our statistical approach and of our novel findings. That we are able to validate previous findings using a separate methodology and independent dataset increases the confidence that these computationally described effects are real biological phenomena. It also increases the confidence that our novel discoveries of sex-biases in mutation density, mutational signatures and subclonal reconstruction are real and robust effects. To the best of our knowledge, the un-starred findings described in Figure 4 have not been previously reported. That being said, of course the literature is broad and it is possible some such findings have been seen in bespoke studies we and our reviewers have not identified.

Nevertheless, we believe this study provides a comprehensive resource of sex-biases in cancer genomics. It does not focus on any individual mutation type or cancer subtype, but performs an unbiased survey of the cancer landscape. A number of previously reported findings are from papers with broader focuses beyond just studies of sex-biases in cancer genomics. In PMID:29101368 for example, the authors describe the sex-biased SBS16 but do not mention a similar bias in signatures SBS1. The remaining majority of previously reported findings come from exome-based TCGA data and our study builds on those by investigating the added information from whole-genome sequencing. Indeed we consider it highly probable that if we did not report these results directly as part of the PCAWG consortium, other researchers would rapidly attempt to replicate them. To some extent, it is an interesting result that the number of novel findings is not larger, suggesting the exome carries many of the most interesting sex-differences present in many cancer types.

The inclusion of previously reported findings alongside new discoveries in our manuscript serve the dual purpose of validating existing knowledge and describing the unknown. The combination of replicated and novel sex differences expands the body of scientific knowledge of sex-biases in cancer genomics.

Reproducibility concerns: The authors state “All data used in the analysis of this paper are available through <https://dcc.icgc.org/releases/PCAWG> and <https://www.synapse.org/> through the accession links provided in methods. It is PCAWG policy not to reproduce these in different versions, as requested by the Reviewer here, as that could lead to confusion.” Unfortunately, this is not particularly useful as the PCAWG release has 27 folders with thousands of files. Only for mutational signatures and driver mutations, there are multiple versions of results from the different tools that were applied to the PCAWG data. I understand that the authors cannot reproduce data in supplementary. However, can they provide in a Supplementary note and/or table a set of links to the files that they used in their analysis. For example:

Age information: <https://dcc.icgc.org/releases/PCAWG/XXX/file1.txt>

Ancestry information: <https://dcc.icgc.org/releases/PCAWG/XXX/file2.txt>

Mutational signatures: <https://dcc.icgc.org/releases/PCAWG/XXX/file3.txt>

Driver mutations: <https://dcc.icgc.org/releases/PCAWG/XXX/file4.txt>

We appreciate the difficulty in finding specific files among a busy and complicated file system. Indeed the Reviewer’s comments echo data access concerns raised to the PCAWG Consortium and as a result the Consortium has standardized a resource reporting format. We have added detailed file links and descriptions in Supplementary Table 10 and included more detail in our data availability and processing section at Lines 381-399:

In addition, the analyses in this paper used a number of datasets that were derived from the raw sequencing data and variant calls (Supplementary Table 10). The individual data sets are available at Synapse (<https://www.synapse.org/>), and are denoted with synXXXXX accession numbers (listed under Synapse ID); all these datasets are also mirrored at <https://dcc.icgc.org>, with full links, filenames, accession numbers and descriptions detailed in Supplementary Table 10. Tumour histological classifications were reviewed and assigned by the PCAWG Pathology and Clinical Correlates Working Group (annotation version 9; syn10389158, syn10389164). Ancestry imputation was performed using an ADMIXTURE-like algorithm by the PCAWG Germline Cancer Genome Working Group based on germline SNP profiles determined by whole-genome sequencing of the reference sample (syn4877977). The consensus somatic SNV and indel (syn7357330) file covers 2778 whitelisted samples from 2583 donors. Driver events were called by the PCAWG Drivers and Functional Interpretation Group (syn11639581). Consensus CNA calls from the PCAWG Structural Variation Working Group were downloaded in VCF format (syn8042988). Subclonal reconstruction was performed by the PCAWG Evolution and Heterogeneity Working Group (syn8532460). SigProfiler mutation signatures were determined by the PCAWG Mutation Signatures and Processes Working Group for single base substitution (syn11738669), doublet base substitution (syn11738667) and indel (syn11738668) signatures. Quality control measures data was provided by the PCAWG Technical Working Group (syn5864470).

REVIEWERS' COMMENTS:

Reviewer #3 (Remarks to the Author):

The authors have addressed all my concerns related to the statistical analysis and reproducibility of the results presented in the manuscript. I do not have any additional technical concerns.

Frankly, I am simply unsure whether this manuscript is novel enough. Indeed, figure 4 summarizes the overall findings of the manuscript. In my opinion, all of the exciting results have either "*" or "***" (meaning they are confirmations of results from prior studies). All of the additional findings seem incremental to me as they relate either to relatively unknown mutation signatures found in liver cancers or to CNAs/coding/non-coding mutational burden in kidney and liver cancers. To be clear, this is a technically sound paper and having the summary of sex differences in figure 4 is rather nice. However, I will leave it to the editorial board to decide whether this paper meets Nature Communication's criteria to "represent important advances of significance to specialists within each field."